# Denoising-autoencoder-facilitated MEMS computational spectrometer with enhanced resolution on a silicon photonic chip

Jing Zhou[1,2,11], Hui Zhang ®[3,4,5,6,11], Qifeng Qiao[7], Heng Chen[1,2], Qian Huang[1,2], Hanxing Wang[1,2], Qinghua Ren[1,2], Nan Wang[1,2], Yiming Ma ®[1,2] ✉ & Chengkuo Lee ®[8,9,10] ✉

Silicon photonics enables the construction of chip-scale spectrometers, in which those using a single tunable interferometer provide a simple and cost-effective solution. Among various tuning mechanisms, electrostatic MEMS reconfiguration stands out as an ideal candidate, given its high tuning efficiency and ultra-low power consumption. Nonetheless, MEMS devices face significant noise challenges arising from their susceptible minuscule components, adversely impacting spectral resolution. Here, we propose a distinct paradigm of spectrometers through synergizing an easily-fabricated MEMS-reconfigurable low-loss waveguide coupler on a silicon photonic chip and a convolutional autoencoder denoising (CAED) mechanism. The spectrometer offers a 300 nm bandwidth and a reconstruction resolution of 0.3 nm in a noise-free condition. In a noisy environment with a signal-to-noise ratio as low as 30 dB, the reconstruction resolution of the interferograms processed by the CAED exhibits an enhancement from 1.2 to 0.4 nm, approaching the noise-free value. Our technology is envisaged to provide a powerful and cost-effective solution for applications requiring accurate, broadband, and energy-efficient spectral analysis.

Optical spectrometry is a highly effective analytical tool employed in both academic and industrial areas[1,2]. Its applications encompass material analysis, medical diagnostics, and environmental monitoring[3–5]. To cater to the demands of portable, handheld, and wearable applications, miniature spectrometers are rapidly advancing[6,7]. Chip-scale spectrometers based on silicon (Si) photonic integrated circuits (PICs) boast several advantages, including CMOS compatibility and high integration level, making them an appealing option for developing high-performance miniature

spectrometers[8,9]. Currently, most on-chip spectrometers utilize planar dispersive optics, narrowband filters, and Fourier transform (FT) interferometers[10–15]. Computational spectrometry has recently emerged as a new paradigm, utilizing computational methods to approximate or reconstruct the incident spectrum from pre-calibrated spectral response information[16]. Computational spectrometers usually comprise arrays of photonic structures such as photonic crystal slabs[17], photonic crystal nanobeam cavities[18], and stratified waveguide filters[7]. The photonic structure arrays and

[1]School of Microelectronics, Shanghai University, Shanghai, China. [2]Shanghai Collaborative Innovation Center of Intelligent Sensing Chip Technology, Shanghai University, Shanghai, China. [3]Institute of Precision Optical Engineering, School of Physics Science and Engineering, Tongji University, Shanghai, China. [4]MOE Key Laboratory of Advanced Micro-Structured Materials, Shanghai, China. [5]Shanghai Institute of Intelligent Science and Technology, Tongji University, Shanghai, China. [6]Shanghai Frontiers Science Center of Digital Optics, Shanghai, China. [7]Shanghai Industrial μTechnology Research Institute (SITRI), Shanghai, China. [8]Department of Electrical and Computer Engineering, National University of Singapore, Singapore, Singapore. [9]Center for Intelligent Sensors and MEMS (CISM), National University of Singapore, Singapore, Singapore. [10]National Centre for Advanced Integrated Photonics (NCAIP), Singapore, Singapore. [11]These authors contributed equally: Jing Zhou, Hui Zhang. ✉e-mail: yimingma@shu.edu.cn; elelc@nus.edu.sg

corresponding detector arrays significantly increase the complexity, footprint, and cost of the PICs. Over the past few years, several spectrometers have been developed that make use of solely a single tunable filter or interferometer paired with a single detector[19–21]. These devices offer a simpler, smaller, and more cost-effective alternative for computational spectrometry.

The tunability in Si PICs is typically realized by thermo-optic modulation and free carrier injection, both relying on the change of the Si refractive index[22,23]. However, because of the weak perturbation of the Si refractive index, these methods frequently result in high power consumption[24]. In comparison, microelectromechanical systems (MEMS) attain modulation by spatially displacing photonic components, consequently improving the modulation efficiency and reducing the power consumption[25,26]. Among a variety of MEMS actuation mechanisms, electrostatic actuation stands out due to its ultra-low standby power and reconfiguration energy consumption[27]. Therefore, reconfiguration using electrostatic MEMS actuation offers a simple, effective, and energy-efficient approach for the construction of on-chip spectrometers.

Nonetheless, the presence of noise detrimentally affects the quality of output generated by actuators used for converting information to physical, chemical, or biological effects[28]. MEMS actuators are particularly susceptible to noise issues due to the movable structures and the small sizes of their electronic, mechanical, and other components[28,29]. The spectral resolution of the spectrometer depends on both the reconstruction algorithm and the measurement noise[30]. In very noisy environments, conventional algorithms frequently produce significant distortion of the reconstructed spectrum[31]. Therefore, it is important to remove noise effects, especially for MEMS-enabled spectrometers. However, it is challenging due to intricate noise mechanisms. The application of deep learning technologies is nowadays considered as a potentially promising solution for this problem in spectrum reconstruction[32–34]. Autoencoder is a deep learning technology that can adaptively learn the structure of data and represent data efficiently[35,36]. Autoencoders have demonstrated markable benefits for molecular property prediction[37], image segmentation[38], and quantum systems[39]. Furthermore, they have been proven to be effective in reducing noise in single-cell RNA sequencing and ultrasonic signals[40,41]. Denoising autoencoders, because of their weak constraints from noise generation mechanisms, show potential for reducing MEMS noise[35].

In this paper, we present a paradigm of computational spectrometers based on the synergy between electrostatic MEMS modulation and convolutional autoencoder denoising (CAED) mechanism. The device features a waveguide coupler reconfigured by an integrated MEMS cantilever actuator. Through a strategic reduction of the MEMS tuning range by revealing its counterintuitive relationship with the reconstruction performance, the device yields high fabrication efficiency and optimum reconstruction resolution. On top of the ultra-low power consumption enabled by the electrostatic MEMS tuning, a CAED strategy is proposed and utilized to minimize the side effects of the associated MEMS noise on the reconstruction performance. The autoencoder is trained on a diverse dataset of chip-collected interferograms, achieving optimal noise reduction with a resolution approaching the noise-free level. Spectrum reconstruction results demonstrate the effectiveness of CAED in mitigating noise effects with a low signal-to-noise ratio (SNR) of 30 dB, resulting in the improvement of the resolution from 1.2 to 0.4 nm. The proposed CAED-facilitated MEMS spectrometer presents a promising solution for broadband high-resolution spectral analysis in applications demanding precision and power efficiency. The utilization of advanced deep learning techniques of denoising autoencoders not only improves the performance of MEMS spectrometers but also presents a universal solution for mitigating noise-related challenges in computational spectrometers with calibration matrices.

## Results
### Design and architecture
Our proposed denoising-autoencoder-facilitated MEMS computational spectrometer consists of a MEMS-enabled computational spectrometer (MECS) and a CAED mechanism. The concept of the computational spectrometer here is analogous to FT spectrometers and centers around the generation of interferograms[42]. The interferograms at the output port are functions of received signal intensity over time and are converted to a wavelength-dependent spectrum via computational algorithms. The MECS is designed as a cantilever-tunable waveguide coupler, consisting of a straight waveguide and a cantilever waveguide (Fig. 1a). Both waveguides in the coupling region are supported by a single-sided structure, while the straight waveguide outside the coupling region is supported by a two-sided structure, thus defining the movable and stationary parts. When applying a bias voltage $V$, the cantilever waveguide can be electrostatically pulled down while the straight waveguide remains immobile. A vertical coupling gap $h$ will be induced between the two waveguides, subsequently resulting in a change in the effective index difference between the symmetric mode (SM0) and the asymmetric mode (SM1) of the waveguide coupler. The effective index difference $\Delta n$ is a function of both the voltage $V$ and the wavelength $\lambda$, thus can be represented by $\Delta n(\lambda, V)$. According to the coupled-mode theory, the output power of the straight waveguide can be described as[43]:

$$P_o(\lambda, V) = A(\lambda)\cos^2\left(\frac{\pi L}{\lambda}\Delta n(\lambda, V)\right) \tag{1}$$

where $A(\lambda)$ is the spectrum of input light, $L$ is the coupling length. When we apply time-variant bias voltage and thus time-domain modulation of the vertical gap $h$, an interferogram $P_o(\lambda, V)$ will be generated at the output port for each wavelength, as depicted in Fig. 1b. Subsequently, we apply spectrum reconstruction algorithms to the interferogram data, and the reconstructed spectrum is shown in Fig. 1c. The existence of noise in the interferograms poses a challenge in reconstructing the spectrum with closely adjacent peaks, thereby limiting the reconstruction resolution.

To address this problem, we propose a CAED mechanism. The architecture of the CAED is illustrated in Fig. 1d, which consists of an encoder $\mathcal{E}$ for compression and a decoder $\mathcal{D}$ for reconstruction. The autoencoder enables denoising by learning a meaningful representation of input data through the compression and reconstruction process. The encoder $\mathcal{E}$ learns a compact representation of input interferogram data, filtering out the irrelevant information from the input, forcing the model to retain only the essential features for reconstruction, and then the decoder $\mathcal{D}$ reconstructs the clean input. The process can be expressed as:

$$P_d(\lambda, V) = (\mathcal{D} \circ \mathcal{E})P_o(\lambda, V) \tag{2}$$

where $\circ$ denotes the sequential application of functions. Convolutional autoencoders, utilizing convolutional layers, prove exceptional proficiency in capturing spatial structures and are particularly effective for denoising tasks related to images or spatiotemporal data[44,45]. The interferogram after denoising is shown in Fig. 1e, and the incident spectrum can be accurately reconstructed from it, even when the two peaks are closely adjacent (Fig. 1f). In other words, the spectrum reconstruction performed to $P_d$ improves the noise robustness of the MECS.

### MEMS spectrometer
We first work on the principle of the proposed MECS. As shown in Fig. 2a–c, the device is fabricated on a silicon-on-insulator (SOI) wafer that consists of a 0.22 μm thick silicon device layer and a 2 μm thick buried oxide (BOX) layer. Both the straight and cantilever waveguides

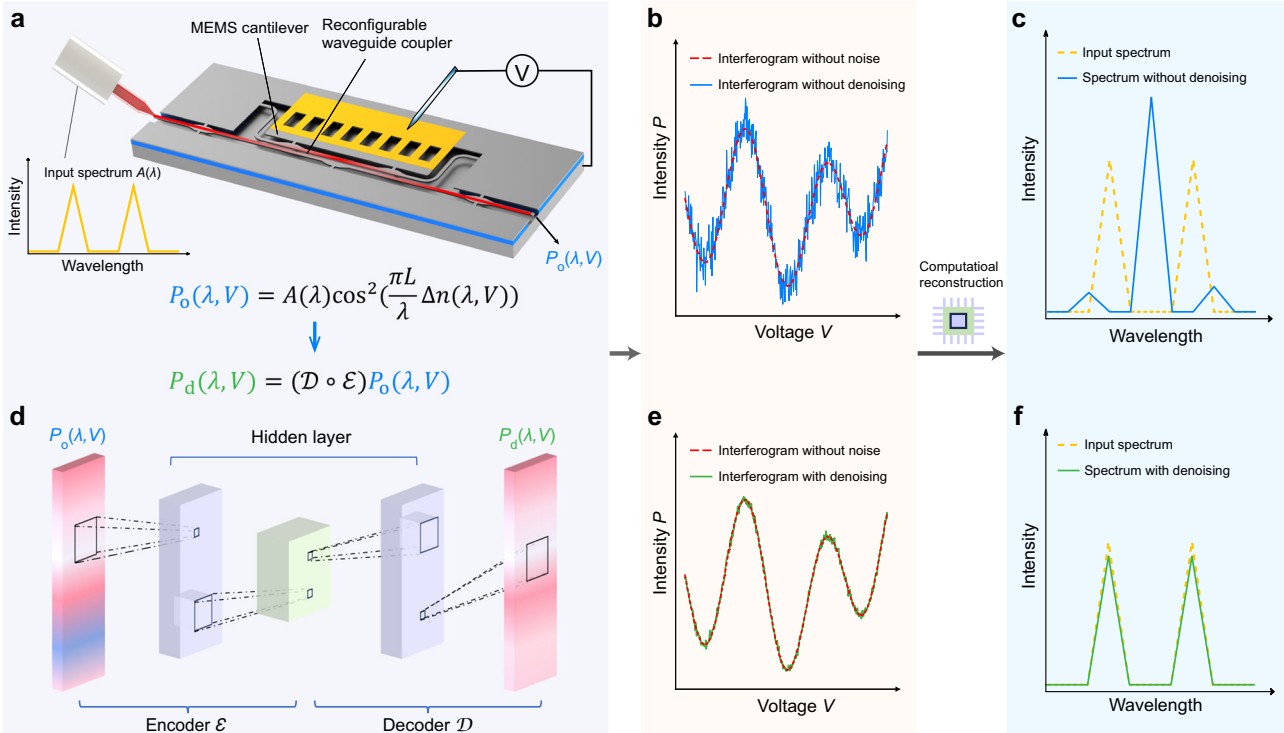

**Fig. 1 | Conceptual illustration of the spectrometer. a** Schematic of the MEMS-enabled computational spectrometer (MECS) featuring a waveguide coupler reconfigured by an integrated electrostatic MEMS cantilever actuator. A spectrum is coupled into the MECS chip for analysis. **b** Interferogram obtained at the MECS output port without denoising. **c** Spectrum reconstruction result without denoising. **d** Architecture of the convolutional autoencoder denoising (CAED) mechanism used for interferogram denoising. **e** Interferogram with CAED. **f** Spectrum reconstruction result with CAED.

are 0.35 μm wide for single transverse-electric (TE) mode propagation. The waveguide coupler is designed with an initial coupling gap of 200 nm and a coupling length of 2030 μm. When a bias voltage is applied between the cantilever and the silicon substrate, electrostatic attraction induces downward displacement of the cantilever waveguide, while the straight waveguide remains stationary due to insulation grooves, as shown in Fig. 2d.

As illustrated in Fig. 2e, the coupling between the cantilever and straight waveguides can be understood by the interference of two supermodes (SM0 and SM1) formed in the waveguide coupler. The vertical coupling gap $h$, in conjunction with the wavelength $\lambda$, defines an effective index difference $\Delta n(\lambda, h)$ between SM0 and SM1, i.e., $\Delta n(\lambda, h) = n_1(\lambda, h) - n_2(\lambda, h)$. Specifically, $\Delta n(\lambda, h)$ can be approximated by a polynomial function:

$$\Delta n(\lambda, h) \approx \left(a_1 + a_2\lambda + a_3\lambda^2\right)\left(b_1 + b_2 h + b_3 h^2 + b_4 h^3\right) = f_1(\lambda) \cdot f_2(h) \quad (3)$$

This approximation is validated using numerical calculations (see Supplementary Note 1). The polynomial approximation can be fitted with a 99.72% R-squared value. $h$ is a function of the applied bias voltage $V$ and can be approximated as:

$$h(V) \approx c_1 + c_2 V + c_3 V^2 + c_4 V^3 \quad (4)$$

We also validate this approximation using numerical calculations (see Supplementary Note 2) and achieve a good fitting with a 99.95% R-squared value. By combining Eqs. (1), (3), and (4), the output power of the proposed spectrometer can be given as:

$$P_o(\lambda, V) = A(\lambda)\cos^2\left(\frac{\pi L f_1(\lambda) \cdot f_2(h(V))}{\lambda}\right) \quad (5)$$

For our designed waveguide coupler with a certain coupling length $L$, an interferogram can be obtained at the output port by applying a time-variant bias voltage, using a light beam with a wavelength of $\lambda$.

We investigate the relationship between the device tuning range and the spectral reconstruction performance using correlation analysis (see Supplementary Note 3 and Fig. S3). We find that the fully decoupled condition beyond a certain range leads to a larger self-correlation width, indicating impaired reconstruction resolution. Therefore, we adopt a moderately decoupled condition, which not only guarantees satisfactory spectral resolution but also reduces the required tuning range to a level achievable by the release of the BOX layer[46]. This approach significantly simplifies the device configuration and fabrication process. Other attempts, such as employing a trapezoidal supporting structure instead of subwavelength grating for the suspended waveguides, possess lower fabrication restrictions and improved wavelength scalability (see Fig. S4). Using a straight waveguide as the bus waveguide, instead of the traditional directional coupler, reduces propagation loss and improves the SNR. Additionally, edge couplers are utilized to enlarge the device bandwidth (see Fig. S5). The static transmission spectrum and the frequency response are provided in Supplementary Note 4. All these attempts contribute to an enhanced, easy-to-fabricate, and large-bandwidth MEMS spectrometer.

## Spectrum reconstruction
Based on the interferograms obtained from the MECS, the spectrum can be reconstructed as follows. We first collect a matrix **Y** that indicates the spectral response of the device at each wavelength and each bias voltage. Interferograms for wavelengths from 1.3 to 1.6 μm (at a step of 0.1 nm) are acquired by applying a sequential bias voltage ranging from 0 to 29.9 V. The MEMS cantilever, measured 33 μm in length, possesses an estimated electrostatic pull-in voltage

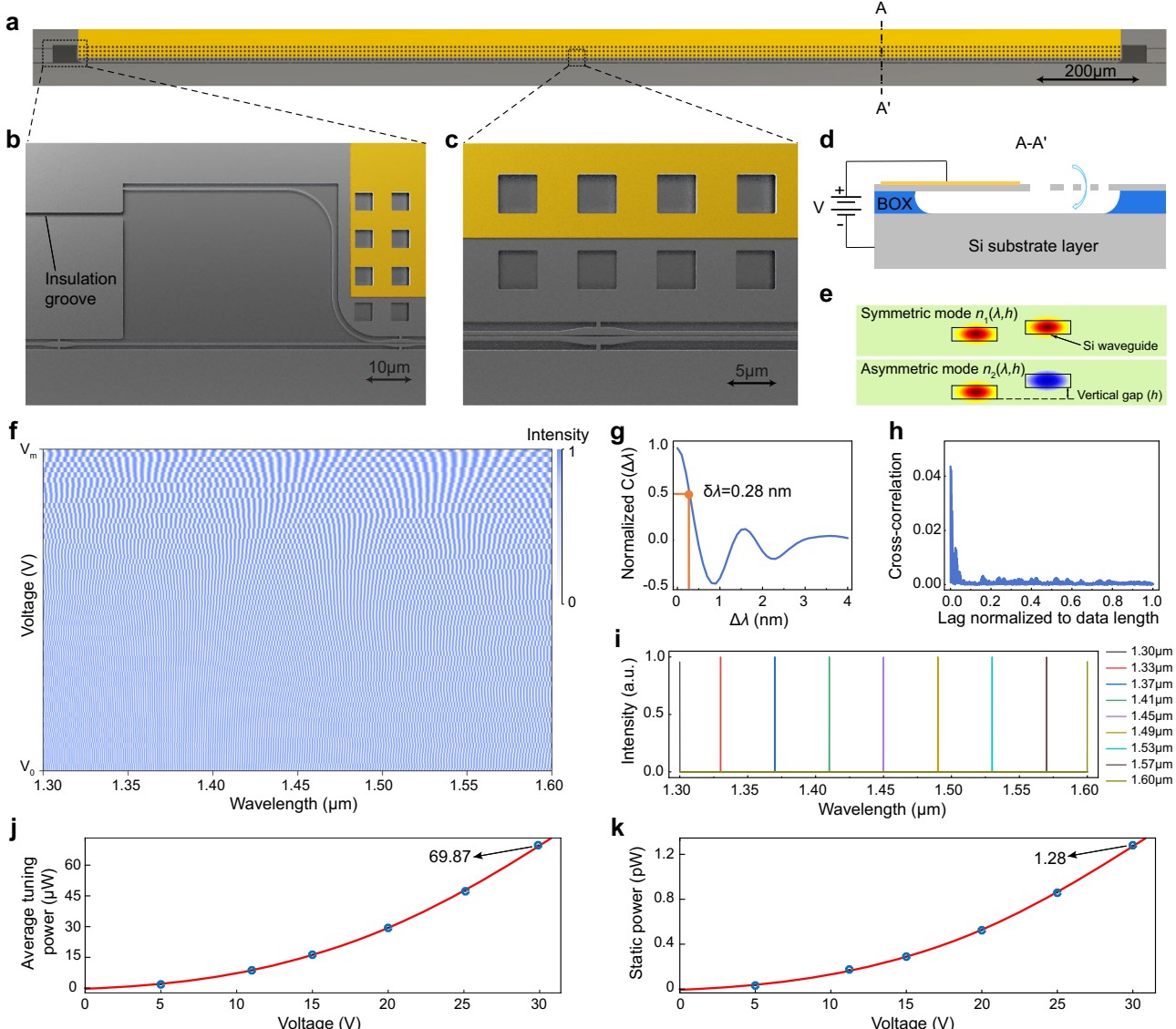

**Fig. 2 | MECS and spectrum reconstruction. a** Overall structure of the MECS on a silicon-on-insulator (SOI) wafer. **b** Front end of the waveguide coupler. **c** Waveguide coupling region. **d** Downward displacement of the MEMS cantilever and the integrated cantilever waveguide under applied bias voltage. **e** Mode profiles of the waveguide coupler under displacement. **f** Calibration matrix **P** of the MECS. The wavelength range is swept in a 0.1 nm high resolution, and 64 steps of DC bias voltage are gradually applied to the MEMS actuator. **g** Calculated spectral self-correlation function C(Δλ) with a self-correlation width δλ of 0.28 nm. **h** Absolute value of the averaged cross-correlation between one specific channel and all the other channels. **i** Several reconstructed single-wavelength spectra over the 300 nm bandwidth. **j** Average tuning power required to reach the corresponding applied voltages. **k** Static power required to hold at the corresponding voltages.

of 34.3 V (see Supplementary Note 2). Due to the small displacement of the cantilever at low voltage levels, which results in a limited optical response, a higher voltage increment is chosen at lower bias voltages, with 64 steps in total. Thus, the $m$-by-$n$ matrix **Y** now has dimensions $m = 64$, $n = 3001$, where $m$ is the number of bias voltages, and $n$ is the number of wavelengths. As the laser intensity, edge coupler efficiency, and detector responsivity vary at different wavelengths, the 3001 elements in each row of the matrix **Y** need to be normalized to cancel out these wavelength-dependent testing system features. The normalization vector $\mathbf{W} = [w_1, w_2, \cdots, w_n]^T$ is the transmission spectrum of a reference straight waveguide on the same chip and with the same design as that of the MECS. Therefore, the calibration matrix **P** of our spectrometer can be given as:

$$\mathbf{P} = \mathbf{Y} \cdot \mathrm{diag}\left(w_1^{-1}, w_2^{-1}, \cdots, w_n^{-1}\right) \qquad (6)$$

where diag represents the diagonal matrix form. Here, each column of **P** (as shown in Fig. 2f) represents the interferogram of the corresponding wavelength.

The performance of a spectrometer is expected to achieve two properties: (i) The spectral response at each sampling channel has diverse features, so that the correlation length in the wavelength span can be small to provide high spectral resolution; (ii) The transmission spectra for any two sampling channels should be very different, i.e., orthogonal, to provide a transmission sampling matrix with a large rank[7]. Spectral self- and cross-correlations are calculated from the calibrated matrix **P** and shown in Fig. 2g, h (see more details in Supplementary Note 5). The self-correlation width, δλ, is read as 0.28 nm, providing an estimation of the spectral resolution. The low cross-correlation approaching almost 0 indicates that the spectra of these sampling channels contain very diverse features, which proves the effectiveness of our designed time-domain modulation channels.

Any output interferogram $\mathbf{I}$, corresponding to a polychromatic signal represented by a column vector $\mathbf{R}$, can be expressed as:

$$\mathbf{I} = \mathbf{P} \cdot \mathbf{R} \qquad (7)$$

Thereby, the incident spectrum $\mathbf{R}$ can be determined from the interferogram $\mathbf{I}$ by solving the regularized regression problem, which involves using inadequate constraints (i.e., the $\mathbf{I}$ with a size of 64) to infer the $\mathbf{R}$ in a size of 3001. To specify a unique solution, the under-constrained system can be solved by using the equation below:

$$\min_{\mathbf{R}}\{\|\mathbf{I} - \mathbf{P} \cdot (\mathbf{R}_1 + \mathbf{R}_2)\|_2^2 + \alpha\|\mathbf{R}_1\|_1 + \beta\|\mathbf{R}_2\|_2^2\} \qquad (8)$$

where $\mathbf{R}_1$ and $\mathbf{R}_2$ denote the discrete and continuous component of $\mathbf{R}$, respectively. $\alpha$ and $\beta$ denote the regularization parameters that embody the intrinsic characteristics of the spectrometer. The optimal values of $\alpha$ and $\beta$ are determined via cross-validation analysis[20]. Using Eq. (8), it is feasible to reconstruct a spectrum of arbitrary shape without specific knowledge of spectral contents (see more discussions on the reconstruction method in Supplementary Note 6). Our model can accurately reconstruct an input single-wavelength spectrum with an accuracy of $\pm 0.1$ nm over the entire working wavelength range (300 nm bandwidth). Figure 2i presents the reconstruction of several single-wavelength spectra across the whole bandwidth.

We characterize the device tuning energies at different applied voltages using the method described in Ref. 27. The average tuning power is derived as the tuning energy divided by the response time and plotted in Fig. 2j. Even at the maximum applied voltage of 29.9 V, the average tuning power is less than 70 μW. Additionally, the capacitor nature of the electrostatic MEMS actuator allows nearly zero standby power consumption, as measured and shown in Fig. 2k.

## Noise-free reconstruction and noise analysis

Before performing the denoising of MECS, we first determine the noise-free reconstruction resolution for dual-wavelength spectra as a reference. With noise present, an interferogram $\mathbf{I}$ consists of two components, i.e., the actual interferogram $\mathbf{I}_a$ and the noise interference $\mathbf{e}$:

$$\mathbf{I} = \mathbf{I}_a + \mathbf{e} \qquad (9)$$

As a result of the stochastic nature of noise, the interferogram of an identical light beam varies with each passage through the same waveguide coupler. Compared to the interferogram $\mathbf{I}_1 = \mathbf{I}_a + \mathbf{e}_1$ recorded during the calibration, the same interferogram recorded during the spectrum reconstruction can be expressed as $\mathbf{I}_2 = \mathbf{I}_a + \mathbf{e}_2 = \mathbf{I}_1 - \mathbf{e}_1 + \mathbf{e}_2$. Since the absolute value of noise is hard to be quantitated, we take the interferogram recorded during the calibration as a reference. The noise in the interferogram during the subsequent spectrum reconstruction is considered as the relative value ($\Delta\mathbf{e} = \mathbf{e}_2 - \mathbf{e}_1$) with respect to this reference. The interferogram recorded during the calibration is referred to as noise-free ($\Delta\mathbf{e} = 0$) in the following content.

As verified by our previous work, a dual-wavelength interferogram can be considered as the linear superposition of two single-wavelength ones[20]. Therefore, the noise-free dual-wavelength interferogram can be obtained by weighted summation of any two column vectors (i.e., single-wavelength interferograms with a wavelength interval of $\Delta\lambda$) in the calibration matrix $\mathbf{P}$, where the weights account for the realistic nonideality of different amplitudes of two laser sources. The spectrum reconstruction results for the synthesized dual-wavelength interferograms are shown in Fig. 3a. Figure 3b, c provide a zoom-in view of the reconstructed spectrum when $\Delta\lambda$ is 0.2 and 0.3 nm, respectively. The reconstruction resolution is defined by the minimum resolvable spacing between the two wavelengths, observed when the

reconstructed spectrum closely matches the input spectrum. In order to quantify the reconstruction accuracy, here we utilize the widely adopted metric named relative error $\varepsilon$, which is defined as[19]:

$$\varepsilon = \frac{\|\mathbf{R} - \widehat{\mathbf{R}}\|_2}{\|\mathbf{R}\|_2} \qquad (10)$$

where $\mathbf{R}$ and $\widehat{\mathbf{R}}$ are the input and reconstructed spectrum, respectively. The calculated $\varepsilon$ in Fig. 3b, c indicates that distinguishing between two peaks with a separation of less than 0.2 nm is challenging, and the noise-free reconstruction resolution of the spectrometer is ~0.3 nm.

However, in practical scenarios, the measured interferograms are inevitably influenced by noise, thus worsening the reconstruction resolution. In fact, due to their tiny and movable components, sources of noise in MEMS devices are quite diverse, including thermal noise, shot noise, $1/f$ noise, and others[28]. In Supplementary Note 7, we analyse the significant effects of thermal and shot noises on cantilever displacement. Figure 3d illustrates the noise-induced floating displacements of the cantilever waveguide, which impose noise on the measured interferograms. After the calibration matrix collection, the interferograms are measured by the MECS again for 1000 wavelengths within the bandwidth. Then, we calculate their relative SNR, which is defined as $10\lg\frac{\|\mathbf{I}_1\|^2}{\|\mathbf{I}_2 - \mathbf{I}_1\|^2}$. The resulting SNR amplitude distribution ranges from 30 to 55 dB, as shown in Fig. 3e. To investigate the impact of noise on the reconstruction resolution, we apply white noise with the highest noise level corresponding to 30 dB SNR to 20 randomly selected dual-wavelength interferograms. The spectrum reconstruction resolution falls in the range of 0.8–1.2 nm, which is significantly worse than the noise-free value of 0.3 nm, as shown in Fig. 3f. According to the Rayleigh criterion (refer to Supplementary Note 8), the length of the waveguide coupler needs to be extended 4 times to achieve the same 0.3 nm resolution. Therefore, it is imperative to mitigate noise effects.

To minimize the impact of noise on reconstruction resolution, an intuitive strategy is to limit the noise during device design, which, however, needs to be delicate while having limited effects. A noise reduction algorithm for interferograms would be a more effective and cost-efficient solution. Nevertheless, the interferograms formed by scatter plots are irregular and lack discernible frequencies, making traditional denoising techniques deficient[47]. Considering that all column vectors in the calibration matrix $\mathbf{P}$ are linearly independent eigenvectors, any dual-wavelength interferogram can be regarded as a linear combination of two single-wavelength interferograms. This represents a form of feature space combination that is well-suited for autoencoder purification[48]. Hereby, we propose a CAED mechanism to acquire a compact representation of input data and eliminate irrelevant information from the input.

## Convolutional autoencoder denoising

Denoising autoencoder aims to learn a representation robust to noise added to the original data. Typically, training a denoising autoencoder aims to reconstruct the original data with minimal error. However, if the original data is complicated, the training process may be time-consuming and may lead to underfitting. Additionally, if the autoencoder is overly specialized for a certain type of input, it may lose generalizability to other patterns, necessitating different models for different input spectra. Hereby, we employ a different, noise-oriented training strategy: instead of training the autoencoder to recover the input pattern, we recover the noise pattern and then subtract it from the initial input data (see details in Supplementary Note 9)[49]. To be specific, consider a noisy observation $\mathbf{I}$, which consists of the original data $\mathbf{I}_a$ and the noise $\mathbf{e}$, i.e., $\mathbf{I} = \mathbf{I}_a + \mathbf{e}$. Since $\mathbf{e}$ is simpler and has a more consistent pattern, we train the autoencoder by learning $\mathbf{e}$ and subtracting it from $\mathbf{I}$, which is more effective than learning $\mathbf{I}_a$ directly. The schematics of the training and testing phases are depicted in Fig. 4a.

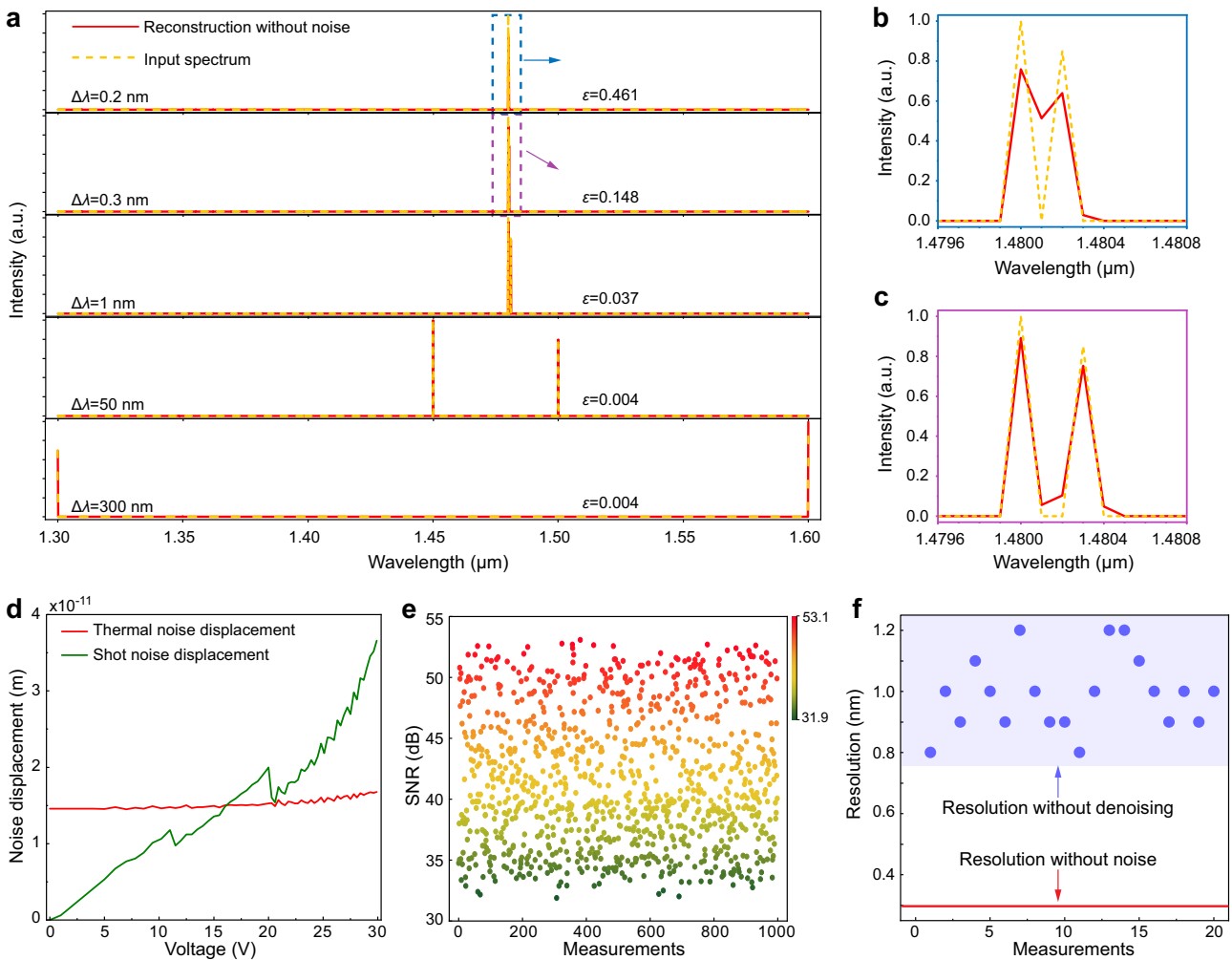

**Fig. 3 | Noise-free spectrum reconstruction and noise analysis. a** Reconstruction of dual-wavelength spectra with different wavelength spacings under the noise-free condition. **b** Zoom-in view of the input and the reconstructed spectra with 0.2 nm spacing. **c** Zoom-in view of the input and the reconstructed spectra with 0.3 nm spacing. **d** Thermal noise displacement and shot noise displacement as functions of the applied bias voltage. **e** Distribution of SNR magnitudes. **f** Distribution of the resolution for spectrum reconstruction under an SNR of 30 dB.

The parameters of the autoencoder (i.e., encoder $f_\theta$ and decoder $g_{\theta'}$) are optimized as follows:

$$\theta^*, \theta'^* = \arg\min_{\theta, \theta'} \frac{1}{M} \sum_{i=1}^{M} \mathcal{L}\left(\mathbf{e}^{(i)}, g_{\theta'}\left(f_\theta\left(\mathbf{I}^{(i)}\right)\right)\right) \qquad (11)$$

where $\mathcal{L}$ is a loss function of mean squared error (MSE) between two inputs. During training phase, the $\mathbf{e}^{(i)}$ is derived by subtracting the ground truth $\mathbf{I}_a^{(i)}$ from the input sample $\mathbf{I}^{(i)}$. In test phase, we employ the trained autoencoder to predict $\widetilde{\mathbf{e}}^{(j)}$ and subtract it from the input sample to derive the regenerated data $\widetilde{\mathbf{I}}_a^{(j)}$, which can be represented as follows for all $j \in \{1, \ldots, L\}$:

$$\widetilde{\mathbf{I}}_a^{(j)} = \mathbf{I}^{(j)} - g_{\theta'^*}\left(f_{\theta^*}\left(\mathbf{I}^{(j)}\right)\right) \qquad (12)$$

While it is not necessary to include all possible patterns that will appear in the testing phase in the training set, maintaining diversity is crucial to ensure that the autoencoder learns the noise pattern rather than the pattern of any specific input type. Therefore, we construct a dataset comprising various input interferogram patterns, by sampling from the calibration matrix **P** (Fig. 4b). Taking dual-wavelength

interferogram as an example, to ensure that each column feature of matrix **P** has the same probability of being sampled in the synthesized dataset, we reshape the transpose of **P** (**P′** size of 3001 × 64) by concatenating its first 100 rows to its end, forming a new matrix **Q** with a size of 3101 × 64. Index pairs $(i, j)$ are randomly generated, where $0 < j - i \leq 100$ and $1 \leq i \leq 3001$, and the $i$th row of matrix **Q** is added with the $j$th row of **Q** to form dual-wavelength interferograms. Similarly, index triplets $(i, j, k)$ and index quadruplets $(i, j, k, l)$ are randomly generated to construct triple-wavelength and quadruple-wavelength data. The three types of data, each with 10,000 samples, together construct the mixed dataset (matrix **M**) of 30,000 samples. Gaussian white noises corresponding to SNRs of 30 and 36 dB are then added to each row of matrix **M**, forming an interferogram dataset **N** consisting of 60,000 noisy interferograms. Noises of different levels are added to enhance the diversity of the dataset, so as to improve the generalizability of the trained model[50].

The architecture of the convolutional autoencoder is depicted in Fig. 4c, which consists of an encoder for compression and a decoder for reconstruction. The encoder comprises convolutional layers, maximum pooling layers, and residual blocks, while the decoder includes transpose convolution layers and convolutional layers. The performance of CAED is optimized by employing the residual block and fine-tuning the convolution kernel size and the number of

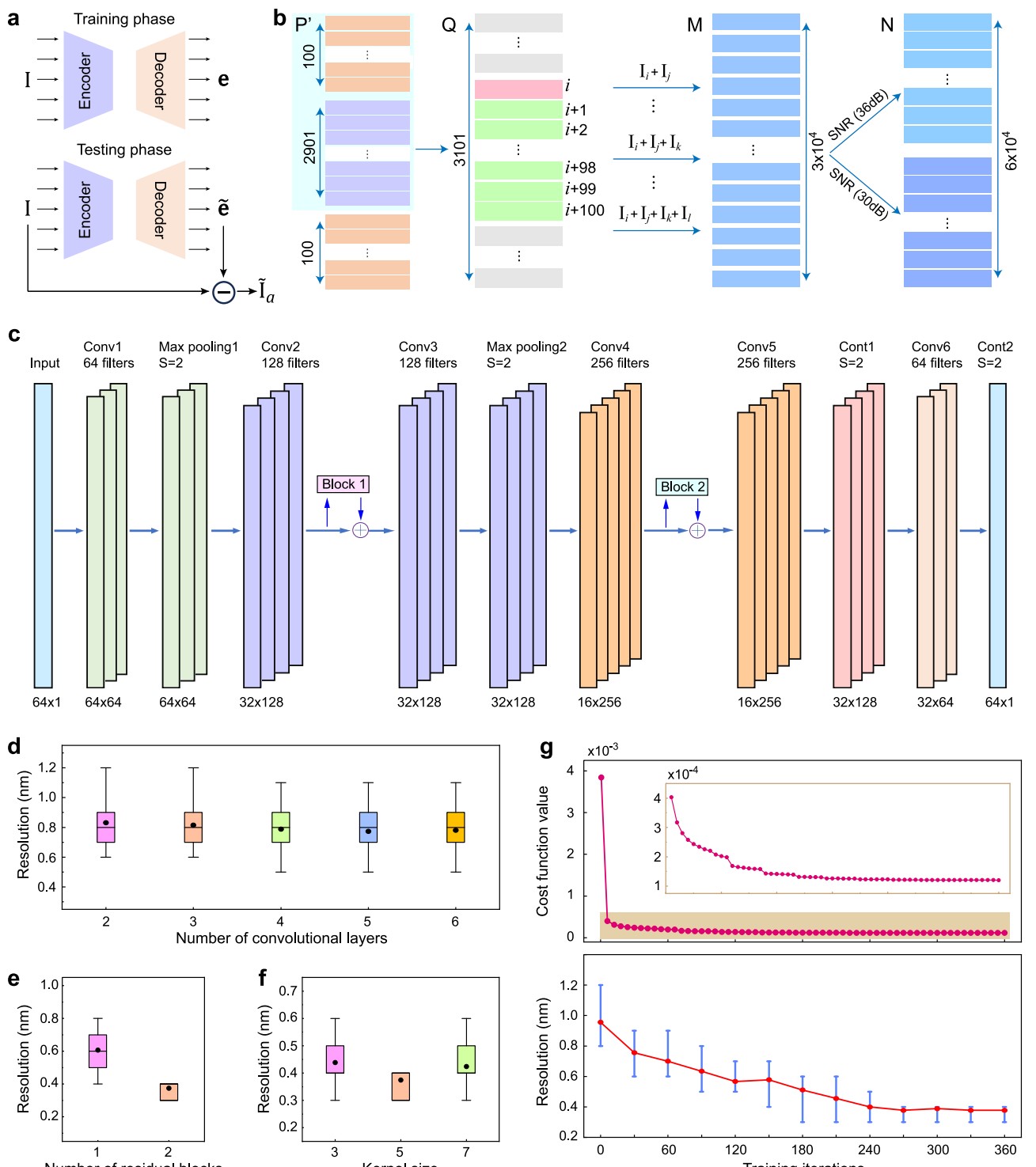

**Fig. 4 | Construction of convolutional denoising autoencoder. a** Noise-oriented training scheme for the denoising autoencoder. **b** Make the noise dataset. **c** Architecture of the convolutional autoencoder. **d**–**f** Optimization of convolutional neural network (CNN) structural parameters based on reconstruction resolution, including the number of convolutional layers, the number of residual blocks, and the kernel size. Boxplots indicate median (middle line), 25th, 75th percentile (box), and 5th, 95th percentile (whiskers). **g** Evolution of the mean squared error (MSE) and the resolution in each generation, with increasing generations. The top and bottom bars represent the maximum and minimum values of resolution, respectively.

convolutional layers. According to Fig. 4d–f, the CAED model showing the optimal noise reduction performance contains 5 convolution kernels, 2 residual blocks, and 6 convolutional layers. During the training process, MSE is used as the loss function, and the resolution of spectrum reconstruction is evaluated on the test set (Fig. 4g). The MSE decreases rapidly during training, while the resolution gradually

improves and eventually stabilizes. The trained autoencoder demonstrates great generalizability in denoising a variety of input interferogram patterns without the need for retraining for each specific kind of pattern (see details in Supplementary Note 9).

Figure 5a–c shows the non-denoised, denoised, and noise-free reconstruction resolutions of 20 sets of dual-wavelength

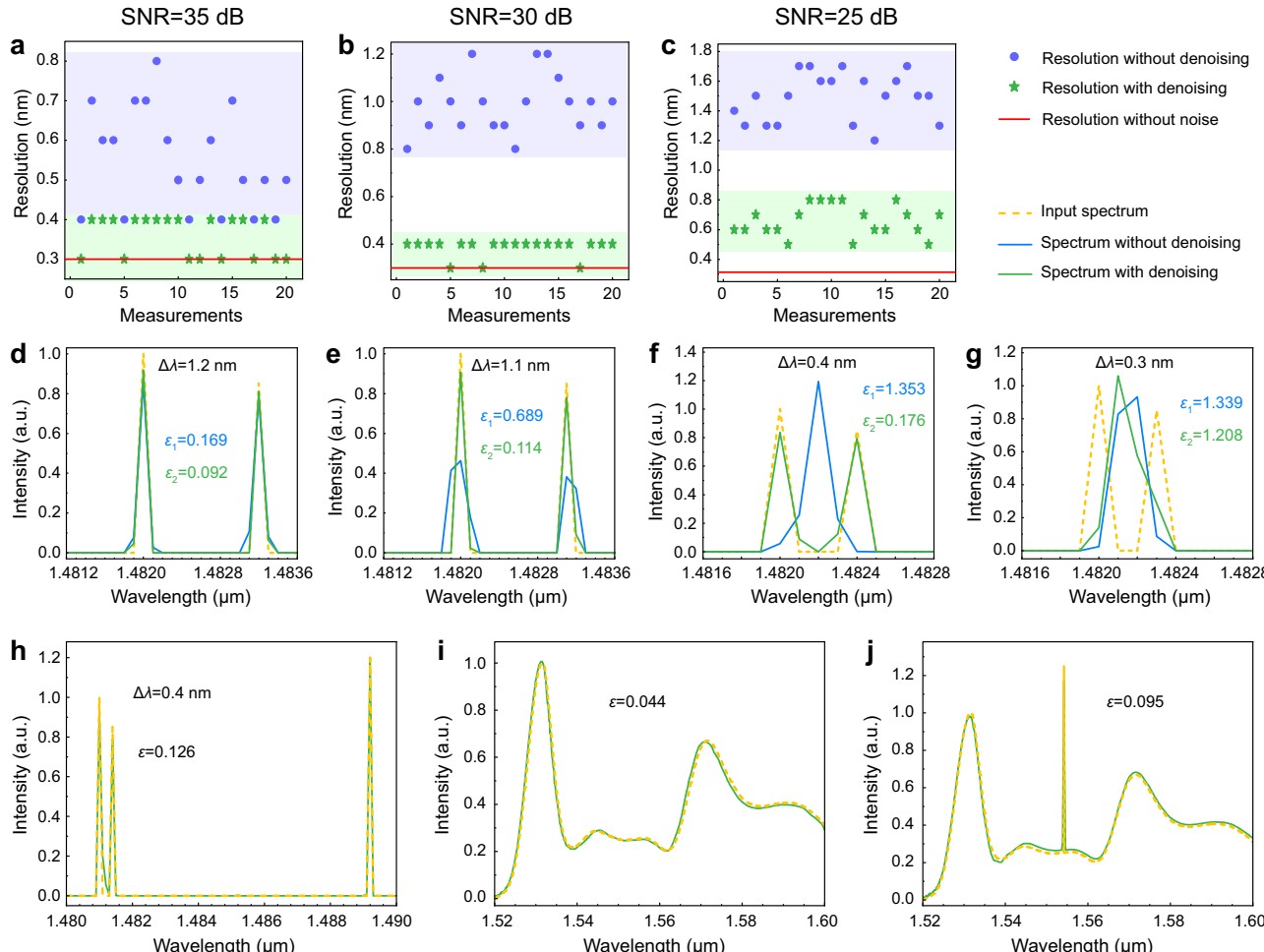

**Fig. 5 | Test of CAED effectiveness for spectrum reconstruction.** Distribution of reconstruction resolution with or without denoising under different SNR levels of (**a**) 35 dB, (**b**) 30 dB, and (**c**) 25 dB. Reconstruction of dual-wavelength spectrum with or without denoising at different wavelength spacings of (**d**) 1.2 nm, (**e**) 1.1 nm, (**f**) 0.4 nm, and (**g**) 0.3 nm. **h**–**j** Reconstruction of a diverse range of incident spectra: (**h**) triple-wavelength spectrum, (**i**) broadband spectrum, and (**j**) mixed broadband/narrowband spectrum. All the reconstructions are performed throughout 3001 wavelength channels. For clarity, all the relative errors $\varepsilon$ are calculated within the displayed wavelength ranges.

interferograms under different SNR conditions of 35, 30, and 25 dB, respectively. The results show that CAED improves the reconstruction resolution in all three noise scenarios (from 0.4–0.8 to 0.3–0.4 nm for 35 dB SNR, from 0.8–1.2 to 0.3–0.4 nm for 30 dB SNR, and from 1.2–1.7 to 0.5–0.8 nm for 25 dB SNR). The resolution can be improved to nearly the noise-free value for SNR of 30 dB and above. The denoising performance of CAED at lower SNR levels of 20, 15, and 8 dB is presented in Supplementary Note 10, where resolution improvement is also observed. For these lower SNR levels, a corresponding noise training dataset may achieve better denoising results. Therefore, our CAED mechanism can effectively work across the entire SNR range in real-life applications.

Using the trained model, we first assess the effectiveness of CAED on dual-wavelength interferograms by combining two tunable laser sources via a 50/50 optical coupler. Figure 5d–g depict the reconstruction results of the dual-wavelength interferograms with different wavelength spacings. At a wavelength spacing of 1.2 nm (Fig. 5d) or greater, the input spectrum can be accurately reconstructed regardless of whether the interferogram is denoised or not. Wavelength spacings between 0.4 and 1.1 nm (Fig. 5e, f) are where reconstruction without denoising is not feasible, whereas denoising the interferogram enables the spectrum reconstruction. At a wavelength spacing of 0.3 nm or less (Fig. 5g), the incident spectrum cannot be reconstructed even when the interferogram is denoised. Therefore, the

reconstruction resolution is successfully improved from 1.2 to 0.4 nm by CAED, almost approaching the noise-free value of 0.3 nm. In the synthesized dataset used to train the model, the maximum wavelength spacing of dual-wavelength spectra is 10 nm. To verify the effectiveness of CAED when the spacing between two wavelengths exceeds 10 nm, we also studied three scenarios with wavelength spacings of 20, 30, and 50 nm in Supplementary Note 11. Consistent reconstruction performance is observed, further illustrating that the denoising is independent of the input spectrum pattern. Our spectrometer also demonstrates robustness to temperature fluctuation of ±8 °C, which can be further extended to 10-70 °C as long as the calibration matrix at each temperature is pre-recorded (see Supplementary Note 12), covering the reasonable operating temperature range for practical applications. In Supplementary Note 13, we further analyse the tolerance of our spectrometer to fabrication errors.

Beyond the dual-wavelength experiment, a more challenging triple-wavelength testing is conducted. The result presented in Fig. 5h illustrates the successful reconstruction of three laser peaks and a spectral spacing of 0.4 nm between the two nearest peaks with a relative error $\varepsilon$ of 0.126. In addition, the reconstruction of a broadband spectrum is demonstrated using an amplified spontaneous emission (ASE) source as the input. As shown in Fig. 5i, the spectral features are well recovered with a low relative error $\varepsilon$ of 0.044. Furthermore, a mixed spectrum is examined, which combines a broadband signal (the

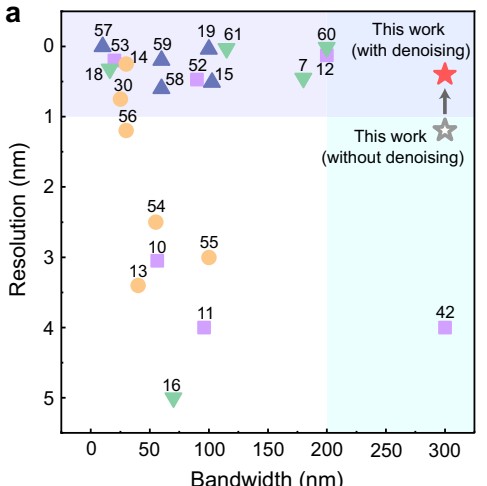
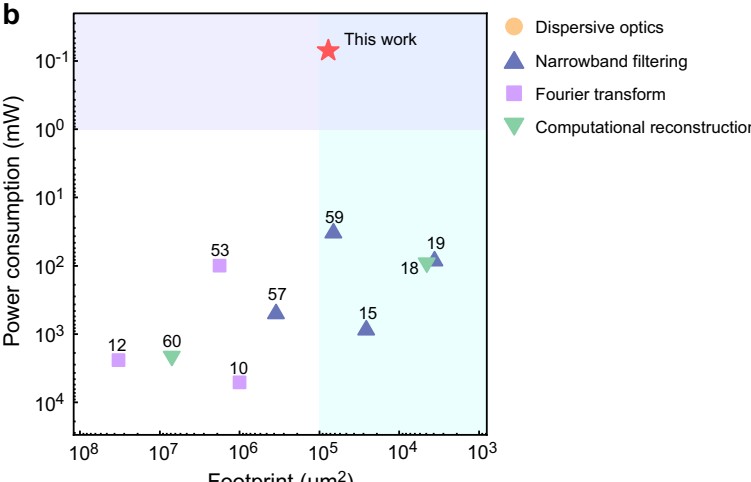

**Fig. 6 | Comparison of reported on-chip spectrometers.** The performance is compared in terms of (**a**) reconstruction resolution and bandwidth, (**b**) device power consumption and footprint. The experimental performance of our device with denoising is indicated by the solid red star, whereas the gray hollow star represents the performance of our device without denoising.

ASE source) with a narrowband signal (a laser source) via a 50/50 optical coupler. Figure 5j presents the resolved mixed spectrum with an $\varepsilon$ of 0.095, showing that a high reconstruction accuracy can still be attained. In Supplementary Note 14, we perform the reconstruction of a more broadband spectrum by simulation, providing additional evidence of the 300 nm bandwidth. Our spectrometer can find numerous real-life applications, for example, spectroscopic sensing of various molecules, such as N-methylaniline with a well-defined absorption fingerprint near 1.5 μm[51].

## Discussion

To benchmark our spectrometer, we conduct a comprehensive comparison with on-chip spectrometers that have been previously reported (see details in Supplementary Note 15)[7,10–16,18,19,30,42,52–61]. In most reported spectrometers, a distinct trade-off exists between resolution and bandwidth, as shown in Fig. 6a. Specifically, when the resolution surpasses 1 nm, the bandwidth tends to narrow down to less than 200 nm. Our proposed MEMS spectrometer demonstrates state-of-the-art performance in terms of bandwidth, and further, with the assistance of a denoising autoencoder, breaks through the trade-off limitation between bandwidth and reconstruction resolution, achieving a bandwidth of 300 nm and a reconstruction resolution of 0.4 nm. In addition to improving reconstruction resolution and robustness, the autoencoder denoising, executed using the relative error between the calibration matrix and measurements, has the potential to enhance the tolerance of spectrometers to manufacturing imperfections in high-volume production. Some recent demonstrations, leveraging narrowband filtering and computational reconstruction, have improved the bandwidth-to-resolution ratios to several thousands, but often come with the drawback of requiring extended sampling times. Additionally, they commonly employed thermal tuning, necessitating meticulous temperature control and high power consumption of over 30 mW. In comparison, our device features ultra-low power consumption of less than 70 μW thanks to the electrostatic MEMS reconfiguration, which is three orders of magnitude lower, as illustrated in Fig. 6b.

Recently, physically multi-stage structures have become popular practices for designing high-performance spectrometers by creating abundant sampling channels[60,61]. As a pioneer in MEMS spectrometers, our device, although limited to the simplest case, i.e., a single physical stage, can see significant performance improvements by further leveraging a multi-stage structure as illustrated in Fig. 7a. In this simulation demonstration, we implement a 3-stage design with 8

voltage states per stage (i.e., 512 sampling channels in total). The corresponding calibration matrix is shown in Fig. 7b. Thanks to the improved channel decorrelation and increased channel number, the reconstruction resolution is improved by one order of magnitude (see Supplementary Note 16 and Fig. S19). Meanwhile, despite the estimated noise cumulated to 19 dB SNR in the 3-stage structure, our current denoising autoencoder still achieves over twofold improvement of resolution to 40 pm, as shown in Fig. 7c–f, which could be further enhanced by accordingly optimizing the design of the autoencoder network. The multi-stage structure and the CAED mechanism also work well in the reconstruction of triple-wavelength, broadband, and mixed broadband/narrowband spectra (see Fig. S20). Regarding footprint, our device is significantly smaller than those physically multi-stage spectrometers, while less compact compared to some of the narrowband-filter-based spectrometers. Nonetheless, narrowband filters usually suffer from an inherent compromise between the SNR and spectral resolution[62]. To further shrink the footprint, we can reduce the coupling gap of the waveguide coupler by replacing the cantilever actuator with a comb-drive actuator, changing the current out-of-plane reconfiguration scheme to an in-plane one (see Supplementary Note 17). The driving voltage can also be reduced through optimizing the structural parameters of the MEMS actuator (see Supplementary Note 18).

In conclusion, our study represents a significant advancement in the domain of on-chip optical spectrometry, specifically focusing on MEMS-enabled devices. Our work highlights the inherent limitations of existing on-chip spectrometers, emphasizing the advantages offered by Si PICs and the efficacy of electrostatic MEMS modulation. Recognizing the susceptibility of MEMS actuators to noise, we develop CAED - a deep learning technology - to effectively mitigate noise effects and elevate spectrum reconstruction resolution. The results underscore the tangible potential of the proposed CAED-facilitated MEMS spectrometer. With the noise reduction capability at 30 dB SNR, the reconstruction resolution of the spectrometer is improved from 1.2 to 0.4 nm, approaching the noise-free value of 0.3 nm. Our approach lays a solid foundation for broadband high-resolution spectral analysis, particularly in applications demanding precision, power efficiency, and noise resilience. Moreover, it is worth highlighting that the presented CAED mechanism would have broad applicability in computational spectrometers using calibration matrices, due to its weak restrictions from noise generation mechanisms. Beyond its immediate implications, the denoising autoencoders are ready to provide a

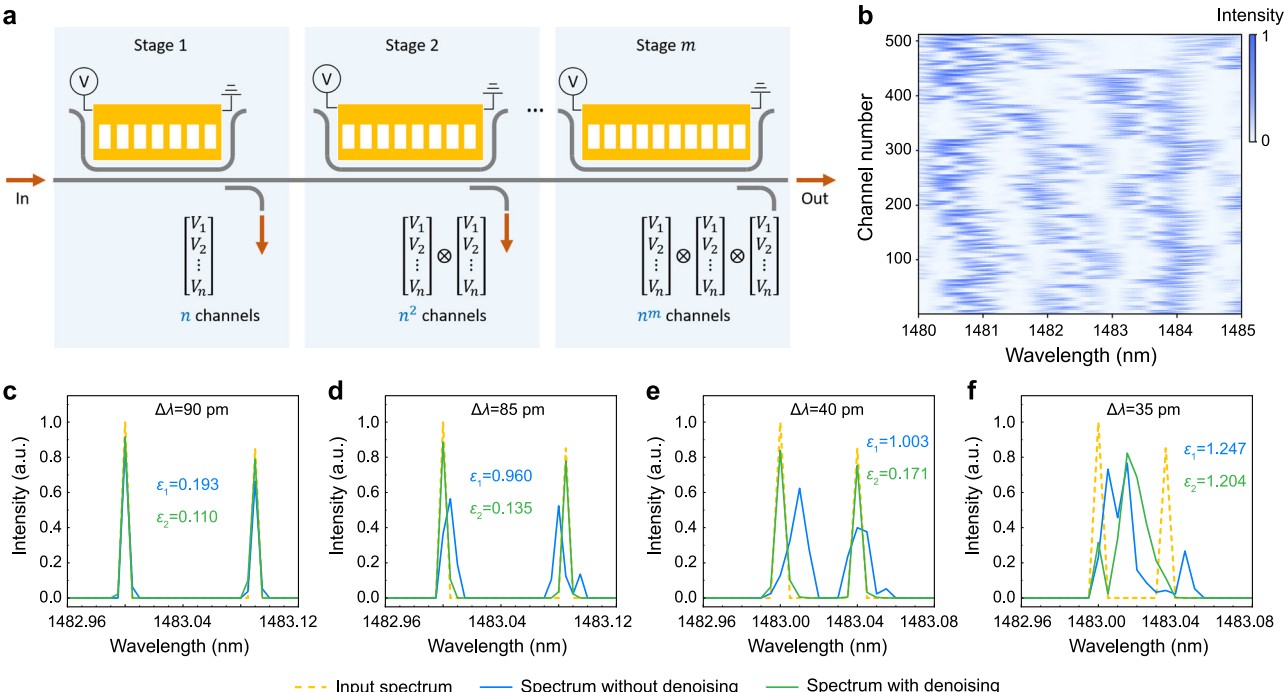

**Fig. 7 | Investigation of multi-stage design. a** Schematic illustration of the multi-stage MEMS spectrometer and voltage modulation channels. **b** Calibration matrix of the 3-stage spectrometer. Reconstruction of dual-wavelength spectrum with or without denoising at different wavelength spacings of (**c**) 90 pm, (**d**) 85 pm, (**e**) 40 pm, and (**f**) 35 pm.

strategic solution with far-reaching impacts on the ongoing evolution of miniaturized optical devices[36,39,63,64].

## Methods

### Device fabrication

The MECS is fabricated on an 8-inch SOI wafer using 193 nm deep ultraviolet (DUV) photolithography. The directional coupler and the MEMS actuator are formed by reactive ion etching (RIE). An aluminium (Al) thin film is then deposited and patterned for electrical connection to power the MEMS actuator. At last, hydrofluoric acid (HF) vapor etching is used to locally remove the BOX layer and release the directional coupler and the MEMS cantilever.

### Device characterization

For single-, dual- and multi-wavelength characterization, a set of tunable lasers (Santec TSL-510, 550, and 710) are adopted as the input, which are also used to measure the calibration matrix. For broadband characterization, a C + L band ASE broadband light source (Amonics ALS-CL-13-B-FA) is used as the input. A polarization controller is used to ensure that only TE-polarized light is injected into the on-chip spectrometer. The spectrometer chip is mounted on an XYZ stage for fiber-chip alignment, with the temperature controlled by a temperature controller. The light is coupled in and out of the chip through two on-chip adiabatically tapered edge couplers for broadband operation. The output light from the MECS is collected by a photodetector (Thorlabs PDA-10CS-EC). Input spectra are also recorded using an optical spectrum analyzer (OSA, Yokogawa AQ6370D) as references. The recorded spectra from OSA have a fine resolution of 20 pm. The reference input spectra are created by resampling the raw data into a 3001-point sequence with a coarser resolution of 100 pm. A semiconductor characterization system (Keithley 4200-SCS) is employed for time-sequenced bias voltage supply to implement time-domain modulation of the MECS. The sampling time grid is ~0.1 s, resulting in a total sampling time of ~6.4 s given 64 sampling steps. The sampling process can be accelerated by synchronizing the electrical voltage scanning and optical power detection with a shared trigger signal.

### Reconstruction implementation

Spectrum reconstruction is implemented using a MATLAB package of iterative regularization methods and test problems for linear inverse problems (IR Tools). This method can be used to reconstruct all types of spectra, including discrete, continuous, and mixed spectra. The CAED implementation in Python 3.6 uses Keras and its TensorFlow backend. Adam is used for optimization with a learning rate of 0.002. The learning rate is multiplied by 0.5 if the loss does not improve for 30 epochs. ReLU is used as the activation function for all layers except the last output layer. The final output layer utilizes LeakyReLU as the activation function. Our training of autoencoder is deployed on 4 pieces of NVIDIA TITAN Xp GPU. Training stops after 360 epochs. A batch size of 128 is used for all datasets. For the dataset volume of 30,000, each training epoch takes 1 s, the time taken for the total 360 epochs is ~6 min. After training, the model can be applied to random input samples, with each sample taking 0.15 μs for prediction.

## Data availability

The data that support the findings of this study are included in the article and its Supplementary Information. Other data are available from the corresponding authors upon request.

## Code availability

The codes in support of the results of this study are available from the corresponding authors upon request.

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

## Acknowledgements

This work was supported by National Natural Science Foundation of China (NSFC) Grant (62405173 to Y.M.), Shanghai Pujiang Program (23PJ1413700 to H.Z.), Fundamental Research Funds for the Central Universities (22120240566 to H.Z.), Agency for Science, Technology and Research (A*STAR) RIE Advanced Manufacturing and Engineering (AME) Programmatic Grant (A18A4b0055 to C.L.), Ministry of Education (MOE) Singapore Academic Research Fund Tier 2 (MOE-T2EP50220-0014 to C.L.), and National Research Foundation (NRF) Singapore Mid-Sized Centre Grant through the National Centre for Advanced Integrated Photonics (NRF-MSG-2023-0002 to C.L.).

## Author contributions

Y.M., J.Z., and H.Z. conceived the idea. J.Z. and H.Z. performed the device design, fabrication, and characterization with assistance from Y.M., Q.Q., H.C., Q.H., and H.W. H.Z. and J.Z. wrote the convolutional autoencoder algorithms and control programs for demonstration. The results were discussed by all authors. Y.M. and H.Z. wrote the manuscript with comments from J.Z., N.W., and Q.R. Y.M. and C.L. supervised the project.

## Competing interests

The authors declare no competing interests.
