## [Transparent Peer Review file · Nature Communications]

Denoising-autoencoder-facilitated MEMS computational spectrometer with enhanced resolution on a silicon photonic chip

Corresponding Author: Professor Chengkuo Lee

Version 0:

Reviewer comments:

Reviewer #1

(Remarks to the Author)

In this paper, the authors developed a single-channel MEMS computational spectrometer on the SOI platform, incorporating Autoencoder algorithms for denoising purposes. The paper detailed the technical framework and methodologies employed. However, upon careful evaluation, I believe that this spectrometer represents an incremental work over the existing MEMS-based spectrometric devices, while the application of Autoencoder techniques appears to serve primarily as a remedial measure to offset the intrinsic limitations of MEMS actuators. Additionally, if compared with those on-chip spectrometers based on photonic integrated circuits (PICs), this MEMS spectrometer's performance can merely be considered moderate, especially considering that PIC-based spectrometers generally do not face similar noise challenges. Furthermore, the experimental validation presented in the study is narrowly focused on the reconstruction of dual-peak spectral signals (as the Autoencoder is only trained with dual-peak signals). There are no demonstrations regarding any other spectral signals, such as multiple single-peak or continuous broadband signals. A demonstration at this level can not be accepted. Therefore, considering the high benchmarks for novelty and advancement required by a journal like Nature Communications, I recommend a rejection of this paper. Detailed comments are provided below for further reference:

1. First, the proposed device is highly similar to many previous proposed MEMS spectrometers in terms of working principles or device structure. For example, in Qiao Q, Liu X, Ren Z, et al. MEMS-enabled on-chip computational mid-infrared spectrometer using silicon photonics[J]. ACS Photonics, 2022, 9(7): 2367-2377, the authors have already reported a single-channel MEMS spectrometer that leverages the time-domain modulation of reconfigurable waveguide couplers to generate various interferograms for spectrum reconstruction.
2. Second, the reported spectrometer does not offer discernible performance benefits (with a 300nm bandwidth and 0.4nm resolution) in comparison to previously reported devices. There are many reported on-chip spectrometers that can readily achieve bandwidths spanning hundreds of nanometers and resolutions down to picometer scale. Also, the footprint of this MEMS device is rather large, which is also critical for on-chip spectrometers.
3. The design of transmission matrix (i.e. the channel spectral responses) is critical for the performance of computational reconstruction. Ideally, the channel spectral responses should exhibit low self-correlation and minimal cross-correlation, thereby enhancing spectral sampling performance and noise resilience. However, for MEMS spectrometers, the channel spectral responses are periodic interferograms with varying FSRs, which inherently are not suitable for efficient spectral sampling. So, despite the authors employing an Autoencoder strategy to improve noise tolerance up to a 30dB SNR this enhancement is not very impressive, especially considering the relatively high reconstruction errors depicted in Figure 5. For example, Ye Y, Zhang J, Liu D, et al. Research on a spectral reconstruction method with noise tolerance[J]. Current Optics and Photonics, 2021, 5(5): 562-575, it is shown that with well-designed channel spectral responses, even conventional reconstruction algorithms can exhibit good noise tolerance above 30dB SNR. Thus, a MEMS structure that incorporates a single tunable interferometer may naturally be unsuitable to serve as a reconstructive spectrometer.
4. The authors must demonstrate the reconstruction of a diverse range of incident spectra for a comprehensive validation of the spectrometer's versatility. However, given the FSR limitations of the interferograms and the necessity to tailor the training of the autoencoder for various input waveforms, I am skeptical about the MEMS spectrometer's ability to deliver satisfactory

performance across different complex incident spectra.

Reviewer #2

(Remarks to the Author)

This paper combines a couple of interesting techniques like MEMS and convolutional autoencoder denoising mechanism to boost the performance of silicon computational spectrometer. The performance, particularly the spectral resolution is demonstrated to be enhanced at noisy environment. Given the complex working scenarios of integrated spectrometers, improving its robustness against noise is vital. Therefore, I recommend the acceptance when addressing following issues:

1. I agree with the value of using MEMS in terms of modulation efficiency. However, for the compactness, the device shown in Fig.2a has a length of almost 7mm. While other demonstrated silicon computational spectrometers seem to be much more compact. I suggest to modify this argument.
2. Another drawback of using MEMS is the extremely high driving voltage. As frequently mentioned, integrated spectrometers are suggested to be used in portable devices and the drivers should be CMOS electronics. How to deliver such high voltage in practical applications?
3. The minimum SNR tested is 25dB, which is still quite optimistic and doesn't make too much sense. In real-life applications such as smartwatch based healthcare monitoring, the SNR can be as small as a few dB.
4. Recently it is reported that computational spectrometer is robust to temperature change "A.Li et al., An integrated single-shot spectrometer with large bandwidth-resolution ratio and wide operation temperature range, *Photonix* 4(1) 2023". Is this also true for this MEMS spectrometer? how would the performance vary at fluctuating temperature?
5. Besides resolution, the reconstruction accuracy should also be vulnerable to noise, but from the experimental results, it seems that the reconstructions are always clean. I suggest to go with more complex spectra and more data points in the reconstructions.
6. Some other information regarding the MEMS device should be provided, like the total insertion loss, the optical bandwidth, modulation speed etc. Particularly I am interested in the fiber/chip coupler, as it can support a wide range of wavelengths from 1.3 μ m to 1.6 μ m. A static transmission spectrum of the device is suggested to be included.

Reviewer #3

(Remarks to the Author)

Key Results

The authors demonstrate a chip-scale spectrometer with a bandwidth of 300 nm and resolution of 0.3 nm. They claim to overcome the challenges of chip-scale spectrometers, namely poor resolution, by leveraging a time-domain MEMS controlled system with neural network processing for enhancing resolution. The authors rely on MEMS actuation of active optical elements to achieve spectral separation. The spectral resolution claimed is impressive, especially considering the proposed bandwidth. This resolution matches some more conventional spectrometers, by leveraging computational reconstruction of spectra using a neural network.

Questions and Comments

1. Based on the current form of the manuscript, the key novelty here seems to be on the analysis of the data using the deep learning CAED method. However, innovation on the photonics side must be inferred. Can the authors state the innovation in their design more clearly in the abstract/introduction because currently it appears that there is no major innovation in the physical on-chip design?
2. The authors provide good examples of spectrum reconstruction demonstrating the resolution of the device; however, bandwidth is another important parameter for any spectrometer. While the authors claim a bandwidth of 300 nm, there is no direct evidence of this in the figures beyond the spectral peaks at 1.33 μ m and 1.57 μ m in Figure 3a, which is only ~250 nm. Can the authors provide a demonstration of the 300 nm bandwidth of the device? Perhaps, they can use a broadband input source like a mercury halogen lamp?
3. Another interesting demonstration would be analyzing the spectrum of a real sample with features within the bandwidth of this device. The authors speak at length on the applications and merits of on-chip spectroscopy, so for a journal of this caliber and broad readership, a more practical demonstration would add significant value.
4. Figure 2. Even though Figures b, c are zoomed versions of a, the axis labels and units should be included for clarity.
5. While the result in Figure 2b,c is impressive and demonstrates the resolution of the system, a similar demonstration without applying denoising would help the reader understand the true contribution of CAED to this work and the technology at large.
6. What is the efficiency or run time of the algorithm the authors have developed? Most practical applications of spectrometers require moderate to high speed, so an understanding of the processing time will help gauge the efficacy of this approach.
7. Could the authors please add more details to the methods section? Important information such as what specific type/model of tunable laser was used, what kind of detector was used, what was used for power supply and voltage modulation is not included. These specifications will allow readers to more accurately repeat the results presented here.
8. Similarly, details of how light is coupled into and out of the system needs to be added to the Methods section.

Recommendation

Overall, I believe this manuscript is well-written and presents a significant improvement in the on-chip spectrometer space. After some proposed revisions, this manuscript should be ready for publication.

Reviewer #4

(Remarks to the Author)

In this work, the authors present a novel MEMS-based spectrometer architecture along with a signal processing technique which suppresses noise and subsequently improves spectral resolution. The authors fabricate an experimental device and perform some experiments to demonstrate that their methods work on the fabricated device. I believe that both the spectrometer design and approach to denoising are significant and will contribute meaningfully to the advancement of on-chip spectrometers. Power consumption of existing thermally-actuated spectrometers and sensitivity to noise are important issues in current technology, and this paper aptly points out these issues and provides effective methods to address them. In particular, the thorough analysis of the relationship between the SNR and the reconstructed resolution is an important analysis which provides a more holistic understanding of the system performance, and is a great inclusion, as is the noise analysis of the MEMS structure in Supplementary Note 4. Finally, the paper is well-written, cites appropriate references, and has clear and concise analysis supporting the MEMS design, spectral extraction, and denoising processes.

Though the foundations of this work are strong, I believe that the experimental testing performed is not adequate to support the claimed resolution, signal-to-noise, and bandwidth specifications of the realized systems. In particular, it appears as if this spectrometer uses 64 measurements (corresponding to 64 different positions of the MEMS actuator) to extract a spectrum with 3001 data points, representing a highly underdetermined system. While numerous previous works have shown that spectra can be reconstructed even in such highly underdetermined cases (e.g., ref. [60] in the manuscript), such systems must be tested on a wide variety of spectral sources to demonstrate that the regularization being used to enable reconstruction in such a highly underdetermined problem is in fact valid for diverse input spectra. For example, ref. [60] demonstrated reconstruction of a spectrum with $> 10,000$ spectral channels from just 729 measurements, but rigorously backed up this result through both theoretical analysis, simulations, and experimental testing of the spectrometer on narrowband, broadband, and mixed sources. This current manuscript, on the other hand, appears to only perform reconstruction on pairs of narrowband sources, and as such its versatility is not demonstrated. Supplementary Note 6 contains some testing showing that the reconstruction accuracy decreases with increasing wavelength spacing of dual-wavelength sources due to how the denoising system was trained, indicating that performance of both the denoising and regression-based reconstruction may deteriorate when the measured spectrum deviates from the training data. The claimed figures for resolution, bandwidth, and signal-to-noise performance can only be fairly compared to other devices in literature (as is done in Fig. 6) if these figures hold true across a wide range of possible spectral measurements.

To address my concerns I recommend the following:

1. Demonstrate that the proposed spectrometer can reconstruct broadband and mixed sources. If further experiments are possible, this would require simply collecting interferograms (using the procedure described in the manuscript) for a broadband source (e.g., an erbium doped fiber amplifier with no input or the output of a fiber-coupled superluminescent diode) and running the proposed spectral extraction on these interferograms. The same experiment can be repeated using a mixed broadband/narrowband source simply by using a 50/50 coupler and inputting both a broadband source and a tunable laser source into the device simultaneously.
2. Perform some dual-wavelength interferogram measurements used for resolution characterization experimentally. It appears that all dual-wavelength extraction results in this manuscript are the result of simply adding two independent single-wavelength interferograms and performing extraction on the result (with some additional noise added). While this method is understandable for the purposes of training the denoising network and initial demonstrations, I believe that for final demonstration of the system's resolution and performance, experimental measurements of two different laser sources simultaneously should be performed. This would demonstrate robustness to nonidealities in the input that are not present in the synthetic data. Again, this experiment could be performed simply using two laser sources and a 50/50 coupler.

I understand that additional experiments cannot always be performed, and that the authors may not have easy access to this equipment. If this is the case, I would still recommend that at least the broadband/mixed source experiments be simulated using the measured calibration matrix of the spectrometer, which is a straightforward process. If these tests are performed in simulation, realistic nonidealities should be included where possible (e.g. for the dual-wavelength interferogram measurements, differing amplitudes of the two laser sources).

Finally, I would like to emphasize that if the results of this additional testing or simulation indicate worse performance of the proposed system on this more diverse set of test data, I would still encourage the authors to present this data, and I would support publication of such data. Given the diverse range of spectrometer architectures and reconstruction methods being published in the field, it is crucial that publications show clearly both the strengths and weaknesses of their system, so that different approaches can be compared and further advancements can be made to improve the shortcomings of existing architectures.

If these additional experiments or simulations can be added to the manuscript, I believe that this is a very original and impactful piece of research that would be useful for many other researchers in this field, and would support publication.

Version 1:

Reviewer comments:

Reviewer #1

(Remarks to the Author)

I appreciate the authors' detailed responses and revisions. The added experiments and discussions have significantly enhanced the quality of the paper, especially the exploration of the multi-stage structure and the noise-oriented autoencoder scheme. However, there are still issues that need to be addressed.

1. First, there remains a question about the novelty of the single-bus MEMS structure. Although the authors have provided a thorough explanation distinguishing this work from their previous publication in ACS Photonics, the changes in the DC coupling conditions, supporting structure, and the grating coupler versus edge coupler setup still seem incremental. I would like to understand whether these modifications have led to decisive performance improvements, and if so, what those improvements are. I believe this should be discussed directly in the main text, possibly even in the introduction.
2. Drawing inspiration from multi-stage spectrometers based on photonic circuits, the authors propose a multi-stage MEMS-based spectrometer. This approach is supported by simulations of dual-peak signal reconstruction, which demonstrate performance improvements. Given that the single-channel MEMS spectrometer's performance presented in the paper is moderate, the multi-stage MEMS spectrometer concept is indeed worth further exploration. Ideally, a multi-stage device should be fabricated and experimentally validated, but given the significant workload, a more extensive simulation-based investigation would be acceptable. Specifically, the simulations could be expanded in two ways: 1) by modelling a wider variety of input spectra, and 2) more importantly, by examining the impact of noise in a multi-stage system—whether there might be cumulative effects and how the proposed noise reduction algorithms would perform under such conditions. These points are crucial and deserve detailed discussion in the main text.
3. Additionally, the paper could benefit from further discussions on topics such as the device's sampling time, fabrication tolerance, and temperature tolerance to provide a more comprehensive view of the work.

Reviewer #2

(Remarks to the Author)

As for my previous comments, I could see the authors trying to address them in a high quality, such as additional measurements about temperature robustness, reconstructions of complex spectra and additional analysis regarding the SNR. Overall, I am satisfied with current revision. But as stated by the authors, the MEMS based spectrometer is still accompanied with several drawbacks compared with its counter-parts, such as high voltage, large footprint etc. Therefore, I suggest the authors precisely positioning the suitable applications of such spectrometer, instead of presenting it as a general integrated spectrometers.

Reviewer #3

(Remarks to the Author)

After reviewing the authors' response to my previous comments, I believe the authors have faithfully modified their manuscript to account for my concerns and questions. In its current form, I believe the manuscript is scientifically sound with sufficient evidence presented to support the authors' claims. Overall, it is my opinion that the results presented here are highly relevant and of benefit to the broader community. Therefore, I recommend publication of this modified manuscript in its current form.

Reviewer #4

(Remarks to the Author)

The authors' revisions have added a substantial amount of information to the paper and have satisfactorily addressed my previous comments. I support publication once the following questions are answered/clarified in the manuscript:

1. The authors include a reconstruction of the emission spectrum from an Amonics ASE C + L band source.
 - 1a. (minor) Could the authors provide the model number of this source so readers can more easily compare the spectrum measured in the paper to the spectrum specified by the manufacturer?
 - 1b. Referring to the Amonics website, it seems that their C+L band ASE sources cover a much broader range than the reconstructions shown in Fig. 5i-j (this is also evidenced by the cut-off peak at the short wavelengths in these figures). Given the wide bandwidth of the spectrometer, why was the entire reconstruction not shown in these figures?
2. The authors include a spectrum reconstruction of the absorption spectrum of N-methylaniline, which is a good way to demonstrate the utility of the spectrometer. However, I am confused by the description of how this spectrum is measured, or rather, mimicked: "Considering the limitation in the experimental setup, we mimic the input absorption spectrum of N-methylaniline by tailoring the tunable laser intensity at each wavelength." Does this mean that the authors simply collected dozens of input interferograms from narrowline sources, adjusted their amplitude, added them together, and performed the reconstruction? If so, I would argue that this is not a realistic broadband reconstruction and I question its inclusion in the paper.

3. The authors mention using modified regularization for the broadband reconstruction (Eq. 14 in the manuscript). Are high quality reconstructions of all sources obtained if this same regularization is used for all reconstructions? Or do the results shown in this work rely on picking the “best” regularization method depending on the nature of the spectrum? I understand that this same mixed broadband/narrowline regularization has been demonstrated in previous works (as cited by the authors), and can be used on diverse inputs - if this is the case, I recommend the authors use (14) on all reconstructions in the paper. If not, the authors should emphasize this point, as requiring a priori information about the nature of an unknown spectrum to pick the best regularization method/hyperparameters is a major limitation for practical use.

4. Fig. 5j shows a great quality mixed broadband/narrowband reconstruction, which is nice to see. Can the authors give more information regarding the OSA settings used to obtain the reference spectrum? In particular, it would be useful to know the resolution setting of the OSA, as this can impact the relative height between the narrowband peak and the broadband features.

5. I may have missed it, I would like clarification regarding exactly what “input spectrum” refers to in each figure. Was the “input spectrum” always measured using an OSA? Certain figures which make me especially curious about this are 3b-3c, 5d-5g, etc. (reconstructions of narrowband sources). It seems like the “input spectrum” has the exact same resolution and positioning of x-axis points as the reconstruction. Is this an OSA measurement with the measurement parameters chosen to be the same as the reconstruction? Or is this a synthetic estimate of the input spectrum with delta functions simply placed at the wavelengths that the tunable lasers were set to?

Version 2:

Reviewer comments:

Reviewer #1

(Remarks to the Author)

Again, I appreciate the authors' detailed responses in the previous revision round. After carefully reviewing the paper, a few important issues still need to be addressed:

1. In the Abstract and Introduction, the authors added some new statements, such as: “The proposed CAED-facilitated MEMS spectrometer presents a promising solution for broadband high-resolution spectral analysis in applications demanding precision, power efficiency, and noise resilience, such as personal healthcare, remote sensing, and marine research.” However, this might be an overstatement. The mechanical noise is an inherent issue for MEMS, not for other free-space or PIC-based spectrometers. The authors could argue that their CAED scheme helps mitigate noise in MEMS, but noise tolerance should not be presented as a key selling point. Also, in general, I don't think waveguide-based spectrometers are suitable for applications like remote sensing or health care monitoring, as the micrometer-scale mode size limits the amount of optical power that can be coupled. In real-world scenarios, such as remote sensing, where the spectra to be detected are typically scattered light with low power density, the proposed MEMS spectrometer may struggle to capture any meaningful signal. Thus, I strongly recommend revising these statements to avoid potential misunderstandings.

On the other hand, using “high-resolution” in the title seems somewhat inappropriate, as many on-chip spectrometers have already achieved picometer-level resolution, while the device presented here only reaches 0.2 nm. Please consider revising this.

2. Additionally, the authors claim a 300 nm broadband bandwidth, but it seems that there is only one dual-peak laser experiment to support this, while all the remaining reconstruction experiments focus on the long wavelength region. I recommend adding some simulation results (especially for continuous spectra) to demonstrate that the device still performs well in the short wavelength range.

3. In lines 381 to 383, the authors state: “In order to show the efficacy of our spectrometer in real-life applications, we further demonstrate the reconstruction of the absorption spectrum of N-methylaniline, which possesses a well-defined absorption fingerprint near 1.5 μm (see Supplementary Note 14)”. Such a statement is vague and gives the impression that the authors conducted an experiment, but upon reviewing Supplementary Note 14, I found this to be merely a simulation. Additionally, the description of the simulation is confusing. The authors mention adjusting the tunable laser intensity at each wavelength to mimic N-methylaniline's absorption spectrum, but it's unclear whether they are reconstructing a discrete or continuous spectrum, which is a significant distinction for computational spectrometers. Therefore, I recommend that the authors remove this section.

4. A structural suggestion: I recommend adding a figure to the main text to better explain the multi-stage design, as currently, all the details are in the supplementary materials, making it difficult for readers to grasp the concept intuitively. A dedicated figure in the main text would visually illustrate the multi-stage design and further highlight its role in improving the spectrometer's performance.

Reviewer #4

(Remarks to the Author)

The authors' changes are thorough, thoughtful, and have satisfactorily addressed my comments. The manuscript is well-written and convincingly demonstrates the device and novel reconstruction methods. I recommend publication.

Version 3:

Reviewer comments:

Reviewer #1

(Remarks to the Author)

Thanks for all the efforts and I have no more comments

Dear Reviewers,

Thank you very much for your great efforts, valuable comments, and helpful suggestions, all of which greatly help improve the quality of our manuscript. We have carefully gone through all the comments and revised the manuscript accordingly. The following are our point-to-point responses along with your comments. Accordingly, the revised portions are **marked in red** in the revised manuscript.

POINT-BY-POINT RESPONSE TO THE REVIEWERS' COMMENTS

To Reviewer #1:

Comment 1: In this paper, the authors developed a single-channel MEMS computational spectrometer on the SOI platform, incorporating Autoencoder algorithms for denoising purposes. The paper detailed the technical framework and methodologies employed. However, upon careful evaluation, I believe that this spectrometer represents an incremental work over the existing MEMS-based spectrometric devices, while the application of Autoencoder techniques appears to serve primarily as a remedial measure to offset the intrinsic limitations of MEMS actuators. Additionally, if compared with those on-chip spectrometers based on photonic integrated circuits (PICs), this MEMS spectrometer's performance can merely be considered moderate, especially considering that PIC-based spectrometers generally do not face similar noise challenges. Furthermore, the experimental validation presented in the study is narrowly focused on the reconstruction of dual-peak spectral signals (as the Autoencoder is only trained with dual-peak signals). There are no demonstrations regarding any other spectral signals, such as multiple single-peak or continuous broadband signals. A demonstration at this level can not be accepted. Therefore, considering the high benchmarks for novelty and advancement required by a journal like Nature Communications, I recommend a rejection of this paper. Detailed comments are provided below for further reference:

Answer 1: Thank you very much for your valuable efforts and expert comments. We would like to highlight again the main contribution of our work. Compared with reported PIC-based spectrometers that commonly rely on the power-hungry thermo-optic tuning, the electrostatic MEMS reconfiguration employed in our work realizes effective tuning with **ultra-low power consumption** (please see our answer to your Comment 3 below for a detailed characterization of the power consumption). Simultaneously, we develop a convolutional autoencoder denoising (CAED) mechanism to solve the noise issue and improve the spectral reconstruction performance. It is noted that all devices including PIC-based spectrometers are affected by noise, although the noise sources may be different. Our developed CAED mechanism would have **broad applicability** in various types of spectrometers, thanks to its weak restrictions from noise sources.

Nonetheless, we totally agree that reconstruction demonstrations of more diversified spectra including multi-peak and broadband ones are necessary. We have revised the manuscript based on your valuable comments, and a detailed point-by-point response is provided below.

Comment 2: First, the proposed device is highly similar to many previous proposed MEMS spectrometers in terms of working principles or device structure. For example, in Qiao Q, Liu X, Ren Z, et al. MEMS-enabled on-chip computational mid-infrared spectrometer using silicon photonics[J]. ACS Photonics, 2022, 9(7): 2367-2377, the authors have already reported a single-channel MEMS spectrometer that leverages the time-domain modulation of reconfigurable waveguide couplers to generate various interferograms for spectrum reconstruction.

Answer 2: Thank you for the comment. Firstly, we would like to point out that this ACS Photonics paper is our previous work and has been properly cited in this manuscript. Besides, there is no other reported work that is similar to our proposed device in terms of both working principle and device

structure, to the best of our knowledge. Compared with our previous ACS Photonics work, we would like to highlight five points of innovations in terms of working principle and device structure:

1) In the previous work, it was thought that the directional coupler needs to be modulated from the initial coupled condition to a fully decoupled condition to facilitate the spectral reconstruction. Such a coupling condition transition requires a large tuning displacement of the MEMS cantilever, which exceeds the tuning range that can be offered by the release of the BOX layer, with the electrostatic pull-in effect taken into account^{R1}. As a result, a non-standard and low-yield flip-chip bonding process was needed to enable the large tuning displacement. Here in this work, we have first investigated the relationship between the device tuning range and the spectral reconstruction performance using correlation analysis. The self- and cross-correlations of three tuning ranges (from the initial coupled condition to a weakly/moderately/fully decoupled condition) are studied. As shown in **Fig. R1a, b**, the moderately decoupled condition results in the smallest self-correlation width as well as considerably low cross-correlation. The fully decoupled condition can further decrease the cross-correlation, but at the expense of larger self-correlation width, thus tends not to improve the spectral resolution. The dual-wavelength spectra reconstruction results as shown in **Fig. R1c-e** confirm that the moderately decoupled condition offers the finest reconstruction resolution. The adoption of a moderately decoupled condition in our work not only guarantees a satisfactory spectral resolution but also reduces the required tuning range to a value that can be offered by the release of the BOX layer, which largely simplifies the device configuration and fabrication process.

Fig. R1 a Self-correlation function and b Cross-correlation of three tuning ranges from the initial coupled condition to a weakly/moderately/fully decoupled condition. c-e Spectrum reconstruction results at the weakly/moderately/fully decoupled condition, respectively. The moderately decoupled condition exhibits the finest reconstruction resolution.

- 2) In the previous work, subwavelength grating (SWG) was employed as the supporting structure for the suspended waveguides in the mid-infrared. However, when migrating the SWG design to the telecommunication wavelengths, the minimum feature size is typically below 150 nm, which is challenging for common silicon photonics foundries^{R2,3}. In this work, we employ and optimize a trapezoidal supporting structure for low-loss suspension of the waveguides, as shown in **Fig. R2**. Compared to SWG, the trapezoidal supporting structure possesses lower fabrication restriction and better wavelength scalability. We fix the beam width to be 0.4 μm , and then optimize the structural parameters of the trapezoid. With a height, baseline length, and topline length of 0.4, 8, and 3 μm , respectively, the insertion loss of the trapezoidal supporting structure is optimized to 0.098 dB, which is comparable with the reported elliptical supporting structure^{R4}. The minimum feature size of 0.4 μm can be conveniently fabricated by common silicon photonics foundries.

Fig. R2 Design and optimization of trapezoidal supporting structure. **a** Schematic illustration of structural parameters. **b** Optimization of the trapezoid height. **c** Optimization of the trapezoid baseline length. **d** Optimization of the trapezoid topline length, inset shows the electric field profile with the optimized structural parameters.

- 3) Unlike the normal directional coupler employed in the previous work, here we simplify the bus waveguide to a straight waveguide, which reduces the propagation loss and improves the signal-to-noise ratio (SNR).
- 4) Unlike the grating couplers employed in the previous work, adiabatically tapered edge couplers are used to facilitate the fiber-to-chip coupling in our work, which significantly enlarge the device bandwidth. **Fig. R3** shows the transmission spectrum of the edge coupler, showing a low coupling loss of below 4 dB/facet across the wavelength range of 1.3-1.6 μm .

Fig. R3 Transmission spectrum of the edge coupler.

- 5) The residual stress in the Si device layer will induce significant unevenness of the cantilever and deteriorate the coupling between the two waveguides, limiting the achievable coupling length. Thus, the long coupling length was offered by two cascaded waveguide couplers in our previous work. Here in this work, we deposit an Al layer with proper length and thickness onto the Si cantilever to improve the stiffness and thus the flatness of the cantilever^{R5}. Consequently, we are able to implement a long coupling length using a single waveguide coupler. The good flatness of the cantilever ensures uniform coupling along the whole coupling length, which is confirmed by the measured calibration matrix.

The above results and discussions have been added as **the third paragraph of the MEMS spectrometer part in the Results section** and **Supplementary Note 3** in the revised manuscript.

References:

- R1. O’Brien, G., Monk, D. J. & Lin, L. MEMS cantilever beam electrostatic pull-in model. *Proc. SPIE* **4593**, 31–41 (2001).
- R2. Cheben, P., Halir, R., Schmid, J. H., Atwater, H. A. & Smith, D. R. Subwavelength integrated photonics. *Nature* **560**, 565–572 (2018).
- R3. Chen, L. R., Member, S. & Wang, J. Subwavelength Grating Waveguide Devices for Telecommunications Applications. *IEEE J. Sel. Top. Quantum Electron.* **25**, 8200111 (2019).
- R4. Fukazawa, T., Hirano, T., Ohno, F. & Baba, T. Low Loss Intersection of Si Photonic Wire Waveguides. *Jpn. J. Appl. Phys.* **43**, 646–647 (2004).
- R5. Gyger, S. *et al.* Reconfigurable photonics with on-chip single-photon detectors. *Nat. Commun.* **12**, 1408 (2021).

Comment 3: Second, the reported spectrometer does not offer discernible performance benefits (with a 300nm bandwidth and 0.4nm resolution) in comparison to previously reported devices. There are many reported on-chip spectrometers that can readily achieve bandwidths spanning hundreds of nanometers and resolutions down to picometer scale. Also, the footprint of this MEMS device is rather large, which is also critical for on-chip spectrometers.

Answer 3: Thank you for the comment. We provide a comprehensive performance comparison of reported on-chip spectrometers in **Table R1**, which is added as **Table S2 in Supplementary Note 13** in the revised manuscript.

Table R1. Performance comparison of reported on-chip spectrometers.

Ref. no. in the main text	Type	Structure	Resolution (nm)	Bandwidth (nm)	Footprint (μm^2)	Voltage (V)	Power consumption (mW)
13	DO	Disordered structure	3.4	40	50×200	/	/
14	DO	Disordered structure	0.25	30	12.8×30	/	/
30	DO	Disordered structure	0.75	25	50×100	/	/
54	DO	PhC reflector	2.5	55	480×800	/	/
55	DO	Branched waveguide	3	100	12×63	/	/
56	DO	Echelle grating	1.2	30	1100×1420	/	/
15	NF	FP cavity array	0.51	102.7	43×600	8.5	873
19	NF	Two coupled MRRs	0.04	100	60×60	NM	85
57	NF	Euler MRR + cascaded MRR array	0.005	10	3.5×10 ⁵	NM	504
58	NF	MRR array	0.6	60	1×10 ⁶	/	/
59	NF	MRR + random gratings	0.2	60	215×310	NM	33.23
10	FT	Single MZI	3.05	56.18	1×10 ⁶	200	5100
11	FT	SWIFTS	4	96	22×512	/	/
12	FT	tFTS + SHS	0.125	200	5500×6000	135	2400
42	FT	Cascaded DCs	4	300	NM	/	/
52	FT	MRR + MZI	0.47	90	NM	NM	1835
53	FT	MZI with embedded switches	0.2	20	630×2820	NM	99

7	CR	Stratified waveguide filter array	0.45	180	35×260	/	/
16	CR	PhC nanobeam cavity array	5	70	6×111	/	/
18	CR	Cascaded PhC nanobeam cavities	0.32	16	18×250	NM	90
60	CR	Cascaded MZIs	0.01	200	1900×3700	NM	2100
61	CR	Distributed MRRs	0.03	115	2000×7600	/	/
This work	CR	Single DC	0.4 (with denoising) 1.2 (without denoising)	300	38×2030	29.9	6.987×10^{-2}

DO: dispersive optics

NF: narrowband filtering

FT: Fourier transform

CR: computational reconstruction

PhC: photonic crystal

FP: Fabry-Pérot

MRR: microring resonator

MZI: Mach-Zehnder interferometer

SWIFTS: stationary-wave integrated Fourier-transform spectrometer

tFTS: tunable Fourier-transform spectrometer

SHS: spatial heterodyne spectrometer

DC: directional coupler

NM: not mentioned

From **Table R1**, we discuss three aspects below:

- 1) Our device features ultra-low power consumption thanks to the utilization of electrostatic MEMS tuning. We characterize the tuning energies of our device at different applied voltages using the method described in Ref. ^{R6}. The average tuning power is derived as the tuning energy divided by the response time and plotted in **Fig. R4a**. Even at the maximum applied voltage of 29.9 V, the average tuning power is less than 70 μ W, which is three orders of magnitude lower than those of reported on-chip spectrometers using thermo-optic tuning, which consumes over 30 mW power. Additionally, the capacitor nature of the electrostatic MEMS actuator allows nearly zero standby power consumption, as measured and shown in **Fig. R4b**.

Fig. R4 Power consumption of our device. **a** Average tuning power required to reach the corresponding applied voltages. **b** Static power required to hold at the corresponding voltages.

The above results and discussions have been added to **the end of the Spectrum reconstruction part of the Results section** and **Fig. 2j, k** in the revised manuscript.

- 2) Most of these reported high-performance spectrometers rely on physically multi-stage or multi-channel structures to create abundant sampling channels. It can be inferred that with zero correlation between different sampling channels, only a small number of sampling channels is needed to reconstruct unknown spectra. However, if the transmission spectra of sampling channels exhibit significant similarity, reducing the information capacity provided by the transmission sampling matrix, a larger number of sampling channels would be required for satisfactory spectrum reconstruction. Therefore, architectural advancements are proposed to create additional sampling channels, so as to compensate for the effect brought by the dependent channel responses (e.g., Refs. ^{60,61} in the main text). Our work is a pioneer work in MEMS spectrometers, thus we limit our study to the simplest case, i.e., single physical stage and single physical channel. By leveraging a multi-stage structure, the performance of our MEMS spectrometer is expected to be improved due to the significantly increased number of sampling channels as well as decreased correlations. A trade-off needs to be made between the reconstruction performance and the device footprint. Please see our answer to your Comment 4 below for the detailed discussion.
- 3) Our device is significantly smaller if compared with the above-mentioned physically multi-stage/channel spectrometers (e.g., Refs. ^{60,61} in the main text), while less compact if compared with some of the narrowband-filter-based spectrometers (Ref. ^{15,19} in the main text). Nonetheless, narrowband filters usually suffer from an inherent compromise between the SNR and spectral resolution^{R7}. The footprint of our device can be further reduced by reducing the coupling gap of the waveguide coupler. The spectral resolution of the proposed spectrometer can be estimated by the Rayleigh criterion^{R8}:

$$\delta\lambda = \frac{\lambda^2}{\Delta n_{\max} L} \quad (\text{R1})$$

where Δn_{\max} represents the maximum effective index difference Δn between SM0 and SM1 during the whole tuning process. A smaller coupling gap will lead to stronger coupling between the cantilever and straight waveguides and thus a larger Δn . The coupling gap can be further narrowed by using more advanced lithography. To be more practical, here we propose a new spectrometer design, in which the cantilever actuator is replaced with a comb-drive actuator, thus changing the current out-of-plane reconfiguration scheme to an in-plane reconfiguration scheme, as illustrated in **Fig. R5a**. The waveguides are kept with the same thickness of 220 nm

and width of 350 nm. The two waveguides are initially separated by 460 nm. The comb fingers in the comb-drive actuator measure 250 nm in width and 4 μm in length. Upon the application of bias voltage between the fixed and movable fingers, the movable waveguide is pushed toward the fixed waveguide, reducing the coupling gap. As shown in **Fig. R5b**, under an applied bias voltage of 33.6 V, the coupling gap can be reduced to 50 nm, a value beyond the resolution offered by common silicon photonics foundries, leading to a drastic increase of Δn . As a result, a noise-free reconstruction resolution of 0.3 nm can be achieved with a coupling length of only 500 μm using the calibration matrix shown in **Fig. R5c**, as depicted in **Fig. R5d**.

Fig. R5 Design of a compact MEMS spectrometer by employing in-plane reconfiguration. **a** Schematic illustration of the spectrometer, where the in-plane reconfiguration is realized by the integrated pair of comb-drive actuators. **b** Coupling gap and Δn as functions of applied bias voltage. **c** Calibration matrix of the proposed spectrometer. **d** Dual-wavelength spectrum reconstruction showing a resolution of 0.3 nm.

The above results and discussions have been added to **the second paragraph of the Discussion and Conclusion section** and **Supplementary Note 15** in the revised manuscript.

References:

- R6. Kim, D. U. *et al.* Programmable photonic arrays based on microelectromechanical elements with femtowatt-level standby power consumption. *Nat. Photonics* **17**, 1089–1096 (2023).
- R7. Xu, H., Qin, Y., Hu, G. & Tsang, H. K. Cavity-enhanced scalable integrated temporal random-speckle spectrometry. *Optica* **10**, 1177–1188 (2023).
- R8. Kita, D. M. *et al.* High-performance and scalable on-chip digital Fourier transform spectroscopy. *Nat. Commun.* **9**, 4405 (2018).

15. Sun, C. *et al.* Broadband and High-Resolution Integrated Spectrometer Based on a Tunable FSR-Free Optical Filter Array. *ACS Photonics* **9**, 2973–2980 (2022).
19. Xu, H., Qin, Y., Hu, G. & Tsang, H. K. Breaking the resolution-bandwidth limit of chip-scale spectrometry by harnessing a dispersion-engineered photonic molecule. *Light Sci. Appl.* **12**, 64 (2023).
60. Yao, C. *et al.* Integrated reconstructive spectrometer with programmable photonic circuits. *Nat. Commun.* **14**, 6376 (2023).
61. Yao, C. *et al.* Broadband picometer-scale resolution on-chip spectrometer with reconfigurable photonics. *Light Sci. Appl.* **12**, 156 (2023).

Comment 4: The design of transmission matrix (i.e. the channel spectral responses) is critical for the performance of computational reconstruction. Ideally, the channel spectral responses should exhibit low self-correlation and minimal cross-correlation, thereby enhancing spectral sampling performance and noise resilience. However, for MEMS spectrometers, the channel spectral responses are periodic interferograms with varying FSRs, which inherently are not suitable for efficient spectral sampling. So, despite the authors employing an Autoencoder strategy to improve noise tolerance up to a 30dB SNR this enhancement is not very impressive, especially considering the relatively high reconstruction errors depicted in Figure 5. For example, Ye Y, Zhang J, Liu D, et al. Research on a spectral reconstruction method with noise tolerance[J]. *Current Optics and Photonics*, 2021, 5(5): 562-575, it is shown that with well-designed channel spectral responses, even conventional reconstruction algorithms can exhibit good noise tolerance above 30dB SNR. Thus, a MEMS structure that incorporates a single tunable interferometer may naturally be unsuitable to serve as a reconstructive spectrometer.

Answer 4: Thank you for the expert comment. You raised concerns about whether the proposed MEMS spectrometer can offer satisfactory self- and cross-correlations since it seems that the channel spectra are periodic interferograms with varying FSRs. To address this concern, we conduct a series of correlation analyses and present our findings on self-correlation, cross-correlation, and the Pearson correlation coefficient, unveiling a satisfactory correlation performance (**Part 1** below). Additionally, we delve into the principle to demonstrate that our interferogram in each distinct sampling channel does not exhibit a simple periodic pattern with a constant FSR, so that interferograms across different sampling channels do not just vary in FSR (**Part 2** below). Specifically, we compare our interferogram (using both simulated models and experimental data) to a mere periodic pattern $\cos^2(\lambda)$ for intuitive visualization.

Nevertheless, considering that our proposed MEMS spectrometer is just in its nascent stage of development as a physically single-stage and single-channel device (similar to the Michelson interferometer in the early days of on-chip spectrometers), it holds immense potential for the development of multiple physical stages or channels. On that note, some notable works proposing strategies to enhance sampling channel diversity (e.g., Refs. ^{60,61} in the main text) offer valuable insights for the further development of MEMS spectrometers. These strategies are indeed applicable to the expansion of the single-channel MEMS spectrometer. Here, we discuss through simulations how the self- and cross-correlations of MEMS spectrometers can be improved by employing multi-stage structures (**Part 3** below).

- 1) **Correlation analysis.** The performance of a spectrometer is expected to achieve the two properties: (i) The spectral response at each sampling channel should have diverse features, so that the correlation length in the wavelength span should be small to provide high spectral resolution; (ii) The transmission spectra for any two sampling channels should be very different, i.e., orthogonal, to provide a transmission sampling matrix with large rank^{R9}. Hereby, self- and cross-correlations are studied on the transmission sampling matrix to evaluate these two properties.

For the transmission sampling matrix, its row vector is referred to as spectral responses (at different channels), while its column vector is the temporal speckle at a single wavelength. Each channel's spectral response should contain adequate sharp features to provide high sampling resolution, thereby achieving high spectral resolution. The spectral resolution depends on the change in wavelength required to generate an uncorrelated speckle pattern, which can be quantified by the **spectral self-correlation function**:

$$C(\Delta\lambda, x) = \frac{\langle I(\lambda, x)I(\lambda+\Delta\lambda, x) \rangle}{\langle I(\lambda, x) \rangle \langle I(\lambda+\Delta\lambda, x) \rangle} - 1 \quad (\text{R2})$$

where $I(\lambda, x)$ is the intensity recorded by the sampling channel x for input wavelength λ , and $\langle \dots \rangle$ represents the average over λ . In our MEMS spectrometer, a small shift in the input wavelength will cause a significant change in the transmitted speckle pattern. The average of the computed $C(\Delta\lambda)$ of our spectrometer across different channels x , i.e., the $\langle C(\Delta\lambda, x) \rangle_x$, is depicted in **Fig. R6a**. C is normalized to 1 at $\Delta\lambda = 0$, and its half-width at half-maximum, namely self-correlation width $\delta\lambda$, is 0.28 nm, meaning that a wavelength shift of $\delta\lambda$ is sufficient to reduce the degree of correlation of the speckle pattern to 0.5. $\delta\lambda$ provides an estimation of the spectral resolution since it is impossible to resolve two wavelengths with highly correlated speckle patterns. The actual resolution also depends on the reconstruction algorithm and the experimental noise of the measurements. The decrease in resolution caused by noise is exactly the part that we would like to get rid of so that the resolution can be rectified to the optimal 0.28 nm, even in a noisy environment. Our proposed autoencoder scheme rectifies the resolution from 1.2 to 0.4 nm.

Since the nature of the computational spectrometer is to use much fewer equations to solve many unknown values, we would expect that each equation (sampling channel) is independent of the others. Ideally, the rank of the transmission sampling matrix needs to be as large as possible, in other words, the transmission spectra of different sampling channels have to be **ideally orthogonal with zero cross-correlation**. Here, we calculate the cross-correlation function and the absolute value of the averaged cross-correlation between one specific sampling channel and all the other sampling channels (**Fig. R6b**). It can be seen that the spectra of these sampling channels contain diverse features with very little cross-correlation value, approaching almost 0, which proves the effectiveness of our designed time-domain modulation channels. Another direct visualization of the sampling channel dependencies is the Pearson correlation coefficient between the spectral responses of any two distinct sampling channels, as depicted in **Fig. R6c**. It is observed that the sampling channels all exhibit high linear independence. In summary, our MEMS spectrometer satisfies the two criteria of low self-correlation and minimal cross-correlation, by numerical calculations using the transformation sampling matrix obtained from the MEMS spectrometer.

Fig. R6 **a** Spectral self-correlation function $C(\Delta\lambda)$ derived from the measured $I(x, \lambda)$ of our MEMS spectrometer. The half-width at half-maximum is 0.28 nm, indicating that a wavelength shift of 0.28 nm reduces the degree of spectral correlation by half. **b** Absolute value of the averaged cross-correlation between one specific sampling channel and all the other sampling channels. **c** Pearson correlation coefficients (P-coefficients) between the transmission spectra of each pair of sampling channels. **d** Histogram of the P-coefficient, with over 96% falling within the range of $[-0.1, 0.1]$, suggesting a nearly zero linear independence.

The above results and discussions have been added as **the second paragraph of the Spectrum reconstruction part in the Results section and Supplementary Notes 5** in the revised manuscript.

- 2) **The interferograms of the MEMS spectrometer are not simply periodic with varying FSRs.** We reiterate here the model of our interferogram, already included in the original manuscript, serves to provide theoretical reference on how our MEMS spectrometer offers an interferogram different from a simple periodic pattern:

“As illustrated in **Fig. 2e**, the coupling between the cantilever and straight waveguides can be understood by the interference of two supermodes (SM0 and SM1) formed in the waveguide coupler. The vertical coupling gap h , in conjunction with the wavelength λ , defines an effective index difference $\Delta n(\lambda, h)$ between SM0 and SM1, i.e., $\Delta n(\lambda, h) = n_1(\lambda, h) - n_2(\lambda, h)$. Specifically, $\Delta n(\lambda, h)$ can be approximated by a polynomial function:

$$\Delta n(\lambda, h) \approx (a_1 + a_2\lambda + a_3\lambda^2)(b_1 + b_2h + b_3h^2 + b_4h^3) = f_1(\lambda) \cdot f_2(h) \quad (3)$$

This approximation is validated using numerical calculations (see **Supplementary Note 1**). The polynomial approximation can be fitted with a 99.72% R-squared value. h is a function of the applied bias voltage V and can be approximated as:

$$h(V) \approx c_1 + c_2V + c_3V^2 + c_4V^3 \quad (4)$$

We also validate this approximation using numerical calculations (see **Supplementary Note 2**) and achieve a good fitting with a 99.95% R-squared value. By combining **Eqs. 1, 3, and 4**, the output power of the proposed spectrometer can be given as:

$$P_o(\lambda, V) = A(\lambda)\cos^2\left(\frac{\pi L f_1(\lambda) \cdot f_2(h(V))}{\lambda}\right) \quad (5)$$

For our designed waveguide coupler with a certain coupling length L , an interferogram can be obtained at the output port by applying a time-variant bias voltage, using a light beam with a wavelength of λ .”

When we combine **Eqs. 3** and **5** from the main text, the output power is:

$$P_0(\lambda, V) = A(\lambda) \cos^2 \left((a_1 \frac{1}{\lambda} + a_2 + a_3 \lambda) \pi L f_2(h(V)) \right) \quad (\text{R3})$$

For each specific V , the spectra responses can roughly be simulated by $\cos^2 \left(C(a_1 \frac{1}{\lambda} + a_2 + a_3 \lambda) \right)$, where C is a constant value calculated by $\pi L f_2(h(V))$. The function $\cos^2 \left(C(a_1 \frac{1}{\lambda} + a_2 + a_3 \lambda) \right)$ has λ as a variable and a_1, a_2 , and a_3 can be decided by the fitting model.

While it involves the trigonometric function \cos , which is periodic, the parameter $a_1 \frac{1}{\lambda} + a_2 + a_3 \lambda$ does not exhibit periodicity as λ varies - thus, the entire function is not periodic.

In terms of the transmission matrix and self-correlation function, we compare the simulated function $\cos^2 \left(a_1 \frac{1}{\lambda} + a_2 + a_3 \lambda \right)$, where a_1, a_2 and a_3 are taken from the fitting model, to the periodic function $\cos^2(C\lambda)$ as an intuitive visualization in **Fig. R7**. As evident, the pattern exhibited by our simulated model significantly contrasts with that of a periodic pattern, showcasing **an exponential convergence** as opposed to the linear convergence observed in self-correlation.

Besides, our interferogram function (**Eq. R3**), compared to traditional Michelson spectrometers (typically modelled as $\cos^2 \left(C \frac{1}{\lambda} \right)$), includes the higher order terms - this is because we have specially designed the waveguide dimensions to optimize the spectrometer's performance (resolution and matrix rank), as shown in **Fig. R8**. As observed, when the waveguide becomes narrower, both the self-correlation width and the cross-correlation decrease thanks to the more significant higher-order terms, resulting in better spectral reconstruction resolution and accuracy. For waveguide widths of 300 nm and lower, the optical mode cannot be well confined in the waveguide core and would lead to high propagation loss, which lowers the SNR and may worsen the reconstruction performance. Therefore, a waveguide width of 350 nm is chosen.”

The above results and discussions have been added to **Supplementary Notes 5** in the revised manuscript as a complimentary addition.

Fig. R7 Interferogram and self-correlation function of a simulated model of **a, b** the MEMS spectrometer and **c, d** a periodic function, revealing an exponential convergence in the self-correlation within our simulated device, as opposed to the linear convergence observed in periodic functions.

Fig. R8 The spectrometer's performance, i.e., **a** self-correlation function and **b** cross-correlation obtained from the transmission sampling matrix, with varying waveguide width. The $\delta\lambda$ for the four waveguide widths is 0.28, 0.81, 1.76, and 3.11 nm, respectively. The mean cross-correlation value for the four waveguide widths is 0.0010, 0.0040, 0.0080, and 0.0136, respectively.

- 3) **Multi-stage structure for enhancing the performance.** Considering that our proposed MEMS spectrometer is just in its nascent stage of development as a physically single-stage and single-channel device (similar to the Michelson interferometer in the early days of on-chip spectrometers), it holds immense potential for the development of multiple physical stages or channels. On that note, some notable works proposing strategies to enhance sampling channel

diversity (e.g., Refs. ^{60,61} in the main text) offer valuable insights for the further development of MEMS spectrometers. These strategies are indeed applicable to the expansion of the single-channel MEMS spectrometer. Here, we discuss through simulations how the self-/cross-correlations and reconstruction resolution of MEMS spectrometers can be improved by employing multi-stage structures.

Inspired by the multi-stage design demonstrated in Ref. ⁶⁰ in the main text, here we investigate a multi-stage structure in improving the performance of the MEMS spectrometer through simulation. We assessed multiple cascaded MEMS spectrometers, examining m stages. We explore every conceivable combination of voltage modulation channels, totalling n^m channels, as shown in **Fig. R9a**. To accommodate computational limitations, we choose $n = 8$ and focus on scenarios where m equals 2 and 3. The computed self- and cross-correlations are compared to those of the single-stage device (the one in the manuscript with 64 channels), as depicted in **Fig. R9b, c**. As the number of stages and sampling channels increases, the decreased self-correlation width $\delta\lambda$ indicates improved spectral reconstruction resolution. The improvement is also reflected in the minimum value of the self-correlation function, which represents a measure of anti-correlation and is preferred to be close to zero. Additionally, the resolved resolution is expected to significantly exceed the value of $\delta\lambda$ because of the exponentially increased number of sampling channels, as suggested by Ref. ⁶⁰ in the main text. To verify this, we further conduct the dual-wavelength spectra reconstruction using the calibration matrix of the 3-stage spectrometer (shown in **Fig. R9d**). **Figure R9e** plots the transmission spectra of several representative sampling channels, illustrating a high degree of spectral randomness and sufficient decorrelation between sampling channels. The dual-wavelength spectra reconstruction results as depicted in **Fig. R9f** show a high resolution of 15 pm. The results demonstrate the benefits of employing a multi-stage structure, with further enhancements achievable through an increase in channels. This implies large room for further in-depth research about specific network/cascaded structures for enhancing MEMS spectrometers, which will raise broad research interest and thus lead to further advancements.

Fig. R9 Performance of multi-stage MEMS spectrometer. **a** Schematic illustration of the multi-stage MEMS spectrometer and voltage modulation channels. **b** Self- and **c** Cross-correlations varying with the stage number. **d** Calibration matrix of the 3-stage spectrometer. **e** Several example channel spectral responses, featuring significant spectral differences. **f** Dual-wavelength spectra reconstruction, showing a resolution of 15 pm.

The above results and discussions have been added to **the second paragraph of the Discussion and Conclusion section** and **Supplementary Note 14** in the revised manuscript.

In the work presented by Ye et al., the ratio of wavelength number to sampling channel number equals 1601/100, much lower than the 3001/64 of ours, which makes the reconstruction easier, thus

conventional reconstruction algorithms can show certain noise tolerance. Nonetheless, the authors also show that the noise resilience of conventional reconstruction algorithms is limited, thus they employed a new algorithm named gradient projection for sparse reconstruction (GRSR) algorithm to improve the reconstruction performance under noise. Moreover, this is only a simulation work and the demonstrated resolution is only 11 nm, showing the algorithm can be further improved. Nevertheless, we agree that well-designed channel spectral responses with low correlation (for example, by leveraging a multi-stage structure as shown above) can also improve noise tolerance, which can be combined with advanced algorithms such as the CAED of our work for even better reconstruction performance.

References:

60. Yao, C. *et al.* Integrated reconstructive spectrometer with programmable photonic circuits. *Nat. Commun.* **14**, 6376 (2023).
61. Yao, C. *et al.* Broadband picometer-scale resolution on-chip spectrometer with reconfigurable photonics. *Light Sci. Appl.* **12**, 156 (2023).
- R9. Li, A. & Fainman, Y. On-chip spectrometers using stratified waveguide filters. *Nat. Commun.* **12**, 2704 (2021).

Comment 5: The authors must demonstrate the reconstruction of a diverse range of incident spectra for a comprehensive validation of the spectrometer's versatility. However, given the FSR limitations of the interferograms and the necessity to tailor the training of the autoencoder for various input waveforms, I am skeptical about the MEMS spectrometer's ability to deliver satisfactory performance across different complex incident spectra.

Answer 5.1: Testing diverse incident spectra. Thank you for the expert comment and suggestion. We totally agree that the reconstruction of a diverse range of incident spectra is necessary for a comprehensive validation of the spectrometer's versatility. Therefore, beyond the dual-wavelength experiment, a more challenging triple-wavelength testing is conducted. The result presented in **Fig. R10a** illustrates the successful reconstruction of three laser peaks and a spectral spacing of 0.4 nm between the two nearest peaks with a relative error of 0.136. In addition, the reconstruction of a broadband spectrum is demonstrated using an amplified spontaneous emission (ASE) source as the input. During the reconstruction of the broadband spectrum, the β parameter of **Eq. 8** in the main text is optimized via cross-validation analysis, while the α parameter is held constant. As shown in **Fig. R10b**, the spectral features are well recovered with a low relative error ε of 0.041. Furthermore, a mixed spectrum is examined, which combines a broadband signal (the ASE source) with a narrowband signal (a laser source) via a 50/50 optical coupler. The regularization regression equation **Eq. 8** in the main text is accordingly modified by introducing segmented regularization terms, as:

$$\min_{\mathbf{R}} \{ \|\mathbf{I} - \mathbf{P} \cdot (\mathbf{R}_1 + \mathbf{R}_2)\|_2^2 + \alpha \|\mathbf{R}_1 + \mathbf{R}_2\|_1 + \beta \|\mathbf{R}_2\|_2^2 \} \quad (\text{R4})$$

where \mathbf{R}_1 and \mathbf{R}_2 denote the narrowband and broadband spectral components, respectively. The optimal values of α and β are determined via cross-validation analysis^{R10}. **Figure R10c** presents the resolved mixed spectrum with an ε of 0.106, showing that a high reconstruction accuracy can still be achieved. In order to show the efficacy of our spectrometer in real-life applications, we further

demonstrate the reconstruction of the absorption spectrum of *N*-methylaniline, which possesses a well-defined absorption fingerprint near 1.5 μm ^{R11}. Considering the limitation in the experimental setup, we mimic the input absorption spectrum of *N*-methylaniline by tailoring the tunable laser intensity at each wavelength. As depicted in **Fig. R10d**, the absorption fingerprint of *N*-methylaniline can be successfully reconstructed with a low ϵ of 0.055.

Fig. R10 Reconstruction results of a diverse range of incident spectra. **a** Triple-wavelength spectrum. **b** Broadband spectrum. **c** Mixed broadband/narrowband spectrum. **d** Absorption spectrum of *N*-methylaniline.

The above new spectrum reconstruction results and discussions are added to **the last paragraph of the Results section** and **Fig. 5h-k** in the revised manuscript.

Answer 5.2: Training of autoencoder to make it applicable to diverse input spectra. We agree that the denoising may deteriorate when the measured spectrum deviates from the training data. To address the problem, we employ a modified autoencoder structure: Instead of reconstructing the input pattern, we reconstruct the noise pattern and then subtract it from the initial input data. We showcase the capability of our noise-oriented autoencoder scheme to adapt to unseen input waveforms without the need for retraining for each specific kind of pattern through the testing of multi-peak and broadband spectra. Related descriptions and results are added to **the Convolutional autoencoder denoising part of the Results section** in the main text and **Supplementary Note 9**.

In the main text: Denoising autoencoder aims to learn a representation robust to noise added to the original data. Typically, training a denoising autoencoder aims to reconstruct the original data with minimal error. However, if the original data is complicated, the training process may be time-consuming and may lead to underfitting. Additionally, if the autoencoder is overly specialized for a certain type of input, it may lose generalizability to other patterns, necessitating different models for different input spectra. Hereby, we employ a different, noise-oriented training strategy: instead

of training the autoencoder to recover the input pattern, we recover the noise pattern and then subtract it from the initial input data (see details in **Supplementary Note 9**)^{R12}. To be specific, consider a noisy observation \mathbf{I} , which consists of the original data \mathbf{I}_a and the noise \mathbf{e} , i.e., $\mathbf{I} = \mathbf{I}_a + \mathbf{e}$. Since \mathbf{e} is simpler and has a more consistent pattern, we train the autoencoder by learning \mathbf{e} and subtracting it from \mathbf{I} , which is more effective than learning \mathbf{I}_a directly. The schematics of the training and testing phases are depicted in **Fig. 4a**. The parameters of the autoencoder (i.e., encoder f_θ and decoder $g_{\theta'}$) are optimized as follows:

$$\theta^*, \theta'^* = \arg \min_{\theta, \theta'} \frac{1}{M} \sum_{i=1}^M \mathcal{L} \left(\mathbf{e}^{(i)}, g_{\theta'} \left(f_\theta \left(\mathbf{I}^{(i)} \right) \right) \right) \quad (\text{R5})$$

where \mathcal{L} is a loss function of MSE between two inputs. During training phase, the $\mathbf{e}^{(i)}$ is derived by subtracting the ground truth $\mathbf{I}_a^{(i)}$ from the input sample $\mathbf{I}^{(i)}$. In test phase, we employ the trained autoencoder to predict $\tilde{\mathbf{e}}^{(j)}$ and subtract it from the input sample to derive the regenerated data $\tilde{\mathbf{I}}_a^{(j)}$, which can be represented as follows for all $j \in \{1, \dots, L\}$:

$$\tilde{\mathbf{I}}_a^{(j)} = \mathbf{I}^{(j)} - g_{\theta'^*} \left(f_{\theta^*} \left(\mathbf{I}^{(j)} \right) \right) \quad (\text{R6})$$

In Supplementary Note 9: Denoising autoencoder (DAE) aims to learn a representation robust to noise added to the original data. Typically, training a DAE aims to reconstruct the original data with minimal error. However, if the original has a complicated data pattern, training can be time-consuming and may result in underfitting; additionally, if the DAE is trained too specifically on one type of input spectrum, it may lose generalizability to other patterns of input spectra – raising the concern that different input patterns would require different autoencoder settings. Here, we improve the efficiency and performance of DAE by modifying its structure. Instead of reconstructing the input pattern, we reconstruct the noise pattern and then subtract it from the initial input data^{R12}.

Consider a noisy observation \mathbf{I} , which consists of the original data \mathbf{I}_a (referred to as the signal) and the noise \mathbf{e} , i.e., $\mathbf{I} = \mathbf{I}_a + \mathbf{e}$. The *normal way* to train a DAE on signal aims to capture as much information of \mathbf{I}_a as possible, despite \mathbf{I} being a noisy version of the input. As shown in **Fig. R11a**, the parameters of the signal-based DAE model are optimized by minimizing the average reconstruction error in the training phase as follows:

$$\theta_s^*, \theta_s'^* = \arg \min_{\theta, \theta'} \frac{1}{M} \sum_{i=1}^M \mathcal{L} \left(\mathbf{I}_a^{(i)}, g_{\theta'} \left(f_\theta \left(\mathbf{I}^{(i)} \right) \right) \right) \quad (\text{R7})$$

where \mathcal{L} is a loss function of MSE between two inputs. Then, the j -th regenerated data $\tilde{\mathbf{I}}_{a,s}^{(j)}$ from $\mathbf{y}^{(j)}$ in the test phase can be obtained as follows for all $j \in \{1, \dots, L\}$:

$$\tilde{\mathbf{I}}_{a,s}^{(j)} = g_{\theta_s'^*} \left(f_{\theta_s^*} \left(\mathbf{I}^{(j)} \right) \right) \quad (\text{R8})$$

This can be understood as an attempt to maximize the lower bound on mutual information $M(\mathbf{I}_a; \mathbf{I})$, or as an attempt to find a manifold where \mathbf{I} represents the data into a low dimensional latent space corresponding to \mathbf{I}_a . It may face the problem that the stochastic feature of \mathbf{I}_a to be restored is too complex to regenerate or generalize. Especially in the case of spectrometer, this problem can become evident: if the autoencoder is trained to fit certain input patterns very well, it may have poor generalizability to other input spectra.

In this case, we observe that \mathbf{e} is simpler to regenerate than \mathbf{I}_a . \mathbf{e} is not affected by the overall shape of the input spectrum; it is determined by the value of each data point or its limited neighbour data points in the spectrum. Thus, we naturally come up with the intuition that training

the DAE by learning \mathbf{e} and subtracting it from \mathbf{I} , which should be more effective than learning \mathbf{I}_a directly. The training and testing phases of training on noise are depicted in **Fig. R11b**. The parameters of the noise-based DAE can be optimized as follows:

$$\theta_n^*, \theta_n'^* = \arg \min_{\theta, \theta'} \frac{1}{M} \sum_{i=1}^M \mathcal{L} \left(\mathbf{e}^{(i)}, g_{\theta'} \left(f_{\theta}(\mathbf{I}^{(i)}) \right) \right) \quad (\text{R9})$$

Notice that the only difference from (1) is that $\mathbf{I}_a^{(i)}$ is replaced by $\mathbf{e}^{(i)}$. During training phase, the $\mathbf{e}^{(i)}$ is derived by subtracting the ground truth from the input sample $\mathbf{I}^{(i)}$. In testing phase, the ground truth is no longer needed by employing the trained DAE to predict $\mathbf{e}^{(i)}$. Let $\tilde{\mathbf{I}}_{a,n}^{(j)}$ denote the j -th regenerated data, which can be represented as follows for all $j \in \{1, \dots, L\}$:

$$\tilde{\mathbf{I}}_{a,n}^{(j)} = \mathbf{I}^{(j)} - g_{\theta_n'^*} \left(f_{\theta_n^*}(\mathbf{I}^{(j)}) \right) \quad (\text{R10})$$

To verify that the noise-based DAE improves the efficiency of denoising interferograms, we conducted verification through two aspects:

- 1) When the DAE is trained on a certain type of input spectrum, the noise-based DAE exhibits stronger generalizability than the signal-based DAE. A training set of 600,000 samples of two-peak interferograms with 30 dB SNR is used. As shown in **Fig. R11c**, the MSE error between each pair of ground truth and predicted output is depicted. The blue colour represents the signal-based results (MSE between the pair $(\tilde{\mathbf{I}}_{a,s}^{(j)}, \mathbf{I}_a^{(j)})$). The purple colour represents the noise-based results (MSE between the pair $(\tilde{\mathbf{I}}_{a,n}^{(j)}, \mathbf{I}_a^{(j)})$). Since the model is trained specifically on two-peak data, both DAEs show great performance and are very close. However, when the model is extended to unseen data in the training set, such as three-peak and four-peak data, although both models show generalizability, the noise-based model is superior, with a lower level of MSE. The reconstructed interferograms and the reconstructed spectra are shown in **Fig. R11d**, in which the distinct areas where the noise-based DAE predicts more accurately are circled in red. Cases are depicted where the signal-based DAE fails to reconstruct the spectrum, while the noise-based DAE successfully reconstructs the spectra thanks to the improved accuracy in predicting the input. The model is further extended to various noise levels, and we observe that the performance superiority and generalizability of the noise-based model persist, as shown in **Fig. R11e**.
- 2) By training on a mixture of various input spectrum types, we avoid overfitting to a certain type (like the two-peak in aspect 1), and significantly reducing the required data volume. A mixed dataset of two-peak, three-peak, and four-peak data is used, with 10,000 samples for each type (representing a 10 \times decrease in data volume and training time). Since the model now will not be overfit to a certain type of input spectrum, but rather focus on learning the noise pattern, the MSE is further decreased. As shown in **Fig. R11f**, compared to training on two-peak data only, the MSE error is decreased to a level below 0.001 for all data included in the training set. This method can be extended to other input spectra with high generalizability. For example, on a testing five-peak dataset, the MSE is 0.0009, and on a testing broadband spectrum, the MSE is 0.0014, with almost precisely reconstructed input data (**Fig. R11g**, the reconstructed spectra are also shown). A more diverse dataset is always preferred, not because we need to see the input spectrum shape to reconstruct it, but to prevent the network from overfitting to one type of input

data. **Fig. R11h** shows the generalizability to multiple noise levels. If the SNR level difference exceeds 10 dB, generalizability will decrease. However, in practical situations, it is acceptable to test the approximate noise level and select the appropriate model. With these two demonstrations, we showcase the capability of our noise-based autoencoder scheme to adapt to unseen input waveforms without the need for retraining for each specific kind of pattern.

Fig. R11 Noise-oriented training of the denoising autoencoder. **a** Architecture of signal-oriented training/testing. **b** Architecture of noise-oriented training/testing. **c** MSE comparison between signal- and noise-oriented training schemes when training on two-peak data only, and extending to three-peak and four-peak data to evaluate generalizability. **d** Predicted output of the denoising autoencoder and the reconstructed spectrum. **e** Generalizability across multiple SNR levels. **f** MSE comparison of training on one type of data only versus training on a mixed dataset. A mixed dataset with diverse input patterns is preferred as it prevents the model from overfitting to a single input signal pattern and encourages learning the noise pattern. **g** Predicted output of the denoising autoencoder and the reconstructed spectrum, which shows great generalizability to unseen data in the training set, including five-peak and broadband datasets, with an MSE level of 0.001, allowing for nearly lossless reconstruction of the input spectrum. **h** Generalizability to multiple SNR levels.

References:

- R10. Qiao, Q. *et al.* MEMS-Enabled On-Chip Computational Mid-Infrared Spectrometer Using Silicon Photonics. *ACS Photonics* **9**, 2367–2377 (2022).
- R11. Hu, J. *et al.* Fabrication and testing of planar chalcogenide waveguide integrated microfluidic sensor. *Opt. Express* **15**, 2307–2314 (2007).
- R12. Lee, W. H., Ozger, M., Challita, U. & Sung, K. W. Noise Learning-Based Denoising Autoencoder. *IEEE Commun. Lett.* **25**, 2983–2987 (2021).

To Reviewer #2:

Comment 1: This paper combines a couple of interesting techniques like MEMS and convolutional autoencoder denoising mechanism to boost the performance of silicon computational spectrometer. The performance, particularly the spectral resolution is demonstrated to be enhanced at noisy environment. Given the complex working scenarios of integrated spectrometers, improving its robustness against noise is vital. Therefore, I recommend the acceptance when addressing following issues:

Answer 1: Thank you very much for your recognition of the achievement of our work and constructive suggestions. We have revised the manuscript based on your valuable suggestions, and a detailed point-by-point response is provided below.

Comment 2: I agree with the value of using MEMS in terms of modulation efficiency. However, for the compactness, the device shown in Fig.2a has a length of almost 7mm. While other demonstrated silicon computational spectrometers seem to be much more compact. I suggest to modify this argument.

Answer 2: Thank you for the comment. The device shown in Fig. 2a actually has a length of ~2 mm rather than 7 mm, as illustrated by the scale bar. We have also mentioned in **the first paragraph of the Results section** that the coupling length is 2030 μm .

Nonetheless, we have summarized the performance of reported on-chip spectrometers in **Table R2**, which is added as **Table S2 in Supplementary Note 13** in the revised manuscript, and we agree that our device is not that compact if compared with some of the reported works.

Table R2. Performance comparison of reported on-chip spectrometers.

Ref. no. in the main text	Type	Scheme	Resolution (nm)	Bandwidth (nm)	Footprint (μm^2)	Voltage (V)	Power consumption (mW)
13	DO	Disordered structure	3.4	40	50×200	/	/
14	DO	Disordered structure	0.25	30	12.8×30	/	/
30	DO	Disordered structure	0.75	25	50×100	/	/
54	DO	PhC reflector	2.5	55	480×800	/	/
55	DO	Branched waveguide	3	100	12×63	/	/
56	DO	Echelle grating	1.2	30	1100×1420	/	/
15	NF	FP cavity array	0.51	102.7	43×600	8.5	873
19	NF	Two	0.04	100	60×60	NM	85

		coupled MRRs					
57	NF	Euler MRR + cascaded MRR array	0.005	10	3.5×10^5	NM	504
58	NF	MRR array	0.6	60	1×10^6	/	/
59	NF	MRR + random gratings	0.2	60	215×310	NM	33.23
10	FT	Single MZI	3.05	56.18	1×10^6	200	5100
11	FT	SWIFTS	4	96	22×512	/	/
12	FT	tFTS + SHS	0.125	200	5500×6000	135	2400
42	FT	Cascaded DCs	4	300	NM	/	/
52	FT	MRR + MZI	0.47	90	NM	NM	1835
53	FT	MZI with embedded switches	0.2	20	630×2820	NM	99
7	CR	Stratified waveguide filter array	0.45	180	35×260	/	/
16	CR	PhC nanobeam cavity array	5	70	6×111	/	/
18	CR	Cascaded PhC nanobeam cavities	0.32	16	18×250	NM	90
60	CR	Cascaded MZIs	0.01	200	1900×3700	NM	2100
61	CR	Distributed MRRs	0.03	115	2000×7600	/	/
This work	CR	Single DC	0.4 (with denoising) 1.2 (without denoising)	300	38×2030	29.9	6.987×10^{-2}

DO: dispersive optics
NF: narrowband filtering
FT: Fourier transform
CR: computational reconstruction
MRR: microring resonator
FP: Fabry-Pérot
tFTS: tunable Fourier-transform spectrometer
SHS: spatial heterodyne spectrometer
PhC: photonic crystal
MZI: Mach-Zehnder interferometer
DC: directional coupler
SWIFTS: stationary-wave integrated Fourier-transform spectrometer
NM: not mentioned

Therefore, we have modified the related argument in both Abstract and Introduction.

The **Abstract** is revised as below:

Silicon photonics enables the construction of chip-scale spectrometers, in which those using a single tunable interferometer provide a **simple** and cost-effective solution. Among various tuning mechanisms, electrostatic MEMS reconfiguration stands out as an ideal candidate, given its **high tuning efficiency and ultra-low power consumption**. Nonetheless, MEMS devices face significant noise challenges arising from their susceptible minuscule components, adversely impacting spectral resolution. Here, we propose a distinct paradigm of spectrometers through synergizing an **easily-fabricated MEMS-reconfigurable low-loss** waveguide coupler on a silicon photonic chip and a convolutional autoencoder denoising (CAED) mechanism. The spectrometer offers a 300 nm bandwidth and a reconstruction resolution of 0.3 nm in a noise-free condition. In a noisy environment with a signal-to-noise ratio (SNR) as low as 30 dB, the reconstruction resolution of the interferograms processed by the CAED exhibits an enhancement from 1.2 to 0.4 nm, approaching the noise-free value. Our technology is envisaged to provide a powerful and cost-effective solution for applications requiring rapid and accurate spectral analysis.

The **second paragraph of Introduction** is revised as below:

The tunability in Si PICs is typically realized by thermo-optic modulation and free carrier injection, both relying on the change of the Si refractive index^{22,23}. However, because of the weak perturbation of the Si refractive index, these methods frequently result in **high power consumption**²⁴. In comparison, microelectromechanical systems (MEMS) attain modulation by spatially displacing photonic components, consequently improving the modulation efficiency and **reducing the power consumption**^{25,26}. **Among a variety of MEMS actuation mechanisms, electrostatic actuation stands out due to its extremely low standby power and reconfiguration energy consumption**²⁷. Therefore, reconfiguration using electrostatic MEMS actuation offers a simple, effective, and energy-efficient approach for the construction of on-chip spectrometers.

The **last paragraph of Introduction** is revised as below:

In this paper, we present a new paradigm of computational spectrometers based on the synergy between electrostatic MEMS modulation and convolutional autoencoder denoising (CAED) mechanism. **The device features a waveguide coupler reconfigured by an integrated MEMS cantilever actuator. Through optimizing the waveguide structure and revealing the relationship**

between tuning displacement and reconstruction performance, the device achieves broadband low loss and simplifies fabrication using high-yield standard silicon photonics foundry processes. To enhance the resolution against noise, a CAED strategy is proposed and employed. The autoencoder is trained on a diverse dataset of chip-collected interferograms, achieving optimal noise reduction with a resolution approaching the noise-free level. Spectrum reconstruction results demonstrate the effectiveness of CAED in mitigating noise effects with the signal-to-noise ratio (SNR) down to 30 dB, resulting in the improvement of the resolution from 1.2 to 0.4 nm. The proposed CAED-facilitated MEMS spectrometer presents a promising solution for high-resolution spectral analysis in applications demanding precision and noise resilience. The utilization of advanced deep learning techniques of denoising autoencoders not only improves the performance of MEMS spectrometers but also presents a universal solution for mitigating noise-related challenges in computational spectrometers with calibration matrices. Our approach offers a potential pathway toward the realization of robust and cost-effective on-chip spectrometers suitable for widespread applications, including portable, handheld, and wearable scenarios.

Comment 3: Another drawback of using MEMS is the extremely high driving voltage. As frequently mentioned, integrated spectrometers are suggested to be used in portable devices and the drivers should be CMOS electronics. How to deliver such high voltage in practical applications?

Answer 3: Thank you for the valuable comment. As can be seen from **Table R2**, the driving voltage of our device is not high if compared with the reported works using thermo-optic tuning (Refs. ^{10,12} in the main text). Nonetheless, we agree that the driving voltage of our device is still out of reach of CMOS electronics which typically provide a driving voltage of several volts. In order to make our device able to be driven by CMOS electronics in practical applications, we can make improvements in two aspects:

- 1) The driving voltage provided by CMOS electronics can be amplified to the required value by a high-voltage amplifier and then delivered to our device. High-voltage amplifiers have been widely adopted to drive various MEMS actuators, including electrostatic parallel-plate actuator¹³, electrostatic comb-drive actuator¹⁴, piezoelectric actuator¹⁵, etc.
- 2) The cantilever can be designed softer to lower the driving voltage to several volts by lengthening the cantilever, thinning the Al layer, and enlarging the releasing holes on the cantilever, as shown in **Fig. R12**. It is worth noting that a trade-off between driving voltage and mechanical robustness as well as response time needs to be made.

Fig. R12 a Top-view schematic illustration of the MEMS cantilever. **b-d** Driving voltage required to achieve $0.7\ \mu\text{m}$ displacement with varying **b** length of the cantilever, **c** thickness of the Al layer, **d** side length of the square releasing holes.

The above results and discussions have been added as **Supplementary Note 16** in the revised manuscript.

References:

10. Souza, M. C. M. M., Grieco, A., Frateschi, N. C. & Fainman, Y. Fourier transform spectrometer on silicon with thermo-optic non-linearity and dispersion correction. *Nat. Commun.* **9**, 665 (2018).
12. Xu, H., Qin, Y., Hu, G. & Tsang, H. K. Scalable integrated two-dimensional Fourier-transform spectrometry. *Nat. Commun.* **15**, 436 (2024).
- R13. Horenstein, M. N. *et al.* Ultra-low-power multiplexed electronic driver for high resolution deformable mirror systems. *Proc. SPIE* **7930**, 79300M (2011).
- R14. Takahashi, K. *et al.* Monolithic integration of high voltage driver circuits and MEMS actuators by ASIC-like postprocess. in *The 13th International Conference on Solid-State Sensors, Actuators and Microsystems (TRANSDUCERS)* vol. 1 417–420 (IEEE, 2005).
- R15. Otieno, L. O. *et al.* A high bandwidth, high voltage amplifier for driving fast piezoelectric actuator-based nanopositioners used in atomic force microscopes. *J. Korean Phys. Soc.* **83**, 795–806 (2023).

Comment 4: The minimum SNR tested is 25dB, which is still quite optimistic and doesn't make too much sense. In real-life applications such as smartwatch based healthcare monitoring, the SNR can be as small as a few dB.

Answer 4: Thank you for the constructive comment. We agree that the SNR can be much lower in real-life applications. Therefore, we have investigated the denoising performance of our CAED mechanism for lower SNR levels of 20, 15, and 8 dB, as shown in **Fig. R13**. Both the reconstruction resolutions with and without denoising worsen with increasing noise level (i.e., decreasing SNR). However, for all the SNR levels, the denoising by CAED significantly improves the reconstruction resolution (for 20 dB SNR, from 1.5 to 0.5 nm; for 15 dB SNR, from 1.7 to 0.6 nm; for 8 dB SNR, from 2.1 to 0.8 nm). Additionally, as what have been mentioned in the original manuscript, better denoising results are expected to be achieved if we train the autoencoder using a noise dataset corresponding to these lower SNR values. Therefore, our CAED mechanism would be effective across the whole SNR range in real-life applications.

Fig. R13 Dual-wavelength spectrum reconstruction under lower SNR levels.

The above results and discussions have been added to **the penultimate paragraph of the Results section** and **Supplementary Note 10** in the revised manuscript.

Comment 5: Recently it is reported that computational spectrometer is robust to temperature change "A.Li et al., An integrated single-shot spectrometer with large bandwidth-resolution ratio and wide operation temperature range, *Photonix* 4(1) 2023". Is this also true for this MEMS spectrometer? how would the performance vary at fluctuating temperature?

Answer 5: Thank you for the insightful comment. As stated in the work of Li et al., for computational spectrometers using broadband filters with non-zero transmissions within a wide optical range, the temperature change shifts the transmission spectrum, but for the optical range of interest, the transmissions are still non-zero and the spectral information would not be lost. Temperature change will only generate a new sampling matrix, which could be recorded at the calibration stage, so that the input spectra can still be resolved at different temperatures, simply by

choosing the correct calibration matrix. Therefore, employing a simple and cheap temperature sensor is adequate for computational spectrometers to work in a broad temperature range.

The above statement is also true for our MEMS spectrometer, because the MEMS-tunable directional coupler at each tuning state can be regarded as a broadband filter as well and will not lose spectral information under temperature change. Therefore, input spectra can be successfully reconstructed without performance degradation, as long as the calibration matrix at each temperature is pre-recorded. In order to verify this statement, we have also tested the spectral reconstruction performance of our MEMS spectrometer at fluctuating temperatures in the range of 10-70 °C, which is consistent with the work of Li et al. and covers the reasonable operating temperature range for practical applications. The calibration matrices at these three temperatures are firstly collected and then utilized to reconstruct dual-wavelength spectra at the corresponding temperatures. As shown in **Fig. R14**, both the MEMS spectrometer and the CAED mechanism work well at all these three temperatures, thus maintaining a denoised resolution of 0.4 nm across the whole operating temperature range.

Fig. R14 Dual-wavelength spectrum reconstruction at different temperatures.

The above results and discussions have been added to **the penultimate paragraph of the Results section and Supplementary Note 12** in the revised manuscript.

Comment 6: Besides resolution, the reconstruction accuracy should also be vulnerable to noise, but from the experimental results, it seems that the reconstructions are always clean. I suggest to go with more complex spectra and more data points in the reconstructions.

Answer 6: Thank you for the constructive suggestion. Following your suggestion, we have performed the reconstruction of more complex spectra. In order to quantify the reconstruction

accuracy, here we utilize the widely adopted metric named relative error ε , which is defined as below^{R16}:

$$\varepsilon = \frac{\|\mathbf{R} - \hat{\mathbf{R}}\|_2}{\|\mathbf{R}\|_2} \quad (\text{R11})$$

where \mathbf{R} and $\hat{\mathbf{R}}$ are the input and reconstructed spectrum, respectively. Beyond the dual-wavelength experiment, a more challenging triple-wavelength testing is conducted. The result presented in **Fig. R15a** illustrates the successful reconstruction of three laser peaks and a spectral spacing of 0.4 nm between the two nearest peaks with a relative error of 0.136. In addition, the reconstruction of a broadband spectrum is demonstrated using an amplified spontaneous emission (ASE) source as the input. During the reconstruction of the broadband spectrum, the β parameter of **Eq. 8** in the main text is optimized via cross-validation analysis, while the α parameter is held constant. As shown in **Fig. R15b**, the spectral features are well recovered with a low relative error ε of 0.041. Furthermore, a mixed spectrum is examined, which combines a broadband signal (the ASE source) with a narrowband signal (a laser source) via a 50/50 optical coupler. The regularization regression equation **Eq. 8** in the main text is accordingly modified by introducing segmented regularization terms, as:

$$\min_{\mathbf{R}} \{ \|\mathbf{I} - \mathbf{P} \cdot (\mathbf{R}_1 + \mathbf{R}_2)\|_2^2 + \alpha \|\mathbf{R}_1 + \mathbf{R}_2\|_1 + \beta \|\mathbf{R}_2\|_2^2 \} \quad (\text{R12})$$

where \mathbf{R}_1 and \mathbf{R}_2 denote the narrowband and broadband spectral components, respectively. The optimal values of α and β are determined via cross-validation analysis^{R10}. **Figure R10c** presents the resolved mixed spectrum with an ε of 0.106, showing that a high reconstruction accuracy can still be achieved. In order to show the efficacy of our spectrometer in real-life applications, we further demonstrate the reconstruction of the absorption spectrum of *N*-methylaniline, which possesses a well-defined absorption fingerprint near 1.5 μm ^{R11}. Considering the limitation in the experimental setup, we mimic the input absorption spectrum of *N*-methylaniline by tailoring the tunable laser intensity at each wavelength. As depicted in **Fig. R10d**, the absorption fingerprint of *N*-methylaniline can be successfully reconstructed with a low ε of 0.055.

Fig. R15 Reconstruction results of a diverse range of incident spectra. **a** Triple-wavelength spectrum. **b** Broadband spectrum. **c** Mixed broadband/narrowband spectrum. **d** Absorption spectrum of *N*-methylaniline.

The above new spectrum reconstruction results and discussions are added to **the last paragraph of the Results section** and **Fig. 5h-k** in the revised manuscript.

References:

- R10. Qiao, Q. *et al.* MEMS-Enabled On-Chip Computational Mid-Infrared Spectrometer Using Silicon Photonics. *ACS Photonics* **9**, 2367–2377 (2022).
- R11. Hu, J. *et al.* Fabrication and testing of planar chalcogenide waveguide integrated microfluidic sensor. *Opt. Express* **15**, 2307–2314 (2007).
- R16. Xu, H., Qin, Y., Hu, G. & Tsang, H. K. Breaking the resolution-bandwidth limit of chip-scale spectrometry by harnessing a dispersion-engineered photonic molecule. *Light Sci. Appl.* **12**, 64 (2023).

Comment 7: Some other information regarding the MEMS device should be provided, like the total insertion loss, the optical bandwidth, modulation speed etc. Particularly I am interested in the fiber/chip coupler, as it can support a wide range of wavelengths from 1.3μm to 1.6μm. A static transmission spectrum of the device is suggested to be included.

Answer 7: Thank you for the suggestion. In our work, adiabatically tapered edge couplers are used to facilitate broadband fiber-to-chip coupling. The edge coupler has a tip width of 200 nm and a taper length of 180 μm. As shown in **Fig. R16**, our employed edge coupler enables efficient fiber-to-chip coupling with coupling losses of below 4 dB/facet in the wavelength range of 1.3-1.6 μm. The edge coupler shows a 1-dB bandwidth of ~250 nm.

Fig. R16 Transmission spectrum of edge coupler.

Fig. R17 shows the static transmission spectrum of the whole device at the initial state. Our device shows a total insertion loss of 6.4-10.6 dB across the whole wavelength range of 1.3-1.6 μm . For the collection of the calibration matrix, wavelength-dependent features of laser intensity, edge coupler efficiency, and detector responsivity are cancelled out by taking the transmission spectrum of a straight waveguide on the same chip as a reference.

Fig. R17 Static transmission spectrum of the whole device at the initial state.

The modulation speed of the MEMS actuator is characterized by measuring its frequency response using a laser Doppler vibrometer, as shown in **Fig. R18**. The measured mechanical resonance frequency of 177 kHz agrees with the simulated response time of $\sim 12 \mu\text{s}$.

Fig. R18 Measured frequency response of the MEMS actuator.

The above results and discussions have been added to the **third paragraph of the MEMS spectrometer part of the Results section** and **Supplementary Notes 3&4** in the revised manuscript.

To Reviewer #3:

Comment 1:

Key Results

The authors demonstrate a chip-scale spectrometer with a bandwidth of 300 nm and resolution of 0.3 nm. They claim to overcome the challenges of chip-scale spectrometers, namely poor resolution, by leveraging a time-domain MEMS controlled system with neural network processing for enhancing resolution. The authors rely on MEMS actuation of active optical elements to achieve spectral separation. The spectral resolution claimed is impressive, especially considering the proposed bandwidth. This resolution matches some more conventional spectrometers, by leveraging computational reconstruction of spectra using a neural network.

Recommendation

Overall, I believe this manuscript is well-written and presents a significant improvement in the on-chip spectrometer space. After some proposed revisions, this manuscript should be ready for publication.

Answer 1: Thank you very much for your recognition of the achievement of our work and constructive suggestions. We have revised the manuscript based on your valuable suggestions, and a detailed point-by-point response is provided below.

Questions and Comments

Comment 2: Based on the current form of the manuscript, the key novelty here seems to be on the analysis of the data using the deep learning CAED method. However, innovation on the photonics side must be inferred. Can the authors state the innovation in their design more clearly in the abstract/introduction because currently it appears that there is no major innovation in the physical on-chip design?

Answer 2: Thank you for the constructive suggestion. We agree that the innovation in the physical on-chip design needs to be stated more clearly. In our previous work^{R10}, we have reported a MEMS spectrometer that leverages the time-domain modulation of reconfigurable waveguide couplers to generate various interferograms for spectrum reconstruction. Here in this work, we have further improved the physical design from five aspects as below:

- 1) In the previous work, it was thought that the directional coupler needs to be modulated from the initial coupled condition to a fully decoupled condition to facilitate the spectral reconstruction. Such a coupling condition transition requires a large tuning displacement of the MEMS cantilever, which exceeds the tuning range that can be offered by the release of the BOX layer, with the pull-in effect taken into account^{R1}. As a result, a non-standard and low-yield flip-chip bonding process was needed to enable the large tuning displacement. Here in this work, we have firstly investigated the relationship between the device tuning range and the spectral reconstruction performance using correlation analysis. The self- and cross-correlations of three tuning ranges (from the initial coupled condition to a weakly/moderately/fully decoupled condition) are studied. As shown in **Fig. R19a, b**, the moderately decoupled condition results

in the smallest self-correlation width as well as considerably low cross-correlation. The fully decoupled condition can further decrease the cross-correlation, but at the expense of larger self-correlation width, thus tends not to improve the spectral resolution. The dual-wavelength spectra reconstruction results as shown in **Fig. R19c-e** confirm that the moderately decoupled condition offers the finest reconstruction resolution. The adoption of a moderately decoupled condition in our work not only guarantees a satisfactory spectral resolution but also reduces the required tuning range to a value that can be offered by the release of the BOX layer, which largely simplifies the device configuration and fabrication process.

Fig. R19 **a** Self-correlation function and **b** Cross-correlation of three tuning ranges from the initial coupled condition to a weakly/moderately/fully decoupled condition. **c-e** Spectrum reconstruction results at the weakly/moderately/fully decoupled condition, respectively. The moderately decoupled condition exhibits the finest reconstruction resolution.

- 2) In the previous work, subwavelength grating (SWG) was employed as the supporting structure for the suspended waveguides in the mid-infrared. However, when migrating the SWG design to the telecommunication wavelengths, the minimum feature size is typically below 150 nm, which is challenging for common silicon photonics foundries^{R2,3}. In this work, we employ and optimize a trapezoidal supporting structure for low-loss suspension of the waveguides, as shown in **Fig. R20**. Compared to SWG, the trapezoidal supporting structure possesses lower fabrication restriction and better wavelength scalability. We fix the beam width to be 0.4 μm , and then optimize the structural parameters of the trapezoid. With a height, baseline length, and topline length of 0.4, 8, and 3 μm , respectively, the insertion loss of the trapezoidal supporting structure is optimized to 0.098 dB, which is comparable with the reported elliptical supporting structure^{R4}. The minimum feature size of 0.4 μm can be conveniently fabricated by common silicon photonics foundries.

Fig. R20 Design and optimization of trapezoidal supporting structure. **a** Schematic illustration of structural parameters. **b** Optimization of the trapezoid height. **c** Optimization of the trapezoid baseline length. **d** Optimization of the trapezoid topline length, inset shows the electric field profile with the optimized structural parameters.

- 3) Unlike the normal directional coupler employed in the previous work, here we simplify the bus waveguide to a straight waveguide, which reduces the propagation loss and improves the signal-to-noise ratio.
- 4) Unlike the grating couplers employed in the previous work, adiabatically tapered edge couplers are used to facilitate the fiber-to-chip coupling in our work, which significantly enlarge the device bandwidth. **Fig. R21** shows the transmission spectrum of the edge coupler, showing a low coupling loss of below 4 dB/facet across the wavelength range of 1.3-1.6 μm .

Fig. R21 Transmission spectrum of edge coupler.

- 5) The residual stress in the Si device layer will induce significant unevenness of the cantilever and deteriorate the coupling between the two waveguides, limiting the achievable coupling

length. Thus, the long coupling length was offered by two cascaded waveguide couplers in our previous work. Here in this work, we deposit an Al layer with proper length and thickness onto the Si cantilever to improve the stiffness and thus the flatness of the cantilever^{R5}. Consequently, we are able to implement a long coupling length using a single waveguide coupler. The good flatness of the cantilever ensures uniform coupling along the whole coupling length, which is confirmed by the measured calibration matrix.

The above results and discussions have been added as **the third paragraph of the MEMS spectrometer part in the Results section and Supplementary Notes 3** in the revised manuscript.

The Abstract and Introduction have also been revised accordingly to state the innovation in the physical on-chip design more clearly.

The **Abstract** is revised as below:

Silicon photonics enables the construction of chip-scale spectrometers, in which those using a single tunable interferometer provide a **simple** and cost-effective solution. Among various tuning mechanisms, electrostatic MEMS reconfiguration stands out as an ideal candidate, given its **high tuning efficiency and ultra-low power consumption**. Nonetheless, MEMS devices face significant noise challenges arising from their susceptible minuscule components, adversely impacting spectral resolution. Here, we propose a distinct paradigm of spectrometers through synergizing **an easily-fabricated MEMS-reconfigurable low-loss** waveguide coupler on a silicon photonic chip and a convolutional autoencoder denoising (CAED) mechanism. The spectrometer offers a 300 nm bandwidth and a reconstruction resolution of 0.3 nm in a noise-free condition. In a noisy environment with a signal-to-noise ratio (SNR) as low as 30 dB, the reconstruction resolution of the interferograms processed by the CAED exhibits an enhancement from 1.2 to 0.4 nm, approaching the noise-free value. Our technology is envisaged to provide a powerful and cost-effective solution for applications requiring rapid and accurate spectral analysis.

The **second paragraph of Introduction** is revised as below:

The tunability in Si PICs is typically realized by thermo-optic modulation and free carrier injection, both relying on the change of the Si refractive index^{22, 23}. However, because of the weak perturbation of the Si refractive index, these methods frequently result in **high power consumption**²⁴. In comparison, microelectromechanical systems (MEMS) attain modulation by spatially displacing photonic components, consequently improving the modulation efficiency and **reducing the power consumption**^{25, 26}. Among a variety of MEMS actuation mechanisms, **electrostatic actuation stands out due to its extremely low standby power and reconfiguration energy consumption**²⁷. Therefore, reconfiguration using electrostatic MEMS actuation offers a simple, effective, and energy-efficient approach for the construction of on-chip spectrometers.

The **last paragraph of Introduction** is revised as below:

In this paper, we present a new paradigm of computational spectrometers based on the synergy between electrostatic MEMS modulation and convolutional autoencoder denoising (CAED) mechanism. **The device features a waveguide coupler reconfigured by an integrated MEMS cantilever actuator. Through optimizing the waveguide structure and revealing the relationship between tuning displacement and reconstruction performance, the device achieves broadband low loss and simplifies fabrication using high-yield standard silicon photonics foundry processes. To enhance the resolution against noise, a CAED strategy is proposed and employed. The autoencoder**

is trained on a diverse dataset of chip-collected interferograms, achieving optimal noise reduction with a resolution approaching the noise-free level. Spectrum reconstruction results demonstrate the effectiveness of CAED in mitigating noise effects with the signal-to-noise ratio (SNR) down to 30 dB, resulting in the improvement of the resolution from 1.2 to 0.4 nm. The proposed CAED-facilitated MEMS spectrometer presents a promising solution for high-resolution spectral analysis in applications demanding precision and noise resilience. The utilization of advanced deep learning techniques of denoising autoencoders not only improves the performance of MEMS spectrometers but also presents a universal solution for mitigating noise-related challenges in computational spectrometers with calibration matrices. Our approach offers a potential pathway toward the realization of **robust and cost-effective** on-chip spectrometers suitable for widespread applications, including portable, handheld, and wearable scenarios.

References:

- R1. O'Brien, G., Monk, D. J. & Lin, L. MEMS cantilever beam electrostatic pull-in model. *Proc. SPIE* **4593**, 31–41 (2001).
- R2. Cheben, P., Halir, R., Schmid, J. H., Atwater, H. A. & Smith, D. R. Subwavelength integrated photonics. *Nature* **560**, 565–572 (2018).
- R3. Chen, L. R., Member, S. & Wang, J. Subwavelength Grating Waveguide Devices for Telecommunications Applications. *IEEE J. Sel. Top. Quantum Electron.* **25**, 8200111 (2019).
- R4. Fukazawa, T., Hirano, T., Ohno, F. & Baba, T. Low Loss Intersection of Si Photonic Wire Waveguides. *Jpn. J. Appl. Phys.* **43**, 646–647 (2004).
- R5. Gyger, S. *et al.* Reconfigurable photonics with on-chip single-photon detectors. *Nat. Commun.* **12**, 1408 (2021).
- R10. Qiao, Q. *et al.* MEMS-Enabled On-Chip Computational Mid-Infrared Spectrometer Using Silicon Photonics. *ACS Photonics* **9**, 2367–2377 (2022).

Comment 3: The authors provide good examples of spectrum reconstruction demonstrating the resolution of the device; however, bandwidth is another important parameter for any spectrometer. While the authors claim a bandwidth of 300 nm, there is no direct evidence of this in the figures beyond the spectral peaks at 1.33 μm and 1.57 μm in Figure 3a, which is only ~ 250 nm. Can the authors provide a demonstration of the 300 nm bandwidth of the device? Perhaps, they can use a broadband input source like a mercury halogen lamp?

Answer 3: Thank you for the helpful suggestion. To provide direct evidence of the 300 nm bandwidth, we have also reconstructed single-wavelength spectra at 1.3 and 1.6 μm , as shown in **Fig. R22**. **Figure 2i** in the manuscript is updated accordingly.

Fig. R22 Several reconstructed single-wavelength spectra across the whole 300 nm bandwidth.

In terms of broadband spectrum reconstruction, unfortunately, we do not have a broadband light source that can provide emission across the whole 300 nm wavelength range. Additionally, the output light from a mercury halogen lamp may not be able to be effectively coupled into the waveguide due to the low spatial coherence. Nonetheless, we have added broadband spectrum reconstruction results by using an amplified spontaneous emission (ASE) broadband light source. As shown in **Fig. R23**, the spectral features are well recovered with a low relative error ε of 0.041.

Fig. R23 Reconstruction of ASE spectrum.

Figure R23 has been added as **Fig. 5i** and the corresponding discussion has been added to **the last paragraph of the Results section** in the revised manuscript.

Comment 4: Another interesting demonstration would be analyzing the spectrum of a real sample with features within the bandwidth of this device. The authors speak at length on the applications and merits of on-chip spectroscopy, so for a journal of this caliber and broad readership, a more practical demonstration would add significant value.

Answer 4: Thank you for the helpful suggestion. We totally agree that a more practical demonstration would add significant value to the manuscript. The introduced applications, such as material analysis, medical diagnostics, and environmental monitoring, center around the infrared absorption spectroscopy of various chemical and biological molecules. The bandwidth of 1.3-1.6 μm also covers the absorption fingerprints of numerous molecules. Here, we choose to demonstrate the reconstruction of the absorption spectrum of *N*-methylaniline, which possesses a well-defined absorption fingerprint near 1.5 μm ^{R11}. Due to the limitation in experimental setup and time, we mimic the input absorption spectrum of *N*-methylaniline by tailoring the tunable laser intensity at each wavelength. As depicted in **Fig. R24**, the absorption fingerprint of *N*-methylaniline can be successfully reconstructed with a low relative error ε of 0.055.

Fig. R24 Reconstruction of the absorption spectrum of *N*-methylaniline.

Figure R24 has been added as **Fig. 5k** and the corresponding discussion has been added to **the last paragraph of the Results section** in the revised manuscript.

Reference:

- R11. Hu, J. *et al.* Fabrication and testing of planar chalcogenide waveguide integrated microfluidic sensor. *Opt. Express* **15**, 2307–2314 (2007).

Comment 5: Figure 2. Even though Figures b, c are zoomed versions of a, the axis labels and units should be included for clarity.

Answer 5: Thank you for the suggestion. We suppose that you are asking to add axis labels and units for **Figs. 3b, c** and we have done so in the revised manuscript, also as shown in **Fig. R25b, c** below. In **Fig. 3a** of the revised manuscript and **Fig. R25a** below, we have also changed the wavelength spacing from 200 to 300 nm to provide additional evidence of the 300 nm bandwidth.

Fig. R25 Noise-free spectrum reconstruction and noise analysis. **a** Reconstruction of dual-wavelength spectra with different wavelength spacings under the noise-free condition. **b** Zoom-in view of the input and the reconstructed spectra with 0.2 nm spacing. **c** Zoom-in view of the input and the reconstructed spectra with 0.3 nm spacing.

Comment 6: While the result in Figure 2b,c is impressive and demonstrates the resolution of the system, a similar demonstration without applying denoising would help the reader understand the true contribution of CAED to this work and the technology at large.

Answer 6: Thank you for the suggestion. We truly agree that similar dual-wavelength spectra reconstruction without applying denoising is necessary. Actually, these results have been presented in **Figs. 5d-g**, together with corresponding results with applying denoising, so as to intuitively show the contribution of CAED in the improvement of reconstruction resolution.

Comment 7: What is the efficiency or run time of the algorithm the authors have developed? Most practical applications of spectrometers require moderate to high speed, so an understanding of the processing time will help gauge the efficacy of this approach.

Answer 7: Thank you for the suggestion. The denoising autoencoder is deployed on 4 GPUs (NVIDIA TITAN Xp) and trained with a batch size of 128. The dataset volume is 600,000. Each training epoch takes 20 seconds. Therefore, for a total of 360 training epochs, the time taken is approximately 2 hours. After training, the model can be applied to random input samples, with each sample taking 0.03 milliseconds for prediction.

In our optimized training strategy described in **the Convolutional autoencoder denoising part of the Results section and Supplementary Note 9**, the data volume is significantly reduced to 30,000. The time taken for each epoch is approximately 1 second, resulting in a total training time of 6 minutes.

The above discussion has been added to **the Reconstruction implementation part of the Methods section** in the revised manuscript.

Comment 8: Could the authors please add more details to the methods section? Important information such as what specific type/model of tunable laser was used, what kind of detector was used, what was used for power supply and voltage modulation is not included. These specifications will allow readers to more accurately repeat the results presented here.

Answer 8: Thank you for the suggestion. We have revised **the Device characterization part of the Methods section** as below to include more details on the equipment specifications:

Device characterization. For single-, dual- and multi-wavelength characterization, a set of tunable lasers (Santec TSL-510, 550, and 710) are adopted as the input, which are also used to measure the calibration matrix. For broadband characterization, Amonics C + L band ASE broadband light source is used as the input. A polarization controller is used to ensure that only TE-polarized light is injected into the on-chip spectrometer. The spectrometer chip is mounted on an XYZ stage for fiber-chip alignment, with the temperature controlled by a temperature controller. The light is coupled in and out of the chip through two on-chip adiabatically tapered edge couplers for broadband operation. The output light from the MECS is collected by a photodetector (Thorlabs PDA-10CS-EC). Input spectra are also recorded using an optical spectrum analyzer (Yokogawa AQ6370D) as references. A semiconductor characterization system (Keithley 4200-SCS) is employed for time-sequenced bias voltage supply to implement time-domain modulation of the MECS.

Comment 9: Similarly, details of how light is coupled into and out of the system needs to be added to the Methods section.

Answer 9: Thank you for the suggestion. We have revised **the Device characterization part of the Methods section** as to include of how light is coupled into and out of the system. Please kindly check our answer to your Comment 8 above.

To Reviewer #4:

Comment 1: In this work, the authors present a novel MEMS-based spectrometer architecture along with a signal processing technique which suppresses noise and subsequently improves spectral resolution. The authors fabricate an experimental device and perform some experiments to demonstrate that their methods work on the fabricated device. I believe that both the spectrometer design and approach to denoising are significant and will contribute meaningfully to the advancement of on-chip spectrometers. Power consumption of existing thermally-actuated spectrometers and sensitivity to noise are important issues in current technology, and this paper aptly points out these issues and provides effective methods to address them. In particular, the thorough analysis of the relationship between the SNR and the reconstructed resolution is an important analysis which provides a more holistic understanding of the system performance, and is a great inclusion, as is the noise analysis of the MEMS structure in Supplementary Note 4. Finally, the paper is well-written, cites appropriate references, and has clear and concise analysis supporting the MEMS design, spectral extraction, and denoising processes.

Answer 1: Thank you very much for your recognition of the achievement of our work.

Comment 2: Though the foundations of this work are strong, I believe that the experimental testing performed is not adequate to support the claimed resolution, signal-to-noise, and bandwidth specifications of the realized systems. In particular, it appears as if this spectrometer uses 64 measurements (corresponding to 64 different positions of the MEMS actuator) to extract a spectrum with 3001 data points, representing a highly underdetermined system. While numerous previous works have shown that spectra can be reconstructed even in such highly underdetermined cases (e.g., ref. [60] in the manuscript), such systems must be tested on a wide variety of spectral sources to demonstrate that the regularization being used to enable reconstruction in such a highly underdetermined problem is in fact valid for diverse input spectra. For example, ref. [60] demonstrated reconstruction of a spectrum with $> 10,000$ spectral channels from just 729 measurements, but rigorously backed up this result through both theoretical analysis, simulations, and experimental testing of the spectrometer on narrowband, broadband, and mixed sources. This current manuscript, on the other hand, appears to only perform reconstruction on pairs of narrowband sources, and as such its versatility is not demonstrated. Supplementary Note 6 contains some testing showing that the reconstruction accuracy decreases with increasing wavelength spacing of dual-wavelength sources due to how the denoising system was trained, indicating that performance of both the denoising and regression-based reconstruction may deteriorate when the measured spectrum deviates from the training data. The claimed figures for resolution, bandwidth, and signal-to-noise performance can only be fairly compared to other devices in literature (as is done in Fig. 6) if these figures hold true across a wide range of possible spectral measurements.

Answer 2: Thank you very much for your constructive suggestions. We totally agree that our spectrometer needs to be tested on a wide variety of spectral sources, including narrowband, broadband, and mixed sources. Additionally, we agree that both the denoising and regression-based reconstruction may deteriorate when the measured spectrum deviates from the training data. Therefore, we improve the efficiency and performance of the denoising autoencoder with a modification of its structure. Instead of reconstructing the input pattern, we reconstruct the noise

pattern and then subtract it from the initial input data. We showcase the capability of our noise-oriented autoencoder scheme to adapt to unseen input waveforms without the need for retraining for each specific kind of pattern through the testing of multi-peak and broadband spectra.

Related results and discussions have been added to **the first three paragraphs of the Convolutional autoencoder denoising part of the Results section** in the main text and **Supplementary Note 9**.

In the main text: Denoising autoencoder aims to learn a representation robust to noise added to the original data. Typically, training a denoising autoencoder aims to reconstruct the original data with minimal error. However, if the original data is complicated, the training process may be time-consuming and may lead to underfitting. Additionally, if the autoencoder is overly specialized for a certain type of input, it may lose generalizability to other patterns, necessitating different models for different input spectra. Hereby, we employ a different, noise-oriented training strategy: instead of training the autoencoder to recover the input pattern, we recover the noise pattern and then subtract it from the initial input data (see details in **Supplementary Note 9**)^{R12}. To be specific, consider a noisy observation \mathbf{I} , which consists of the original data \mathbf{I}_a and the noise \mathbf{e} , i.e., $\mathbf{I} = \mathbf{I}_a + \mathbf{e}$. Since \mathbf{e} is simpler and has a more consistent pattern, we train the autoencoder by learning \mathbf{e} and subtracting it from \mathbf{I} , which is more effective than learning \mathbf{I}_a directly. The schematics of the training and test phases are depicted in **Fig. 4a**. The parameters of the autoencoder (i.e., encoder f_θ and decoder $g_{\theta'}$) are optimized as follows:

$$\theta^*, \theta'^* = \arg \min_{\theta, \theta'} \frac{1}{M} \sum_{i=1}^M \mathcal{L} \left(\mathbf{e}^{(i)}, g_{\theta'} \left(f_\theta \left(\mathbf{I}^{(i)} \right) \right) \right) \quad (\text{R13})$$

where \mathcal{L} is a loss function of mean square error between two inputs. During training phase, the $\mathbf{e}^{(i)}$ is derived by subtracting the ground truth $\mathbf{I}_a^{(i)}$ from the input sample $\mathbf{I}^{(i)}$. In test phase, we employ the trained autoencoder to predict $\tilde{\mathbf{e}}^{(j)}$ and subtract it from the input sample to derive the regenerated data $\tilde{\mathbf{I}}_a^{(j)}$, which can be represented as follows for all $j \in \{1, \dots, L\}$:

$$\tilde{\mathbf{I}}_a^{(j)} = \mathbf{I}^{(j)} - g_{\theta'^*} \left(f_{\theta^*} \left(\mathbf{I}^{(j)} \right) \right) \quad (\text{R14})$$

In Supplementary Note 9: Denoising autoencoder (DAE) aims to learn a representation robust to noise added to the original data. Typically, training a DAE aims to reconstruct the original data with minimal error. However, if the original has a complicated data pattern, training can be time-consuming and may result in underfitting; additionally, if the DAE is trained too specifically on one type of input spectrum, it may lose generalizability to other patterns of input spectra – raising the concern that different input pattern would require different autoencoder settings. Here, we improve the efficiency and performance of DAE with a modification of its structure. Instead of reconstructing the input pattern, we reconstruct the noise pattern and then subtract it from the initial input data^{R12}.

Consider a noisy observation \mathbf{I} , which consists of the original data \mathbf{I}_a (referred to as the signal) and the noise \mathbf{e} , i.e., $\mathbf{I} = \mathbf{I}_a + \mathbf{e}$. The *normal way* to train a DAE on signal aims to capture as much information of \mathbf{I}_a as possible, despite \mathbf{I} being a noisy version of the input. As shown in **Fig. R26a**, the parameters of the signal-based DAE model are optimized by minimizing the average reconstruction error in the training phase as follows:

$$\theta_s^*, \theta_s'^* = \arg \min_{\theta, \theta'} \frac{1}{M} \sum_{i=1}^M \mathcal{L} \left(\mathbf{I}_a^{(i)}, g_{\theta'} \left(f_\theta \left(\mathbf{I}^{(i)} \right) \right) \right) \quad (\text{R15})$$

where \mathcal{L} is a loss function of mean square error between two inputs. Then, the j -th regenerated

data $\tilde{\mathbf{I}}_{a,s}^{(j)}$ from $\mathbf{y}^{(j)}$ in the test phase can be obtained as follows for all $j \in \{1, \dots, L\}$:

$$\tilde{\mathbf{I}}_{a,s}^{(j)} = g_{\theta_s^*} \left(f_{\theta_s^*}(\mathbf{I}^{(j)}) \right) \quad (\text{R16})$$

This can be understood as an attempt to maximize the lower bound on mutual information $M(\mathbf{I}_a; \mathbf{I})$, or as an attempt to find a manifold where \mathbf{I} represents the data into a low dimensional latent space corresponding to \mathbf{I}_a . It may face the problem that the stochastic feature of \mathbf{I}_a to be restored is too complex to regenerate or generalize. Especially in the case of spectrometer, this problem can become evident: if the autoencoder is trained to fit certain input patterns very well, it may have poor generalizability to other input spectra.

In this case, we observe that \mathbf{e} is simpler to regenerate than \mathbf{I}_a . \mathbf{e} is not affected by the overall shape of the input spectrum; it is determined by the value of each data point or its limited neighbour data points in the spectrum. Thus, we naturally come up with the intuition that training the DAE by learning \mathbf{e} and subtracting it from \mathbf{I} , which should be more effective than learning \mathbf{I}_a directly. The training and test phases of training on noise are depicted in **Fig. R26b**. The parameters of the noise-based DAE can be optimized as follows:

$$\theta_n^*, \theta_n'^* = \arg \min_{\theta, \theta'} \frac{1}{M} \sum_{i=1}^M \mathcal{L} \left(\mathbf{e}^{(i)}, g_{\theta'} \left(f_{\theta}(\mathbf{I}^{(i)}) \right) \right) \quad (\text{R17})$$

Notice that the only difference from (1) is that $\mathbf{I}_a^{(i)}$ is replaced by $\mathbf{e}^{(i)}$. During training phase, the $\mathbf{e}^{(i)}$ is derived by subtracting the ground truth from the input sample $\mathbf{I}^{(i)}$. In test phase, the ground truth is no longer needed by employing the trained DAE to predict $\mathbf{e}^{(i)}$. Let $\tilde{\mathbf{I}}_{a,n}^{(j)}$ denote the j -th regenerated data, which can be represented as follows for all $j \in \{1, \dots, L\}$:

$$\tilde{\mathbf{I}}_{a,n}^{(j)} = \mathbf{I}^{(j)} - g_{\theta_n'^*} \left(f_{\theta_n^*}(\mathbf{I}^{(j)}) \right) \quad (\text{R18})$$

To verify that the noise-based DAE improves the efficiency of denoising interferograms, we conducted verification through two aspects:

- 1) When the DAE is trained on a certain type of input spectrum, the noise-based DAE exhibits stronger generalizability than the signal-based DAE. A training set of 600,000 samples of two-peak interferograms with 30 dB SNR is used. As shown in **Fig. R26c**, the MSE error between each pair of ground truth and predicted output is depicted. The blue colour represents the signal-based results (MSE between the pair $(\tilde{\mathbf{I}}_{a,s}^{(j)}, \mathbf{I}_a^{(j)})$). The purple colour represents the noise-based results (MSE between the pair $(\tilde{\mathbf{I}}_{a,n}^{(j)}, \mathbf{I}_a^{(j)})$). Since the model is trained specifically on two-peak data, both DAEs show great performance and are very close. However, when the model is extended to unseen data in the training set, such as three-peak and four-peak data, although both models show generalizability, the noise-based model is superior, with a lower level of MSE. The reconstructed interferograms and the reconstructed spectra are shown in **Figs. R26d**, where the distinct areas where the noise-based DAE predicts more accurately are circled in red. Cases are depicted where the signal-based DAE fails to reconstruct the spectrum, while the noise-based DAE successfully reconstructs the spectra thanks to the improved accuracy in predicting the input. The model is further extended to various noise levels, and we observe that the performance superiority and generalizability of the noise-based model persist, as shown in **Fig.**

R26e.

- 2) By training on a mixture of various input spectrum types, we avoid overfitting to a certain type (like the two-peak in aspect 1), and significantly reducing the required data volume. A mixed dataset of two-peak, three-peak, and four-peak data is used, with 10,000 samples for each type (representing a $10\times$ decrease in data volume and training time). Since the model now will not overfit to a certain type of input spectrum, but rather focus on learning the noise pattern, the MSE is further decreased. As shown in **Fig. R26f**, compared to training on two-peak data only, the MSE error is decreased to a level below 0.001 for all data included in the training set. This method can be extended to other input spectra with high generalizability. For example, on a testing five-peak dataset, the MSE is 0.0009, and on a testing broadband spectrum, the MSE is 0.0014, with almost precisely reconstructed input data (**Fig. R26g**, the reconstructed spectra are also shown). A more diverse dataset is always preferred, not because we need to see the input spectrum shape to reconstruct it, but to prevent the network from overfitting to one type of input data. **Fig. R26h** shows the generalizability to multiple noise levels. If the SNR level difference exceeds 10 dB, generalizability will decrease. However, in practical situations, it is acceptable to test the approximate noise level and select the appropriate model.

With these two demonstrations, we showcase the capability of our noise-based autoencoder scheme to adapt to unseen input waveforms without the need for retraining for each specific kind of pattern.

Fig. R26 Noise-oriented training of the denoising autoencoder. **a** Architecture of signal-oriented training/test. **b** Architecture of noise-oriented training/test. **c** MSE comparison between signal- and noise-oriented training schemes when training on two-peak data only, and extending to three-peak and four-peak data to evaluate generalizability. **d** Predicted output of the denoising autoencoder and the reconstructed spectrum. **e** Generalizability across multiple SNR levels. **f** MSE comparison of training on one type of data only versus training on a mixed dataset. A mixed dataset with diverse input patterns is preferred as it prevents the model from overfitting to a single input signal pattern and encourages learning the noise pattern. **g** Predicted output of the denoising autoencoder, which shows great generalizability to unseen data in the training set, including five-peak and broadband datasets, with an MSE level of 0.001, allowing for nearly lossless reconstruction of the input spectrum. **h** Generalizability to multiple noise levels. If the SNR level difference exceeds 10 dB, generalizability will decrease. However, in practical situations, it is acceptable to test the approximate noise level and select the appropriate model.

For the reconstruction, we improve its generalizability by changing the lasso method to the elastic-net method. The corresponding description in **the third paragraph of the Spectrum reconstruction part of the Results section** is revised accordingly as below:

A regression model incorporating both L_1 -norm and L_2 -norm, commonly referred to as elastic-net, is employed by considering the spectrum sparsity and reconstruction robustness. This model effectively solves the regularization regression problem:

$$\min_{\mathbf{R}} \{ \|\mathbf{I} - \mathbf{P} \cdot \mathbf{R}\|_2^2 + \alpha \|\mathbf{R}\|_1 + \beta \|\mathbf{R}\|_2^2 \} \quad (\text{R19})$$

where the hyperparameter α and β embody the intrinsic characteristics of the spectrometer. They are critical in minimizing the mean square error (MSE) between the input spectrum \mathbf{R} and the reconstructed spectrum $\hat{\mathbf{R}}$, which is expressed as:

$$\text{MSE} = \frac{1}{n} \sum_{i=1}^n [\mathbf{R}(i) - \hat{\mathbf{R}}(i)]^2 \quad (\text{R20})$$

For input spectra comprising a limited number of peaks, the hyperparameter β is manually set to zero, considering the extreme sparsity of the signal.

Reference:

- R12. Lee, W. H., Ozger, M., Challita, U. & Sung, K. W. Noise Learning-Based Denoising Autoencoder. *IEEE Commun. Lett.* **25**, 2983–2987 (2021).

Comment 3: To address my concerns I recommend the following: 1. Demonstrate that the proposed spectrometer can reconstruct broadband and mixed sources. If further experiments are possible, this would require simply collecting interferograms (using the procedure described in the manuscript) for a broadband source (e.g., an erbium doped fiber amplifier with no input or the output of a fiber-coupled superluminescent diode) and running the proposed spectral extraction on these interferograms. The same experiment can be repeated using a mixed broadband/narrowband source simply by using a 50/50 coupler and inputting both a broadband source and a tunable laser source into the device simultaneously.

Answer 3: Thank you for the helpful suggestion. We demonstrate the reconstruction of a broadband spectrum by using Amonics C + L band ASE broadband light source. During the reconstruction of

the broadband spectrum, the β parameter of Eq. R19 is optimized via cross-validation analysis, while the α parameter is held constant. Further, we examine the case of mixed spectrum by combining the ASE source and a tunable laser source (Santec TSL-710) via a 50/50 optical coupler. Accordingly, the Eq. R19 in the manuscript is modified by introducing segmented regularization terms, as:

$$\min_{\mathbf{R}} \{ \|\mathbf{I} - \mathbf{P} \cdot (\mathbf{R}_1 + \mathbf{R}_2)\|_2^2 + \alpha \|\mathbf{R}_1 + \mathbf{R}_2\|_1 + \beta \|\mathbf{R}_2\|_2^2 \} \quad (\text{R21})$$

where \mathbf{R}_1 and \mathbf{R}_2 denote the narrowband and broadband spectral components, respectively. The optimal values of α and β are determined via cross-validation analysis^{R10}. Using the above-improved denoising autoencoder, both the broadband and mixed spectra are successfully recovered with a low relative error ε , as shown in Fig. R27.

Fig. R27 Reconstruction of **a** Broadband spectrum and **b** Mixed broadband/narrowband spectrum.

The above results and discussions have been added to **the last paragraph of the Results section** and **Figs. 5i, j** in the revised manuscript.

Reference:

R10. Qiao, Q. *et al.* MEMS-Enabled On-Chip Computational Mid-Infrared Spectrometer Using Silicon Photonics. *ACS Photonics* **9**, 2367–2377 (2022).

Comment 4: 2. Perform some dual-wavelength interferogram measurements used for resolution characterization experimentally. It appears that all dual-wavelength extraction results in this manuscript are the result of simply adding two independent single-wavelength interferograms and performing extraction on the result (with some additional noise added). While this method is understandable for the purposes of training the denoising network and initial demonstrations, I believe that for final demonstration of the system’s resolution and performance, experimental measurements of two different laser sources simultaneously should be performed. This would demonstrate robustness to nonidealities in the input that are not present in the synthetic data. Again, this experiment could be performed simply using two laser sources and a 50/50 coupler.

Answer 4: Thank you for the helpful suggestion. We have performed the reconstruction of dual-wavelength spectra again by combining two tunable laser sources (Santec TSL-550 and 710) via a 50/50 optical coupler. The two peaks show different amplitudes due to the varying laser intensity at different wavelengths and the nonideal coupling ratio of the 50/50 optical coupler. The reconstruction results show that the different amplitudes of the two peaks can be reconstructed with high accuracy, confirming the 0.4 nm resolution, as depicted in **Fig. R28**.

Fig. R28 Reconstruction of dual-wavelength spectrum with or without denoising at different wavelength spacings of **a** 1.2 nm, **b** 1.1 nm, **c** 0.4 nm, and **d** 0.3 nm.

Figures 5d-g have been updated accordingly in the revised manuscript.

Comment 5: I understand that additional experiments cannot always be performed, and that the authors may not have easy access to this equipment. If this is the case, I would still recommend that at least the broadband/mixed source experiments be simulated using the measured calibration matrix of the spectrometer, which is a straightforward process. If these tests are performed in simulation, realistic nonidealities should be included where possible (e.g. for the dual-wavelength interferogram measurements, differing amplitudes of the two laser sources).

Finally, I would like to emphasize that if the results of this additional testing or simulation indicate worse performance of the proposed system on this more diverse set of test data, I would still encourage the authors to present this data, and I would support publication of such data. Given the diverse range of spectrometer architectures and reconstruction methods being published in the field, it is crucial that publications show clearly both the strengths and weaknesses of their system, so that different approaches can be compared and further advancements can be made to improve the shortcomings of existing architectures.

If these additional experiments or simulations can be added to the manuscript, I believe that this is a very original and impactful piece of research that would be useful for many other researchers in this field, and would support publication.

Answer 5: Thank you again for your recognition of the achievement of our work. We are also very grateful for your very detailed and helpful suggestions. Following your suggestions, we have improved our experimental demonstration and denoising methodology. Satisfactory performances are achieved on the more diverse set of test data. As such, we believe that our manuscript has been substantially improved.

Dear Reviewers,

Thank you again for your great efforts, valuable comments, and helpful suggestions, all of which greatly help further improve the quality of our manuscript. We have carefully gone through all the comments and revised the manuscript accordingly. The following are our point-to-point responses along with your comments. Accordingly, the revised portions are **marked in red** in the revised manuscript.

POINT-BY-POINT RESPONSE TO THE REVIEWERS' COMMENTS

To Reviewer #1:

Comment 1: I appreciate the authors' detailed responses and revisions. The added experiments and discussions have significantly enhanced the quality of the paper, especially the exploration of the multi-stage structure and the noise-oriented autoencoder scheme. However, there are still issues that need to be addressed.

Answer 1: Thank you very much for your appreciation and expert comments. We have further revised the manuscript based on your valuable suggestions, and a detailed point-by-point response is provided below.

Comment 2: First, there remains a question about the novelty of the single-bus MEMS structure. Although the authors have provided a thorough explanation distinguishing this work from their previous publication in ACS Photonics, the changes in the DC coupling conditions, supporting structure, and the grating coupler versus edge coupler setup still seem incremental. I would like to understand whether these modifications have led to decisive performance improvements, and if so, what those improvements are. I believe this should be discussed directly in the main text, possibly even in the introduction.

Answer 2: Thank you for your insightful comments and for giving us the opportunity to clarify the novel contributions of our work. We appreciate your concern about the perceived incremental nature of the changes to the single-bus MEMS structure and agree that it is crucial to articulate the significance of these modifications.

The primary innovation in the physical structure is the strategic reduction of the MEMS tuning range, which is now directly supported by the release of the BOX layer. This innovation is not merely incremental; it represents a paradigm shift in the fabrication process. By eliminating the complex and low-yield flip-chip bonding process, we have paved the way for low-cost mass production of the device, thereby imbuing the MEMS spectrometer with practical significance and opening up possibilities for multi-stage development in the future. This modification is directly linked to a substantial improvement in the device's performance, particularly in terms of spectral reconstruction. Our careful investigation into the relationship between device tuning range and spectral reconstruction performance has led to the discovery that a moderate decoupling condition, rather than a fully decoupled one, yields the best reconstruction resolution. This finding is not only novel but also provides a valuable guideline for the design and optimization of future MEMS spectrometers.

Nevertheless, we would like to reiterate that the principal motivation of our work lies in the synergistic integration of the MEMS spectrometer with the CAED mechanism. This combination is transformative as it achieves two critical objectives: ultra-low power consumption and high-resolution performance in noisy environments. The physical structure innovations are essential, but it is their interaction with the CAED that markedly distinguishes our work. This synergy is the cornerstone of our contribution, offering a significant leap forward in the practical application of MEMS spectrometers.

In response to your suggestion, we have included more discussions in the main text to clearly state our contributions. We have emphasized the strategic reduction of the MEMS tuning range and its impact on fabrication efficiency and reconstruction performance. Additionally, we have highlighted the innovative integration with the CAED mechanism and its role in achieving the dual goals of low power consumption and high resolution in adverse conditions. Specifically,

- 1) A detailed discussion on the strategic reduction of the MEMS tuning range has been provided in **lines 171-177 of the Results section**:

“We investigate the relationship between the device tuning range and the spectral reconstruction performance using correlation analysis (see **Supplementary Note 3** and **Fig. S3**). We find that the fully decoupled condition beyond a certain range leads to a larger self-correlation width, indicating impaired reconstruction resolution. Therefore, we adopt a moderately decoupled condition, which not only guarantees satisfactory spectral resolution but also reduces the required tuning range to a level achievable by the release of the BOX layer^{R1}. This approach significantly simplifies the device configuration and fabrication process.”

- 2) A concise statement on the contributions of our work is added to **lines 87-92 of the Introduction section**:

“Through a strategic reduction of the MEMS tuning range by revealing its counterintuitive relationship with the reconstruction performance, the device yields high fabrication efficiency and optimum reconstruction resolution. On top of the ultra-low power consumption enabled by the electrostatic MEMS tuning, a CAED strategy is proposed and utilized to minimize the side effects of the associated MEMS noise on the reconstruction performance.”

We hope that these revisions directly address your concerns and provide a clear and comprehensive view of the novel aspects of our work. We hope that these discussions will illustrate the significant performance improvements and the broader implications of our innovations in the field of MEMS spectrometers.

Reference:

- R1. O’Brien, G., Monk, D. J. & Lin, L. MEMS cantilever beam electrostatic pull-in model. *Proc. SPIE* **4593**, 31–41 (2001).

Comment 3: Drawing inspiration from multi-stage spectrometers based on photonic circuits, the authors propose a multi-stage MEMS-based spectrometer. This approach is supported by simulations of dual-peak signal reconstruction, which demonstrate performance improvements. Given that the single-channel MEMS spectrometer's performance presented in the paper is moderate, the multi-stage MEMS spectrometer concept is indeed worth further exploration. Ideally, a multi-stage device should be fabricated and experimentally validated, but given the significant workload, a more extensive simulation-based investigation would be acceptable. Specifically, the simulations could be expanded in two ways: 1) by modelling a wider variety of input spectra, and 2) more importantly, by examining the impact of noise in a multi-stage system—whether there might be cumulative effects and how the proposed noise reduction algorithms would perform under such conditions. These points are crucial and deserve detailed discussion in the main text.

Answer 3: Thank you for the helpful suggestion. Following your suggestion, we have further investigated the multi-stage MEMS spectrometer through simulation. First, the SNR of this multi-stage device is estimated to be 19 dB, following the noise analysis method described in **Supplementary Note 7**. The higher noise level is due to the cascaded structure and longer total coupling length. Next, white noise according to the 19 dB SNR is added to the interferograms, which are then used for the spectrum reconstructions. As shown by the dual-wavelength spectrum reconstruction results in **Figs. R1a-d**, our current denoising autoencoder improves the reconstruction resolution from 90 to 40 pm, approaching the noise-free value of 15 pm. The denoising performance is expected to be further enhanced by accordingly optimizing the design of the autoencoder network. We further conduct the reconstructions of diverse input spectra, including triple-wavelength, broadband, and mixed broadband/narrowband spectra. As shown in **Figs. R1e-g**, all these spectra can be reconstructed with a low relative error ϵ .

Fig. R1 Performance of the multi-stage MEMS spectrometer under 19 dB SNR. **a-d** Reconstruction of dual-wavelength spectrum with or without denoising at different wavelength spacings of **a** 90 pm, **b** 85 pm, **c** 40 pm, and **d** 35 pm. **e-g** Reconstruction of a diverse range of incident spectra: **e** triple-wavelength spectrum, **f** broadband spectrum, **g** mixed broadband/narrowband spectrum.

The above results and discussions have been added to **lines 408-412 of the Discussion and Conclusion section** (as below) and **Supplementary Note 16** in the revised manuscript.

“As a pioneer in MEMS spectrometers, our device, although limited to the simplest case, i.e., a single physical stage, can see significant performance improvements by further leveraging a multi-stage structure. Using a 3-stage design, the reconstruction resolution is improved by one order of magnitude and high reconstruction accuracy is maintained for diversified input spectra. Meanwhile, despite the noise cumulated to 19 dB SNR in the 3-stage structure, our current denoising autoencoder still achieves over twofold resolution improvement, which could be further enhanced by accordingly optimizing the design of the autoencoder network (see details in **Supplementary Note 16**).”

Comment 4: Additionally, the paper could benefit from further discussions on topics such as the device's sampling time, fabrication tolerance, and temperature tolerance to provide a more comprehensive view of the work.

Answer 4: Thank you for the helpful suggestion. We further discuss our device’s sampling time, fabrication tolerance, and temperature tolerance below:

Sampling time: The MEMS tuning is conducted by using a semiconductor characterization system (Keithley 4200-SCS) with a sampling time grid of ~ 0.1 s. The total sampling time is ~ 6.4 s given 64 sampling steps. The sampling process can be accelerated by synchronizing the electrical voltage scanning and optical power detection with a shared trigger signal^{R2}. This information has been added to **the Device characterization part of the Methods section** in the revised manuscript.

Fabrication tolerance: The deviation of waveguide widths (δW_{WG}) is the primary source of fabrication defects, which will lead to differences in the calibration matrix. Nonetheless, since the calibration matrix of each device will be collected prior to its use for spectrum reconstruction, such differences among devices may be inconsequential so long as some key figures of merit remain with marginal fluctuations. There are two crucial features in the spectrum reconstruction, i.e., the ability to identify fine spectral details and the decorrelation among all wavelength channels, which can be quantified by the self-correlation function [$C(\Delta\lambda)$] and singular values (σ_i) of the calibration matrix^{R3}. We compare the $C(\Delta\lambda)$ and σ_i between the spectrometers with or without δW_{WG} of ± 20 nm, which is attainable for most photonic foundries. As depicted in **Fig. R2**, the self-correlation width shows deviation within one wavelength step of 0.1 nm, and σ_i also does not show significant change, indicating satisfactory robustness against fabrication defects.

Fig. R2 Tolerance analysis of fabrication errors. Calculated **a** self-correlation functions [$C(\Delta\lambda)$] and **b** singular values (σ_i) of the calibration matrices with waveguide width deviations δW_{WG} of ± 20 nm.

The above results and discussions have been added to **Supplementary Note 13** in the revised manuscript.

Temperature tolerance: As has been presented in Supplementary Note 12, our device can maintain the denoised resolution of 0.4 nm across a wide temperature range of 10-70 °C, as long as the calibration matrix at each temperature is collected. Another aspect of the temperature tolerance is the temperature fluctuation range in which the input signal can still be recovered to the accuracy of the spectrometer resolution without the recollection of the calibration matrix^{R2,4}. To estimate this aspect of temperature tolerance, we simulate the interferogram of a single-wavelength spectrum with a randomly selected wavelength ($\lambda = 1532$ nm) at different temperatures by changing the

refractive index of the Si according to the thermo-optic coefficient, $dn/dT \approx 1.8 \times 10^{-4} \text{ K}^{-1}$ ^{R5}. We then use the calibration matrix obtained with no temperature variation to perform the spectrum reconstructions. As shown in **Fig. R3**, with a temperature variation of $\pm 8 \text{ }^\circ\text{C}$, the input signal can still be recovered to the accuracy of the spectrometer resolution, i.e., with the offset of center wavelength remaining within $\pm 0.4 \text{ nm}$.

Fig. R3 Single-wavelength spectrum reconstruction at different temperatures without recollection of the calibration matrix.

The above results and discussions have been added to **Supplementary Note 12** in the revised manuscript.

References:

- R2. Yao, C. *et al.* Integrated reconstructive spectrometer with programmable photonic circuits. *Nat. Commun.* **14**, 6376 (2023).
- R3. Xu, H., Qin, Y., Hu, G. & Tsang, H. K. Cavity-enhanced scalable integrated temporal random-speckle spectrometry. *Optica* **10**, 1177–1188 (2023).
- R4. Redding, B., Liew, S. F., Sarma, R. & Cao, H. Compact spectrometer based on a disordered photonic chip. *Nat. Photonics* **7**, 746–751 (2013).
- R5. Komma, J., Schwarz, C., Hofmann, G., Heinert, D. & Nawrodt, R. Thermo-optic coefficient of silicon at 1550 nm and cryogenic temperatures. *Appl. Phys. Lett.* **101**, 041905 (2012).

To Reviewer #2:

Comment 1: As for my previous comments, I could see the authors trying to address them in a high quality, such as additional measurements about temperature robustness, reconstructions of complex spectra and additional analysis regarding the SNR. Overall, I am satisfied with current revision. But as stated by the authors, the MEMS based spectrometer is still accompanied with several drawbacks compared with its counter-parts, such as high voltage, large footprint etc. Therefore, I suggest the authors precisely positioning the suitable applications of such spectrometer, instead of presenting it as a general integrated spectrometers.

Answer 1: Thank you very much for your appreciation of our efforts and the constructive suggestion. As pointed, MEMS spectrometer is a promising yet nascent technology, offering plenty of room for further development with regard to footprint and operation voltage, when compared to its counterparts relying on thermo-optic tuning, which have been more intensively developed in the past few years. Therefore, as suggested, instead of presenting its application as a general integrated spectrometer, we position the suitable applications of the MEMS spectrometer more precisely by revising or removing some statements:

1) The last sentence of Abstract:

“Our technology is envisaged to provide a powerful and cost-effective solution for applications requiring rapid and accurate spectral analysis.”

is revised to:

“Our technology is envisaged to provide a powerful and cost-effective solution for applications requiring **accurate, broadband, energy-efficient, and noise-robust** spectral analysis.”

2) The last sentence of Introduction, which describes general applications:

“Our approach offers a potential pathway toward the realization of robust and cost-effective on-chip spectrometers suitable for widespread applications, including portable, handheld, and wearable scenarios.”

is **removed**.

The applications are specified in **the last paragraph of Introduction:**

“The proposed CAED-facilitated MEMS spectrometer presents a promising solution for **broadband** high-resolution spectral analysis in applications demanding precision, **power efficiency**, and noise resilience, **such as personal healthcare, remote sensing, and marine research.**”

To Reviewer #3:

Comment 1: After reviewing the authors' response to my previous comments, I believe the authors have faithfully modified their manuscript to account for my concerns and questions. In its current form, I believe the manuscript is scientifically sound with sufficient evidence presented to support the authors' claims. Overall, it is my opinion that the results presented here are highly relevant and of benefit to the broader community. Therefore, I recommend publication of this modified manuscript in its current form.

Answer 1: Thank you very much for your appreciation of our revision and support for the publication. Thank you again for your great efforts, valuable comments, and helpful suggestions throughout the whole review process.

To Reviewer #4:

Comment 1: The authors' revisions have added a substantial amount of information to the paper and have satisfactorily addressed my previous comments. I support publication once the following questions are answered/clarified in the manuscript:

Answer 1: Thank you very much for your appreciation of our efforts and expert comments. We have further revised the manuscript based on your valuable suggestions, and a detailed point-by-point response is provided below.

Comment 2: The authors include a reconstruction of the emission spectrum from an Amonics ASE C + L band source.

1a. (minor) Could the authors provide the model number of this source so readers can more easily compare the spectrum measured in the paper to the spectrum specified by the manufacturer?

1b. Referring to the Amonics website, it seems that their C+L band ASE sources cover a much broader range than the reconstructions shown in Fig. 5i-j (this is also evidenced by the cut-off peak at the short wavelengths in these figures). Given the wide bandwidth of the spectrometer, why was the entire reconstruction not shown in these figures?

Answer 2: Thank you for the comment.

1a. The model number of this source is ALS-CL-13-B-FA. This information has been added to **the Device characterization part of the Methods section** in the revised manuscript.

1b. According to the Amonics website, the spectral range of the ASE sources is 1528-1608 nm, which is just slightly broader than the range of 1532-1600 nm shown in Figs. 5i, j. Since our collected calibration matrix covers up to 1600 nm, the spectrum reconstruction can be performed up to 1600 nm as well. For the short wavelength side, we now have extended the reconstruction to 1520 nm, where the spectral density of the ASE source has attenuated to almost zero. Figs. 5i, j has been revised as **Fig. R4** below.

Fig. R4 Reconstruction of **a** broadband spectrum, **b** mixed broadband/narrowband spectrum.

Comment 3: The authors include a spectrum reconstruction of the absorption spectrum of N-methylaniline, which is a good way to demonstrate the utility of the spectrometer. However, I am confused by the description of how this spectrum is measured, or rather, mimicked: “Considering the limitation in the experimental setup, we mimic the input absorption spectrum of N-methylaniline

by tailoring the tunable laser intensity at each wavelength.” Does this mean that the authors simply collected dozens of input interferograms from narrowline sources, adjusted their amplitude, added them together, and performed the reconstruction? If so, I would argue that this is not a realistic broadband reconstruction and I question its inclusion in the paper.

Answer 3: Thank you for the insightful comment. Due to the limitation in experimental setup and time, the absorption spectrum of *N*-methylaniline is mimicked by adjusting the tunable laser intensity at each wavelength. Based on the widely-adopted linear superposition assumption^{R6-8}, the interferogram corresponding to the absorption spectrum of *N*-methylaniline is obtained by a summation of all the interferograms measured at each wavelength, which is then used for the absorption spectrum reconstruction. As you suggested, we have removed this part in the revised manuscript.

Reference:

- R6. Xu, H., Qin, Y., Hu, G. & Tsang, H. K. Scalable integrated two-dimensional Fourier-transform spectrometry. *Nat. Commun.* **15**, 436 (2024).
- R7. Kong, L. *et al.* Single-Detector Spectrometer Using a Superconducting Nanowire. *Nano Lett.* **21**, 9625–9632 (2021).
- R8. Pohl, D. *et al.* An integrated broadband spectrometer on thin-film lithium niobate. *Nat. Photonics* **14**, 24–29 (2020).

Comment 4: The authors mention using modified regularization for the broadband reconstruction (Eq. 14 in the manuscript). Are high quality reconstructions of all sources obtained if this same regularization is used for all reconstructions? Or do the results shown in this work rely on picking the “best” regularization method depending on the nature of the spectrum? I understand that this same mixed broadband/narrowline regularization has been demonstrated in previous works (as cited by the authors), and can be used on diverse inputs - if this is the case, I recommend the authors use (14) on all reconstructions in the paper. If not, the authors should emphasize this point, as requiring a priori information about the nature of an unknown spectrum to pick the best regularization method/hyperparameters is a major limitation for practical use.

Answer 4: Thank you for the insightful comment. The segmented regularization terms $\alpha\|\mathbf{R}_1 + \mathbf{R}_2\|_1$ and $\beta\|\mathbf{R}_2\|_2^2$ in Eq. 14 only set a *general range* of possible characteristics that may occur in a spectrum. It does not require specific knowledge of spectral contents before measurement, as most naturally occurring features can be covered by these terms. In addition, the hyperparameters α and β can be automatically optimized via cross-validation. The optimization process does not require any manual parameter selection. **Using Eq. 14, it is feasible to attain the same level of reconstruction accuracy for diverse input spectra without any priori.** Similar regularized iterative methods have been used in prior studies^{R2,3,6,9}.

To be specific, $\alpha\|\mathbf{R}_1 + \mathbf{R}_2\|_1$ with L₁-norm provides sparsity regularization that compresses spectra into discrete lines, whereas $\beta\|\mathbf{R}_2\|_2^2$ with L₂-norm provides Tikhonov regularization that smoothens spectra into continuous bands. Generally, a spectrum is either discrete or continuous; therefore, the proposed segmented regularization terms can cover most naturally occurring spectral features. A mixed spectrum can also be rebuilt using this method with non-zero values for both α

and β . By using cross-validation, these hyperparameters can be automatically optimized without any manual selection.

To further verify the effectiveness of the proposed modified regularization method, the solvability of Eq. 14 is assessed with the Picard plot. Based on truncated singular value decomposition (SVD), the naïve solution to the inverse problem can be written as:

$$\mathbf{R} = \sum_{i=1}^{N_s} \frac{\mathbf{u}_{(i)}^T \mathbf{I}}{\sigma_i} \mathbf{v}_i \quad (\text{R1})$$

where \mathbf{u}_i denotes the i -th left singular vector, \mathbf{v}_i denotes the i -th right singular vector, σ_i denotes the singular value, and N_s denotes the sampling channel number. Equation R1 can be interpreted as that the naïve solution of the input spectrum ($\widehat{\mathbf{R}}$) is formed on the basis of right singular vectors ($\mathbf{v}_{(i)}$) that are weighted by SVD coefficients ($\mathbf{u}_{(i)}^T \mathbf{I} / \sigma_i$). Thus, to prevent an infinite integral and retrieve all spectral information, it is necessary to ensure that the SVD coefficient levels off to a finite value, also known as the Picard condition^{R10}; otherwise, the integral of channels will be infinite and the iterative process will suffer from a poor convergence. Four distinct types of spectra (\mathbf{R}) are used for testing: plateau, Gaussian, spike, and random functions, as shown in the first row of Fig. R5a. Figures R5b, c show the calculated absolute values of SVD coefficients ($|\mathbf{u}_{(i)}^T \mathbf{I} / \sigma_i|$) derived from the simulation and experimental results, respectively. The curves of sampling components ($|\mathbf{u}_{(i)}^T \mathbf{I}|$) and singular values (σ_i) are also displayed in the same plot. It can be found that the calculated ($|\mathbf{u}_{(i)}^T \mathbf{I} / \sigma_i|$) does not overall increase, indicating that the Picard condition is fulfilled and a convergent solution can always be reached.

Fig. R5 Picard plots. **a** four types of testing spectra (\mathbf{R}). **b, c** Picard plots for the **b** theoretical and **c** experimental transmission matrix.

From another perspective, **the priori knowledge of spectral features also helps to accelerate the process**. For instance, if we already know that the spectrum solely contains discrete spectral lines, then β can be set as zero and only α needs to be optimized. In other words, α (β) will automatically overwhelm β (α) for narrowband (broadband) spectra during the cross-validation procedure, and **the presetting of hyperparameters in the reconstruction of sole narrowband or broadband spectra is only for shortening the optimization time**.

As you recommended, we have reperformed all the reconstructions in Figs. 3 and 5 using Eq. 14 (Eq. 8 in the revised manuscript). The corresponding descriptions have also been revised accordingly in lines 219-227 of the Results section. The above discussions have also been added as **Supplementary Note 6** in the revised manuscript.

References:

- R2. Yao, C. *et al.* Integrated reconstructive spectrometer with programmable photonic circuits. *Nat. Commun.* **14**, 6376 (2023).
- R3. Xu, H., Qin, Y., Hu, G. & Tsang, H. K. Cavity-enhanced scalable integrated temporal random-speckle spectrometry. *Optica* **10**, 1177–1188 (2023).
- R6. Xu, H., Qin, Y., Hu, G. & Tsang, H. K. Scalable integrated two-dimensional Fourier-transform spectrometry. *Nat. Commun.* **15**, 436 (2024).
- R9. Xu, H., Qin, Y., Hu, G. & Tsang, H. K. Breaking the resolution-bandwidth limit of chip-scale spectrometry by harnessing a dispersion-engineered photonic molecule. *Light Sci. Appl.* **12**, 64 (2023).
- R10. Hansen, P. C. The discrete picard condition for discrete ill-posed problems. *BIT* **30**, 658–672 (1990).

Comment 5: Fig. 5j shows a great quality mixed broadband/narrowband reconstruction, which is nice to see. Can the authors give more information regarding the OSA settings used to obtain the reference spectrum? In particular, it would be useful to know the resolution setting of the OSA, as this can impact the relative height between the narrowband peak and the broadband features.

Answer 5: Thank you for the comment. The resolution of the OSA is set to be 20 pm, which is the finest value that can be offered. The reference spectrum is created by resampling the raw data into a 3001-point sequence with a coarser resolution of 100 pm.

Comment 6: I may have missed it, I would like clarification regarding exactly what “input spectrum” refers to in each figure. Was the “input spectrum” always measured using an OSA? Certain figures which make me especially curious about this are 3b-3c, 5d-5g, etc. (reconstructions of narrowband sources). It seems like the “input spectrum” has the exact same resolution and positioning of x-axis points as the reconstruction. Is this an OSA measurement with the measurement parameters chosen to be the same as the reconstruction? Or is this a synthetic estimate of the input spectrum with delta functions simply placed at the wavelengths that the tunable lasers were set to?

Answer 6: Thank you for the comment. All the “input spectrum” in Figs. 3 and 5 are measured using the OSA. The recorded spectra from OSA have a fine resolution of 20 pm. The reference input spectra shown in Figs. 3 and 5 are created by resampling the raw data into a 3001-point sequence

with a coarser resolution of 100 pm (aligning with the wavelength step used for calibration and reconstruction), for the ease of the quantification of reconstruction accuracy by calculating the widely adopted metric named relative error ε between the input and reconstructed spectrum.

The above information has been added to **the Device characterization part of the Methods section** in the revised manuscript.

Dear Reviewers,

Thank you again for your great efforts, valuable comments, and helpful suggestions, all of which greatly help further improve the quality of our manuscript. We have carefully gone through all the comments and revised the manuscript accordingly. The following are our point-to-point responses along with your comments. Accordingly, the revised portions are **marked in red** in the revised manuscript.

POINT-BY-POINT RESPONSE TO THE REVIEWERS' COMMENTS

To Reviewer #1:

Comment 1: Again, I appreciate the authors' detailed responses in the previous revision round. After carefully reviewing the paper, a few important issues still need to be addressed:

Answer 1: Thank you very much for your appreciation and expert comments. We have further revised the manuscript based on your valuable suggestions, and a detailed point-by-point response is provided below.

Comment 2: In the Abstract and Introduction, the authors added some new statements, such as: “The proposed CAED-facilitated MEMS spectrometer presents a promising solution for broadband high-resolution spectral analysis in applications demanding precision, power efficiency, and noise resilience, such as personal healthcare, remote sensing, and marine research.” However, this might be an overstatement. The mechanical noise is an inherent issue for MEMS, not for other free-space or PIC-based spectrometers. The authors could argue that their CAED scheme helps mitigate noise in MEMS, but noise tolerance should not be presented as a key selling point. Also, in general, I don't think waveguide-based spectrometers are suitable for applications like remote sensing or health care monitoring, as the micrometer-scale mode size limits the amount of optical power that can be coupled. In real-world scenarios, such as remote sensing, where the spectra to be detected are typically scattered light with low power density, the proposed MEMS spectrometer may struggle to capture any meaningful signal. Thus, I strongly recommend revising these statements to avoid potential misunderstandings.

On the other hand, using “high-resolution” in the title seems somewhat inappropriate, as many on-chip spectrometers have already achieved picometer-level resolution, while the device presented here only reaches 0.2 nm. Please consider revising this.

Answer 2: Thank you for the constructive suggestions. As you suggested, relevant statements in the Abstract and Introduction are revised to more precisely position the suitable applications of our device:

1) The last sentence of Abstract:

“Our technology is envisaged to provide a powerful and cost-effective solution for applications requiring accurate, broadband, energy-efficient, and noise-robust spectral analysis.”

is revised to:

“Our technology is envisaged to provide a powerful and cost-effective solution for applications requiring accurate, broadband, and energy-efficient spectral analysis.”

2) The penultimate sentence of Introduction:

“The proposed CAED-facilitated MEMS spectrometer presents a promising solution for broadband high-resolution spectral analysis in applications demanding precision, power efficiency, and noise resilience, such as personal healthcare, remote sensing, and marine research.”

is revised to:

“The proposed CAED-facilitated MEMS spectrometer presents a promising solution for

broadband high-resolution spectral analysis in applications demanding precision and power efficiency.”

Also, as you suggested, the title is revised to “Denoising-autoencoder-facilitated MEMS computational spectrometer with enhanced resolution on a silicon photonic chip”.

Comment 3: Additionally, the authors claim a 300 nm broadband bandwidth, but it seems that there is only one dual-peak laser experiment to support this, while all the remaining reconstruction experiments focus on the long wavelength region. I recommend adding some simulation results (especially for continuous spectra) to demonstrate that the device still performs well in the short wavelength range.

Answer 3: Thank you for the helpful suggestion. We have performed the reconstruction of a continuous spectrum spanning the entire 300 nm bandwidth from 1.3 to 1.6 μm by simulation. White noise corresponding to 30 dB SNR is added to the simulation. As shown in Fig. R1, our CAED-facilitated MEMS spectrometer is able to accurately reconstruct the spectrum with a low relative error ε of 0.046.

Fig. R1 Simulated reconstruction of a continuous spectrum spanning from 1.3 to 1.6 μm .

The above result and discussion have been added as **Supplementary Note 14** in the revised manuscript.

Comment 4: In lines 381 to 383, the authors state: “In order to show the efficacy of our spectrometer in real-life applications, we further demonstrate the reconstruction of the absorption spectrum of N-methylaniline, which possesses a well-defined absorption fingerprint near 1.5 μm (see Supplementary Note 14)”. Such a statement is vague and gives the impression that the authors conducted an experiment, but upon reviewing Supplementary Note 14, I found this to be merely a simulation. Additionally, the description of the simulation is confusing. The authors mention adjusting the tunable laser intensity at each wavelength to mimic N-methylaniline’s absorption spectrum, but it’s unclear whether they are reconstructing a discrete or continuous spectrum, which is a significant distinction for computational spectrometers. Therefore, I recommend that the authors remove this section.

Answer 4: Thank you for the suggestion. Due to the limitation in experimental setup and time, we performed the reconstruction of the absorption spectrum of N-methylaniline in a semi-experimental manner. The intensity of the tunable laser at each wavelength was adjusted following N-

methylaniline's absorption spectrum. Based on the widely-adopted linear superposition assumption^{R1-3}, the interferogram corresponding to *N*-methylaniline's absorption spectrum was obtained by a summation of all the interferograms measured at each wavelength, which was then used for the absorption spectrum reconstruction. Nonetheless, we agree that such a demonstration is not fully experimental and somewhat confusing. Therefore, as you suggested, we have removed this part in the revised manuscript.

References

- R1. Xu, H., Qin, Y., Hu, G. & Tsang, H. K. Scalable integrated two-dimensional Fourier-transform spectrometry. *Nat. Commun.* **15**, 436 (2024).
- R2. Kong, L. *et al.* Single-Detector Spectrometer Using a Superconducting Nanowire. *Nano Lett.* **21**, 9625–9632 (2021).
- R3. Pohl, D. *et al.* An integrated broadband spectrometer on thin-film lithium niobate. *Nat. Photonics* **14**, 24–29 (2020).

Comment 5: A structural suggestion: I recommend adding a figure to the main text to better explain the multi-stage design, as currently, all the details are in the supplementary materials, making it difficult for readers to grasp the concept intuitively. A dedicated figure in the main text would visually illustrate the multi-stage design and further highlight its role in improving the spectrometer's performance.

Answer 5: Thank you for the suggestion. As you suggested, we have moved the major results to the main text. The newly added **Fig. 7** is as shown in **Fig. R2** below. Corresponding descriptions have also been added to the **second paragraph of the Discussion and Conclusion** section in the main text.

Fig. R2 Investigation of multi-stage design. **a** Schematic illustration of the multi-stage MEMS spectrometer and voltage modulation channels. **b** Calibration matrix of the 3-stage spectrometer. **c-f** Reconstruction of dual-wavelength spectrum with or without denoising at different wavelength spacings of **c** 90 pm, **d** 85 pm, **e** 40 pm, and **f** 35 pm.

To Reviewer #4:

Comment 1: The authors' changes are thorough, thoughtful, and have satisfactorily addressed my comments. The manuscript is well-written and convincingly demonstrates the device and novel reconstruction methods. I recommend publication.

Answer 1: Thank you very much for your appreciation of our revision and support for the publication. Thank you again for your great efforts, valuable comments, and helpful suggestions throughout the whole review process.